# N-terminal syndecan-2 domain selectively enhances 6-O heparan sulfate chains sulfation and promotes VEGFA$_{165}$-dependent neovascularization

Federico Corti [1], Yingdi Wang[1], John M. Rhodes[1], Deepak Atri[1], Stephanie Archer-Hartmann[2], Jiasheng Zhang[1], Zhen W. Zhuang[1], Dongying Chen[1], Tianyun Wang[1], Zhirui Wang[2], Parastoo Azadi[2] & Michael Simons [1,3]

The proteoglycan Syndecan-2 (Sdc2) has been implicated in regulation of cytoskeleton organization, integrin signaling and developmental angiogenesis in zebrafish. Here we report that mice with global and inducible endothelial-specific deletion of Sdc2 display marked angiogenic and arteriogenic defects and impaired VEGFA$_{165}$ signaling. No such abnormalities are observed in mice with deletion of the closely related Syndecan-4 (Sdc4) gene. These differences are due to a significantly higher 6-O sulfation level in Sdc2 versus Sdc4 heparan sulfate (HS) chains, leading to an increase in VEGFA$_{165}$ binding sites and formation of a ternary Sdc2-VEGFA$_{165}$-VEGFR2 complex which enhances VEGFR2 activation. The increased Sdc2 HS chains 6-O sulfation is driven by a specific N-terminal domain sequence; the insertion of this sequence in Sdc4 N-terminal domain increases 6-O sulfation of its HS chains and promotes Sdc2-VEGFA$_{165}$-VEGFR2 complex formation. This demonstrates the existence of core protein-determined HS sulfation patterns that regulate specific biological activities.

[1] Yale Cardiovascular Research Center, Section of Cardiovascular Medicine, Department of Internal Medicine, Yale University School of Medicine, 300 George Street, New Haven, CT 06511, USA. [2] Complex Carbohydrate Research Center, The University of Georgia, 315 Riverbend Road, Athens, GA 30602, USA. [3] Department of Cell Biology, Yale University School of Medicine, New Haven, CT 06520, USA. These authors contributed equally: Federico Corti, Yingdi Wang. Correspondence and requests for materials should be addressed to M.S. (email: michael.simons@yale.edu)

Proteoglycans are a complex group of heavily glycosylated proteins that play a number of important structural and signaling roles[1]. A typical proteoglycan is made of a core protein with covalently attached glycosaminoglycan (GAG) chains. Heparan sulfate (HS) chains (a type of GAGs) are serine-linked linear polymers elongated as binary alternation of glucuronic acid and N-acetylglucosamine[2]. Regions of a growing GAG chain undergo enzymatic-mediated modifications (i.e., epimerization, N-deacetylation/sulfation, and O-sulfation) that ultimately give rise to distinct sulfation patterns. These patterns display an incredible diversity[3,4], and are responsible for a wide array of biological properties attributed to HS chains, including binding of growth factors and cytokines, interactions with the extracellular matrix, and tissue structural properties among others[5]. The regulation of the sulfation process is poorly understood, and it is assumed that, in a given cell type, a sulfation pattern of any given HS chain is the same regardless of what core protein it is on[6–8]. Indeed, whether a core protein structure or sequence can specify specific sulfation patterns remains a long-standing question in glycobiology.

Syndecans are a distinct four-member family of type-I transmembrane proteoglycans that carry HS and/or chondroitin sulfate (CS) GAG chains[9]. Syndecans are involved in a number of physiological processes including lipoprotein uptake (Sdc1 and Sdc4)[10,11], feeding behavior (Sdc3)[12], regulation of mTOR pathway[13] and endothelial cell alignment to blood flow (Sdc4)[14] among many others[15,16]. As is the case with other proteoglycans, syndecan cores are thought to have distinct binding abilities and engage in specific protein–protein interactions thereby determining functional specificity of various syndecan-dependent biological processes[17].

One of the principle roles of syndecans in endothelial cells (ECs) is facilitation of growth factor signaling, including that fibroblast growth factors (FGFs) and vascular endothelial growth factors (VEGFs). The two growth factor families signal via respective receptor tyrosine kinases (FGFRs and VEGFRs) and play key roles in blood vessel growth and maintenance during development and adult life[18–21]. VEGFR2 is the principal signaling receptor for $VEGFA_{165}$, the predominant circulating heparin-binding VEGFA isoform, and for a number of other VEGFA isoforms with different HS binding abilities[22]. In agreement with a well-described requirement for cell surface HS chains to fully activate $VEGFA_{165}$–VEGFR2 signaling[23,24], syndecans are thought to function as $VEGFA_{165}$ co-receptors by binding the growth factor and increasing its local concentration on the plasma cell membrane thereby facilitating its binding to VEGFR2[22]. Surprisingly, the specificity of a syndecan–$VEGFA_{165}$–VEGFR2 interaction has never been tested. An early study described defective vascular development in zebrafish following morpholino-mediated Sdc2 knockdown[25] that was attributed to a loss of genetic interaction with $VEGFA_{165}$. At the same time, no vascular phenotypes have been described in various mice lines with Sdc1 or Sdc4 knockouts.

To define the role of Sdc2 in vascular developmental and adult angiogenesis, we generated global and endothelial-specific Sdc2 knockout mouse lines. In agreement with the aforementioned zebrafish study, Sdc2 deletion resulted in a number of vascular developmental and adult vascular growth abnormalities consistent with reduced $VEGFA_{165}$ signaling while FGF-mediated angiogenesis remained intact. In vitro studies pinpointed the defect in VEGFA-signaling defect in $Sdc2^{-/-}$ ECs to the enhanced ability of Sdc2, but not Sdc4, HS chains to bind $VEGFA_{165}$ and form a ternary $VEGFA_{165}$–Sdc2–VEGFR2 complex. This, in turn, was traced to higher frequency of 6-O sulfation in Sdc2 HS chains. The latter finding is accounted for by a 59 aminoacid sequence in the N-terminal domain of Sdc2 that conferred the ability to specifically enhance 6-O sulfation. A Sdc4 chimera carrying this Sdc2 region demonstrated an increase in 6-O sulfation of its HS chains. Taken together these results show that a core protein sequence can determine a specific HS chains' sulfation pattern thereby regulating biological activity of these chains in particular and cellular behavior in general.

## Results

### Syndecan-2 deletion delays retinal vascular development and inhibits VEGFA-induced angiogenesis.

Mouse ES cells carrying a mutant Sdc2 knock-in allele were obtained from KOMP repository (Strain ID: $Sdc2^{tm1a(KOMP)Wtsi}$) and used to generate a mouse line carrying a Sdc2 allele with two loxP sites flanking exon 3 of the Sdc2 gene ($Sdc2^{fl/fl}$). These mice were then crossed with specific Cre-recombinase lines (Supplementary Figure 1a) and the absence of mature mRNA after Cre activation was confirmed by quantitative polymerase chain reaction (qPCR) analysis of primary mouse ECs (Supplementary Figure 1b). A cross with a CMV-Cre driver line generated a global null ($Sdc2^{-/-}$) line with the progeny appearing in the expected Mendelian ratio (Supplementary Figure 1c). Examination of aorta cross-sections and whole-retinal mounts demonstrated Sdc2 expression in smooth muscle (SMC) and endothelial cells (ECs) in arterial, venous, and capillary beds (Supplementary Figure 1e). This expression was completely abolished in $Sdc2^{-/-}$ mice (Supplementary Figure 1f).

While knockdown of Sdc2 in zebrafish led to severe vascular alterations during development[25], $Sdc2^{-/-}$ mice were born alive. A careful analysis of postnatal retinal development revealed a delay in vessel outgrowth and decreased vascular branching in $Sdc2^{-/-}$ compared to littermate controls ($Sdc2^{+/+}$) (Supplementary Figure 2a–c). In addition, $Sdc2^{-/-}$ mice displayed a significant delay in skin wound healing (Supplementary Figure 2d, e).

To more accurately investigate Sdc2 role in vascular development, we used $Pdgfb-CreER^{T2}$ and $Cdh5-CreER^{T2}$ driver lines[26,27] to induce EC-specific deletion ($Sdc2^{iPdgfb}$ and $Sdc2^{iCdh5}$). After 5 days of tamoxifen treatment (P1–P5), qPCR examination of primary lung ECs documented profoundly reduced Sdc2 mRNA levels in both mouse lines (Supplementary Figure 1b). In agreement with findings in $Sdc2^{-/-}$ mice, $Sdc2^{iPdgb}$ mice also showed a delay in retinal vessel outgrowth (Fig. 1a, b) and decreased branching (Fig. 1c, d) compared to littermate controls ($Sdc2^{fl/fl}$). Analysis of collagen IV staining detected an equal frequency of empty sleeves between $Sdc2^{iPdgb}$ and control littermates thus suggesting a defect in vessel formation rather that increased vessel pruning in $Sdc2^{iPdgb}$ (Supplementary Figure 3a, b). In support of this explanation, a reduced number of tip cells (Supplementary Figure 3c, d) as well as reduced-ECs proliferation and density were observed following endothelial-specific Sdc2 deletion (Supplementary Figure 3e–g).

Syndecan-4 (Sdc4) is structurally and evolutionary close to Sdc2[28] and is highly expressed in ECs both in vitro and in vivo[29,30], including the retinal endothelium[31]. Since the two syndecans can have overlapping functions[32–34], we investigated whether Sdc4 may also be important for vascular development. Retinas of $Sdc4^{-/-}$ mice, unlike $Sdc2^{iPdgfb}$, did not reveal any vascular abnormalities compared to littermate controls ($Sdc4^{+/+}$) nor to $Sdc2^{fl/fl}$ control mice (Fig. 1a–d).

While VEGFA is the primary growth factor driving retinal angiogenesis, other growth factors are also involved. To examine if there are growth factor specific abnormalities in $Sdc2^{-/-}$ vs. Sdc4 mutants, we employed a cornea pocket assay model which allows examination of an angiogenic response to specific growth factors[35]. $Sdc2^{iCdh5}$ mice showed a significant reduction in $VEGFA_{165}$-induced angiogenesis compared to wild-type control or $Sdc4^{-/-}$ mice (Fig. 1e, f). At the same time, both $Sdc4^{-/-}$ and

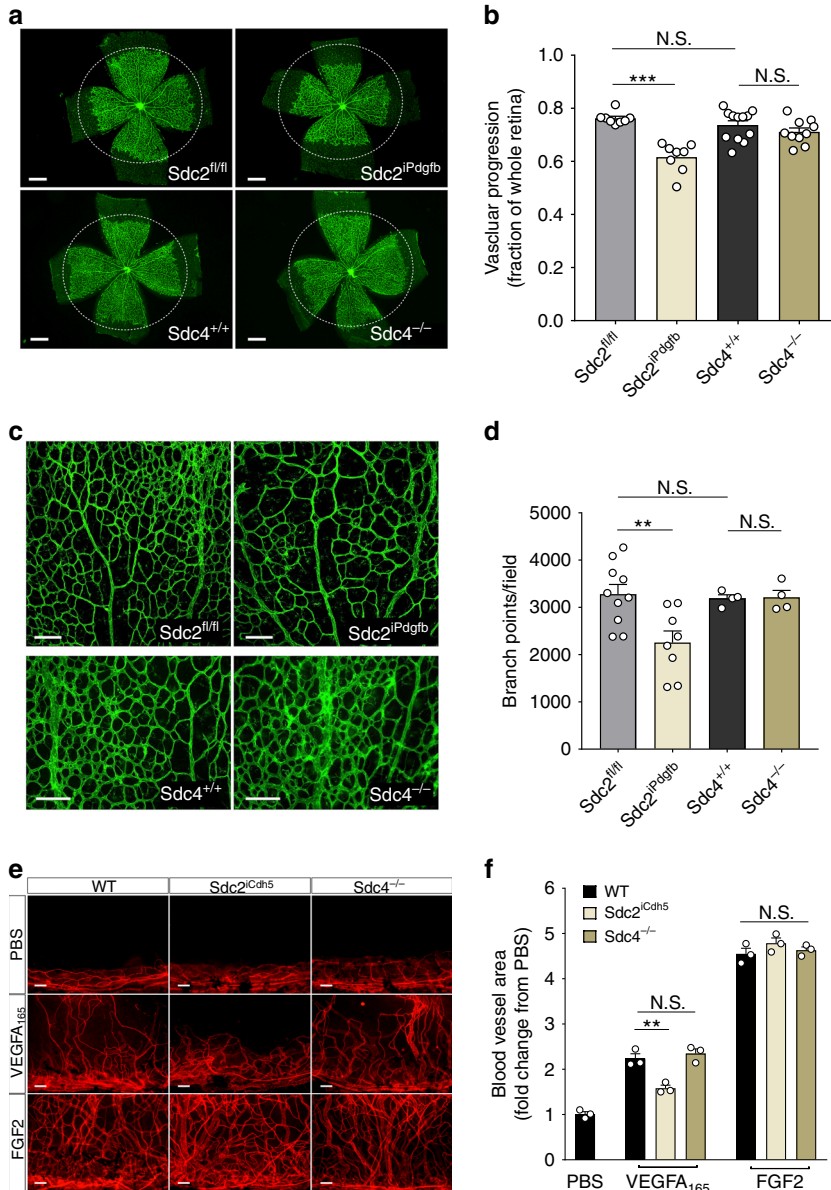

**Fig. 1** Sdc2 EC deletion leads to impaired angiogenesis. **a–d** Retinas from P6 pups were stained with isolectin B4 for specific detection of endothelium (in green). **a** Representative pictures of retinal vascular outgrowth for each genotype (500 μm scale bars). **b** Quantification of vascular progression expressed as ratio between length of vascular front and retina edge ($n = 8$–12 retinas from 4 to 6 mice, each dot corresponds to a different retina). **c** Representative pictures of vascular branching (100 μm scale bars) and quantification (**d**) ($n = 4$–10 retinas from 4 to 5 animals, each dot corresponds to a different retina). **e**, **f** PBS or indicated growth factor pellets were inserted in a cornea micro-pocket and angiogenic response was evaluated by CD31 staining (in red) after 1 week. **e** Representative pictures of cornea-angiogenic responses (100 μm scale bars) and quantification (**f**) ($n = 3$ mice for each treatment and genotype, each dot corresponds to a different cornea). Errors bars represent standard error of the mean (SEM). Statistical analysis was performed by one-way Anova with Bonferroni's multiple comparison test (N.S. not significant, $^{**}P < 0.01$, $^{***}P < 0.001$)

Sdc2[iCdh5] showed normal angiogenesis in response to FGF2 (Fig. 1e, f), suggesting that Sdc2 in specifically involved in VEGFA$_{165}$ but no FGF2 signal transduction.

We next compared arteriogenic responses in adults Sdc2[iCdh5] and Sdc4[−/−] mice. Interruption of a common femoral artery (CFA) in mice induces arteriogenesis at the site of ligation and angiogenesis in distal limb tissues[36]. The former is driven by a combination of events and requires both FGF and VEGFA signaling inputs while the latter is primarily VEGFA-dependent[36]. Analysis of blood flow recovery after CFA ligation using laser-Doppler perfusion imaging demonstrated a marked reduction in perfusion in Sdc2[iCdh5] but not Sdc4[−/−] animals, compared to controls, at multiple time points (Fig. 2a–c). Micro-

CT angiography confirmed a reduction in number of perfused arterial vessels in Sdc2[iCdh5] but not Sdc4[−/−] mice (Fig. 2d, e). The reduction was significant in both calf (at size ≤ 96 μm) (Fig. 2e, top right) and thigh vessels (at size ≤ 48 μm) (Fig. 2e, top left).

To further evaluate the Sdc2-VEGFA link, we studied the effect of Sdc2 deletion on VEGFA-driven proliferation and migration of ECs in vitro. Primary mouse ECs isolated from Sdc2[−/−] mice showed decreased proliferation and migration in response to VEGFA$_{165}$ compared to primary ECs from Sdc4[−/−] mice. At the same time, both WT and Sdc2[−/−] ECs responded similarly to FGF2 (Supplementary Figure 4a–f). Taken together, these data point to an VEGFA-specific signaling defect in Sdc2[−/−] ECs.

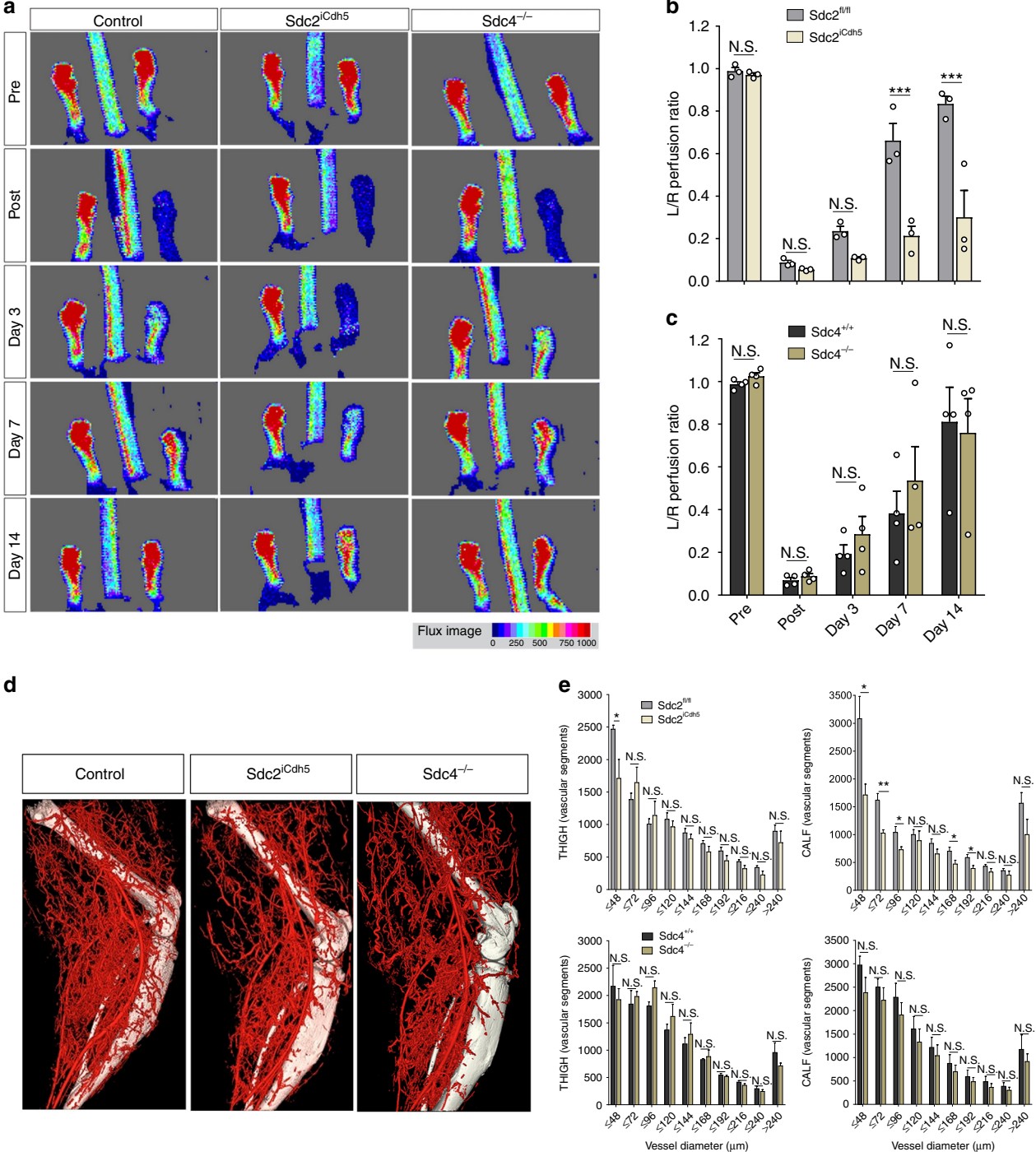

**Fig. 2** Sdc2 promotes blood flow recovery in HLI. **a–c** Left CFA (common femoral artery) was ligated and blood flow recovery was measured by laser doppler at the indicated days. Recovery at each indicated day was quantified as ratio between flow perfusion in ligated vs. contralateral artery (L/R perfusion ratio). **a** Representative pictures of blood flow recovery at different days. Pictures for Control are shown for Sdc2$^{fl/fl}$ mouse. The flux image (lower right) indicates the extent of hind limb blood flow from low (blue) to high (red). **b, c** Quantification of blood flow recovery (*n* = 3–4, each dot corresponds to a different mouse). **d, e** Micro-CT angiography was used for visualization of functional vessels (showed in red) and quantify the number of perfused vessels (grouped by lumen diameter on the *x*-axis). **d** Representative pictures of micro-Ct angiography at day 14 and quantification (**e**). Source data are provided as a Source Data file. Pictures for Control are shown for Sdc2$^{fl/fl}$ mouse. Errors bars represent SEM. Statistical analysis was performed by two-way Anova with Sidak's multiple comparison test (**b, c**) and unpaired *t* test (**e**) (N.S. not significant, $^*P < 0.05$, $^{**}P < 0.01$, $^{***}P < 0.001$)

**Syndecan-2 HS chains are required for a full VEGFR2 activation.** To study why Sdc2 deletion affects endothelial VEGFA signaling, we next examined in vitro signaling responses in primary ECs derived from various knockout mice or in HUVECs following siRNA-based knockdowns. Expression of VEGFR2, its

co-receptor neuropilin-1 (NRP1) and VE-cadherin (VE-Cad) was unchanged in Sdc2$^{-/-}$ ECs (Fig. 3a). However, in line with the in vivo data, VEGFA$_{165}$-induced VEGFR2 activation was decreased in primary Sdc2$^{-/-}$ ECs compared to WT ECs as shown by a significant reduction in VEGFR2 Y1175 site

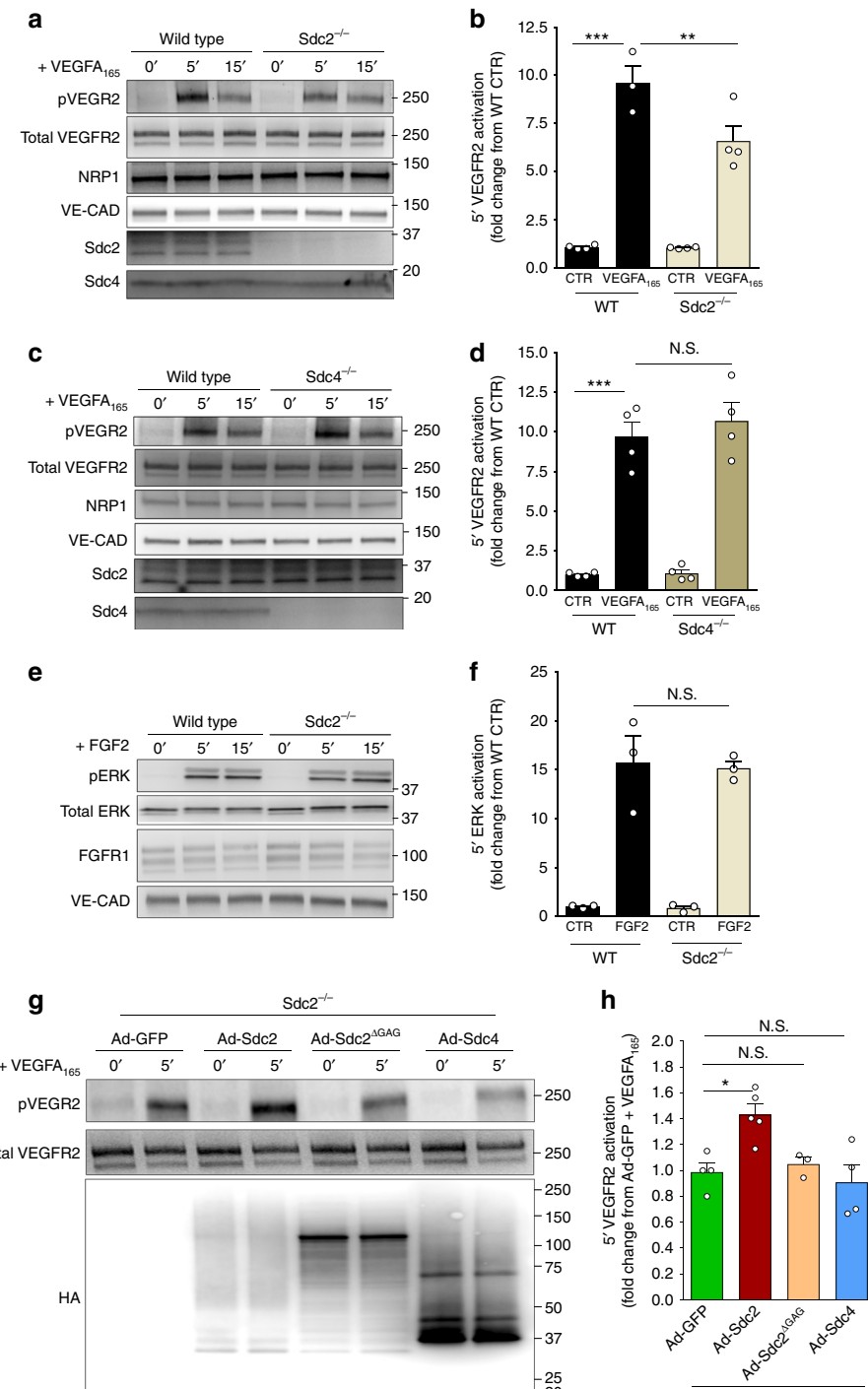

**Fig. 3** Maximal VEGFR2 activation requires Sdc2 HS chains. **a–f** Primary mouse ECs from indicated genotypes were serum-starved for 12 h and then stimulated for 5 and 15 min. Activation of VEGFR2 (pVEGFR2) was assessed after stimulation with VEGFA$_{165}$ (50 ng/ml) in Sdc2$^{-/-}$ ECs (**a** representative picture, **b** quantification) or Sdc4$^{-/-}$ ECs (**c** representative picture, **d** quantification). FGFR1 signaling in Sdc2$^{-/-}$ EC was assessed by stimulation with FGF2 (20 ng/ml) and evaluation of ERK activation (**e** representative picture, **f** quantification) ($n = 3$–4 for VEGFA$_{165}$, $n = 3$ for FGF2). **g**, **h**, Sdc2$^{-/-}$ ECs were transduced with adenovirus expressing the indicated construct for 16 h (MOI = 1–2), starved for 12 h followed by stimulation with VEGFA$_{165}$ (50 ng/ml) for 5 min. Rescue of VEGFR2 activation for indicated construct is shown (**g** representative picture, **h** quantification) ($n = 3$–6). Errors bars represent SEM. (N.S. not significant, $^*P < 0.05$, $^{**}P < 0.01$, $^{***}P < 0.001$, N.S. not significant, by one-way Anova with Bonferroni's multiple comparison test)

phosphorylation (pVEGFR2) (Fig. 3a, b). A similar reduction in VEGFR2 activation was observed in HUVEC after Sdc2 silencing (Supplementary Figure 4g–i). In addition, all major VEGFR2 downstream effectors (i.e., ERK, AKT, Src, and Integrin-β3)[18] showed reduced activation in Sdc2$^{-/-}$ ECs compared to WT (Supplementary Figure 5a, b).

The biological relevance of this finding was validated by analysis of VEGFR2 activation in whole-retina lysates at P7. In agreement with in vitro data, there was a significant reduction in VEGFR2 phosphorylation in Sdc2$^{iCdh5}$ compared to littermate control mice (Supplementary Figure 5c, d). Alteration of VEGFR2 signaling in retinal neurons can impair vascular

development in retina[37]. Thus, to confirm a direct defect in VEGFR2 signaling in the endothelium of Sdc2[iCdh5] mice, we further analyzed VEGFR2 target gene expression in freshly sorted retina ECs using qPCR. In particular, we focused on well-described target genes of the VEGFA/VEGFR2/PLCγ/IP3 pathway[38,39] that have shown strongest induction (≥10–600-fold) across multiple studies and functional validation in modulating VEGFA effects, including RCAN1[40,41], ANGPT2[42,43], EGR3, and NR4A2[44–46]. In agreement with a general reduction of VEGFA–VEGFR2 signaling activation, we observed that ECs from Sdc2[iCdh5] showed lower mRNA expression of all four genes compared to WT ECs (Supplementary Figure 5e). We did not detect any differences in VEGFR2 or NRP1 expression in freshly-sorted retina ECs from Sdc2[iCdh5] or control mice.

At the same time, activation of VEGFR2 signaling in Sdc4[−/−] ECs was not affected (Fig. 3c, e). The absence of increased Sdc2 expression in Sdc4[−/−] mice ECs rules out a compensatory response by Sdc2 (Fig. 3c and Supplementary Figure 7). In line with these findings, we have previously shown that Sdc4 knockdown in HUVEC does not affect VEGFR2 activation nor VEGFA-induced biological effects in vitro[47]. Finally, FGF2 signaling was normal in Sdc2[−/−] ECs, indicating that Sdc2 deletion did not lead to a generalized impairment in growth factor response (Fig. 3e, f).

We next set out to examine if Sdc2 GAG chains or its core protein sequence are involved in VEGFA$_{165}$ signal transduction. Transduction of Sdc2[−/−] ECs with an adenoviral wild-type Sdc2 construct (Ad-Sdc2) fully rescued reduced VEGFR2 phosphorylation in response to VEGFA$_{165}$. At the same time, transduction with either wild-type Sdc4 (Ad-Sdc4) or a Sdc2 mutant devoid of its GAG chains (Ad-Sdc2$^{ΔGAG}$) had no effect (Fig. 3g, h).

All constructs in these experiments carry an N-terminal HA- (Human influenza hemagglutinin) epitope tag for expression comparison and equal levels of expression were obtained in all studies. Similar results were observed after transduction, designed to achieve physiologic expression levels, of nontagged Sdc2 or Sdc4 constructs into Sdc2[−/−] ECs: rescue of VEGFR2 activation was again observed with reintroduction of Sdc2 but not Sdc4 (Supplementary Figure 5f, g)

Syndecan-2 GAG chains are primarily HS chains[9,48] that would be expected to bind VEGFA$_{165}$ and other heparin-binding growth factors. Once bound, VEGFA$_{165}$ can act as a bridge, linking Sdc2 HS chains to VEGFR2 thus leading to formation of a stable HS–VEGFA$_{165}$–VEGFR2 ternary complex. This promotes higher occupancy of VEGFR2 binding sites by VEGFA$_{165}$ compared to a VEGFR2–VEGFA$_{165}$ binary complex[49] leading to higher-VEGFR2 activity. To test this possibility, we carried out a Sdc2 pulldown in Sdc2[−/−] primary ECs transduced with HA-tagged Ad-Sdc2 or Ad-Sdc2$^{ΔGAG}$ constructs. While in the absence of VEGFA$_{165}$ Sdc2 had only minimal association with VEGFR2, the amount of the precipitated receptor increased several fold after VEGFA$_{165}$ treatment (Fig. 4a). At the same time, VEGFA$_{165}$ had no effect on the ability of Ad-Sdc2$^{ΔGAG}$ to form a complex with VEGFR2 (Fig. 4a). Both Ad-Sdc2 and Ad-Sdc2$^{ΔGAG}$ were equally efficient in complexing with a Sdc2 cytoplasmic partner syntenin (Fig. 4a).

To further test the role of Sdc2 HS in Sdc2–VEGFA$_{165}$–VEGFR2 complex formation, we used human umbilical vein ECs (HUVEC). As with mouse ECs, VEGFA$_{165}$ stimulation led to VEGFR2–Sdc2 complex formation (Fig. 4b). The adenoviral vectors used to express various syndecan constructs were used at low MOI (1–2) to achieve expression levels close to endogenous (Supplementary Figure 6a, b). Furthermore, changing the type of the tag employed (Flag vs. HA) or its relative position (C-terminal vs. N-terminal) did not alter the specificity of the Sdc2–VEGFR2 complex formation (Supplementary Figure 6c, d)

Treatment with Heparinases or K5 lyase to degrade cell surface HS chains led to a significant decrease in formation of the ternary Sdc2–VEGFA$_{165}$–VEGFR2 complex while Sdc2 ability to bind syntenin was unaffected (Fig. 4b). Finally, unlike VEGFA$_{165}$, VEGFA$_{121}$, an isoform that does not bind to HS chains, was unable to bring down VEGFR2 after Sdc2 immunoprecipitation (Fig. 4c). Taken together, these data strongly suggest Sdc2 HS chains play a key role in formation of the Sdc2–VEGFA$_{165}$–VEGFR2 complex.

To determine the specificity of Sdc2 HS chains for VEGFA$_{165}$, we treated Ad-Sdc2 transduced HUVEC with FGF2 and failed to observe any significant Sdc2/VEGFR2 co-precipitation (Fig. 4d). Furthermore, transduction of HUVEC with other syndecans showed that only Sdc2 had the ability to promote VEGFA$_{165}$/VEGFR2 complex formation following VEGFA$_{165}$ treatment (Fig. 4e).

**A Sdc2 N-terminal domain is necessary for specific association with VEGFR2.** Next, we generated Sdc2/Sdc4 chimeras by exchanging extracellular and intracellular sequences of the two syndecans. A Sdc2$^{EX}$/Sdc4$^{IN}$ construct (Sdc2 extracellular domain (ED) linked to Sdc4 transmembrane/intracellular domain) was as effective in promoting VEGFR2 immunoprecipitation in response to VEGFA$_{165}$ as Sdc2 itself (Fig. 4f). At the same time, a Sdc4$^{EX}$/Sdc2$^{IN}$ construct (Sdc4 ED linked to Sdc2 transmembrane/intracellular domain) was completely ineffective in this regard (Fig. 4f). To verify that Sdc2–VEGFR2 complex formation increases VEGFR2 signaling, we measured the extent of VEGFR2 phosphorylation in Sdc2[−/−] ECs transduced with Ad-GFP (control) or the two chimeras described above. Transduction with Sdc2$^{EX}$/Sdc4$^{IN}$ but not Sdc4$^{EX}$/Sdc2$^{IN}$ construct was able to restore the VEGFR2 phosphorylation (Fig. 4g, h). Taken together, these findings suggest that Sdc2 ED is unique in its ability to induce formation of VEGFA$_{165}$–VEGFR2 complex and promote VEGFR2 activation.

To better understand this ability of Sdc2 to form a complex with VEGFR2, we examined evolutionary conservation of its ED. Among mammalian orthologs (human, rat, and mouse), the ED alignment demonstrates the presence of two distinct regions (designated D1 and D2, Fig. 5a) based on number of mismatches with the consensus sequence. The N-terminal region (D1, aminoacids 1–59) shows a much higher degree of homology (88.1%) compared with perimembrane region (D2, aminoacids 60–144, 60.5%). Interestingly, compared to Sdc2, a D1 domain equivalent in Sdc4 (aminoacids 1–65) shows much less evolutionary conservation and is not as distinct from a D2 domain equivalent of Sdc2 (Fig. 5b–d). Importantly, the D1 domain is not conserved between the two syndecans (Fig. 5d).

Although both Sdc2 and Sdc4 D1 domains contain the repetitive *SGSG* glycosylation sites (GAG attachment sites usually surrounded by acidic aminoacids[50]), we reasoned that the additional homology in Sdc2 D1 region could be functionally important for the observed differences in Sdc2 vs Sdc4 ability to promote VEGFR2 signaling. To test this, we swapped D1 domains between the two syndecans leaving the rest of the molecule intact. The generated chimeras (Sdc2$^{D1}$/Sdc4$^{D2}$ and Sdc4$^{D1}$/Sdc2$^{D2}$, Fig. 5e) were cloned into adenoviral vectors and used to transduce Sdc2[−/−] primary ECs. When stimulated with VEGFA$_{165}$, Sdc2$^{D1}$/Sdc4$^{D2}$ chimera could form a complex VEGFR2 and did it to same extent as Sdc2 WT (Fig. 5e) in Sdc2[−/−] ECs. Taking this one step further, we generated Sdc2 and Sdc4 mutants that completely lacked D2 domains (Sdc2$^{D1}$ and Sdc4$^{D1}$, Fig. 5f). Transduction of

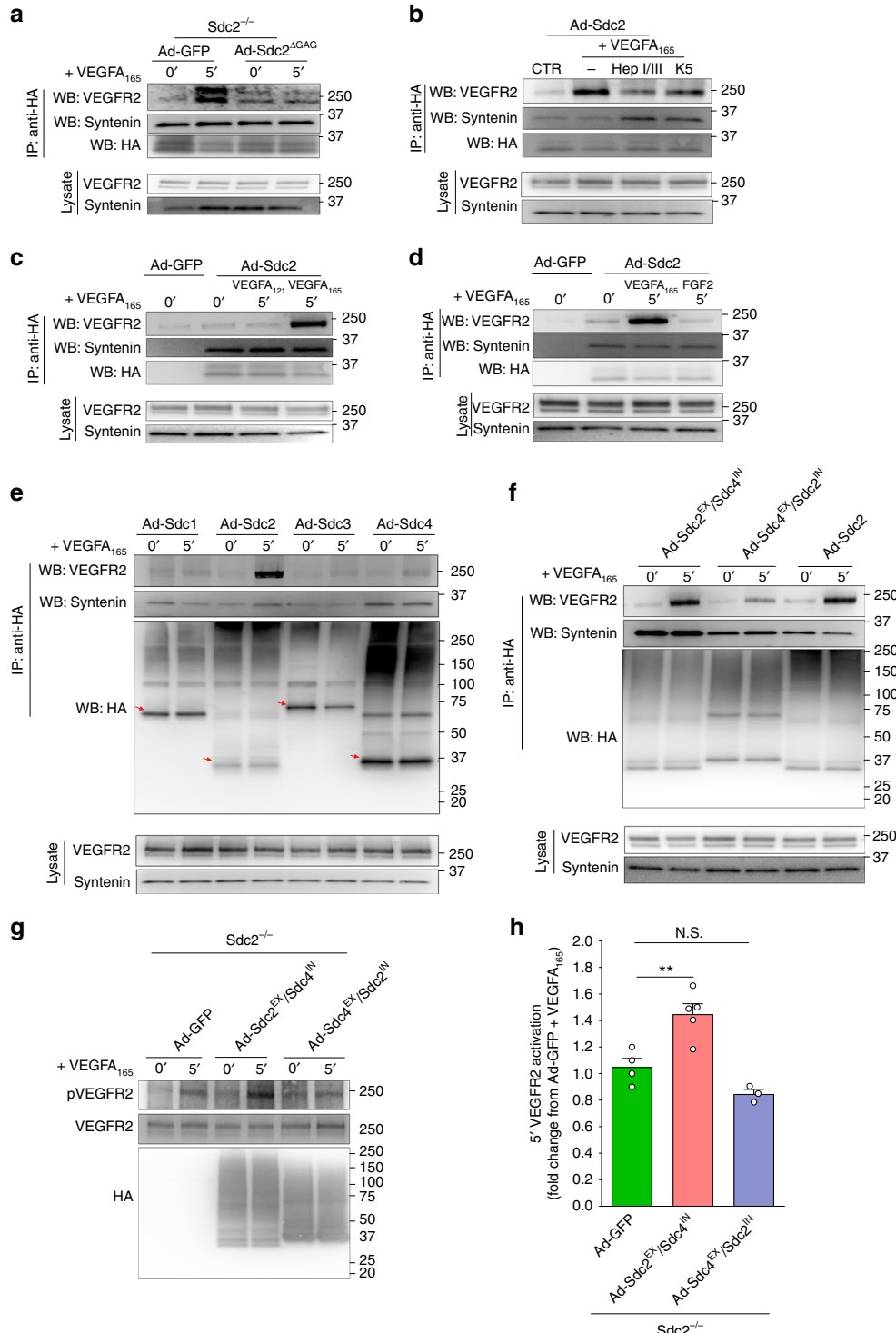

**Fig. 4** VEGFR2 specifically associates with Sdc2 upon VEGFA$_{165}$ stimulation. **a–f** Mouse ECs (**a**) or HUVEC (**b–f**) were transduced for 16 h with adenovirus expressing the indicated construct (MOI = 1–2), starved for 8 h and then stimulated with VEGFA$_{165}$ (50 ng/ml), VEGFA$_{121}$ (50 ng/ml), or FGF2 (20 ng/ml). Anti-HA pulled down (IP) was performed for 2 h at 4 °C followed by western blot analysis to check co-immunoprecipitated proteins (WB). Whole-cell lysates (lysate) were analyzed for total protein levels. **b** HS digestion with Heparinases or K5 lyase (1 h at 37 °C) before VEGFA$_{165}$ cell stimulation prevented formation of VEGFR2–Sdc2 complex. **c** VEGFA$_{121}$, which lacks heparin-binding domain, was unable to promote VEGFR2–Sdc2 association. **d** FGF2 did not promote complex formation between Sdc2 and VEGFR2. **e** Other syndecans displayed minimal or no association with VEGFR2 with or without VEGFA$_{165}$. Red arrows indicate syndecans core in dimeric form with following MW (calculated with signal peptide): Sdc1 ~65, Sdc2 ~44, Sdc3 ~91, Sdc4 ~44. **f** A chimera construct expressing Sdc2 extracellular domain with Sdc4 transmembrane + intracellular domain (Sdc2$^{EX}$ Sdc4$^{IN}$) showed same extent VEGFA-induced association with VEGFR2 as full length Sdc2. **g**, **h** Rescue of VEGFR2 activation is shown with chimera construct Sdc2$^{EX}$/Sdc4$^{IN}$ but not Sdc4$^{EX}$/Sdc2$^{IN}$ (**g** representative picture, **h** quantification) (n = 3–4). Errors bars represent SEM. (N.S. not significant, $^{**}P < 0.01$, by one-way Anova with Bonferroni's multiple comparison test)

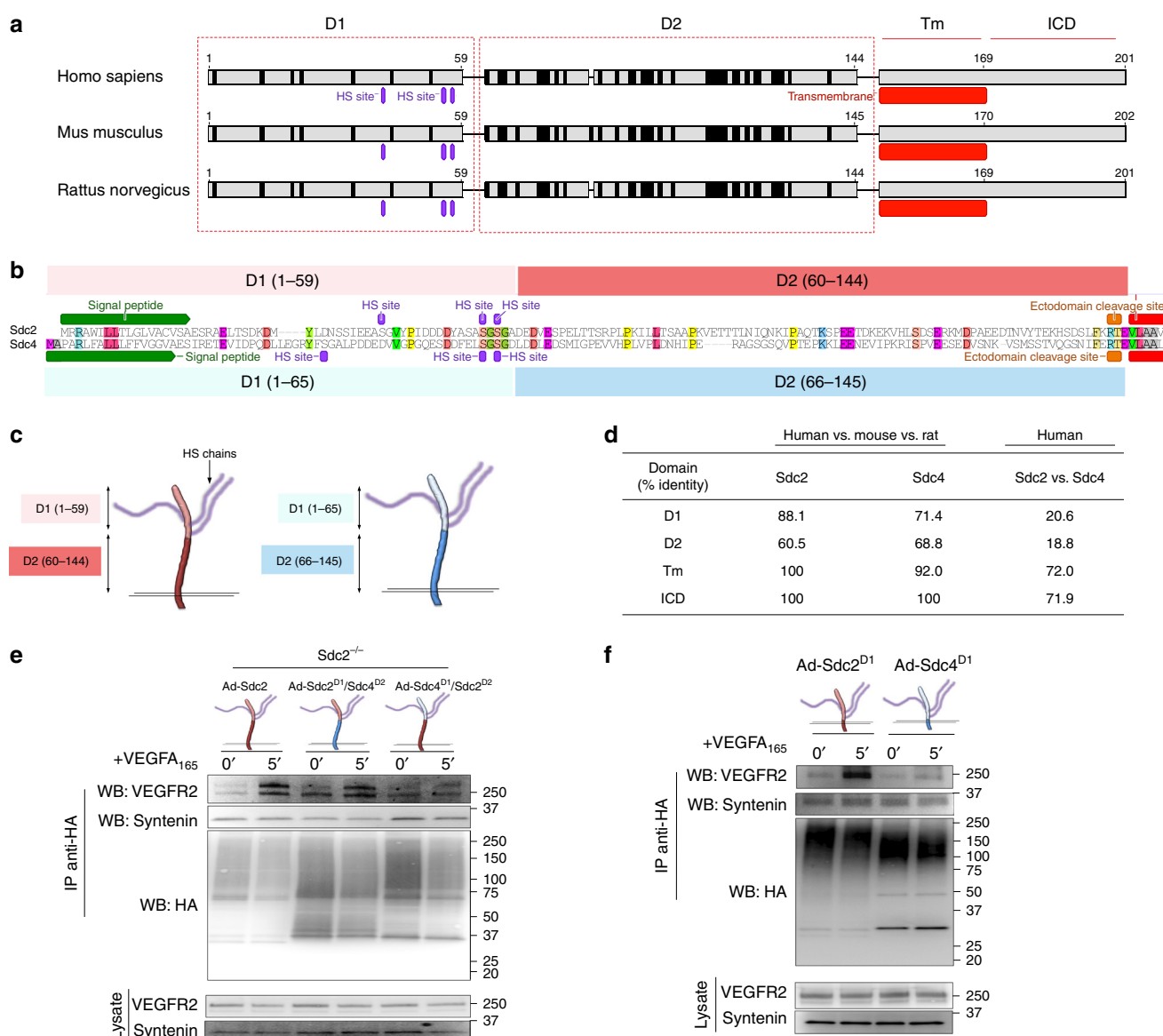

**Fig. 5** A Sdc2 conserved N-terminal region is required for specific association with VEGFR2. **a** Multi-alignment of Sdc2 sequences (human, mouse, and rat) unveiled that Sdc2 extracellular domain present a 59 aminoacid region with high-homology (D1) and a low-homology region (D2). Transmembrane domain (Tm) and intracellular domain (ICD) are also indicated. Black stripes identify sites that are not conserved among the three sequences. **b** Alignment of human Sdc2 with Sdc4 revealed little conservation in extracellular domain (colored aminoacid are conserved). Furthermore, Sdc4 does not present homology differences between D1 and D2 (see panel **d**); however, these regions were defined following alignment with Sdc2. **c** Schematic representation of D1/D2 regions in extracellular domain and relation with HS chains. **d** Percentual identity between various domain of Sdc2 and Sdc4. Identity is calculated by alignment of human sequence with mouse and rat (second and third column) or by alignment of human Sdc2 vs human Sdc4 (fourth column). **e**, **f** Mouse ECs (**e**) or HUVEC (**f**) were transduced for 16 h with adenovirus expressing the indicated construct (MOI = 1–2), starved for 8 h and then stimulated with VEGFA$_{165}$ (50 ng/ml). **e** A chimera construct swapping Sdc4 D1 region with Sdc2 D1 (Sdc2$^{D1}$/Sdc4$^{D2}$) showed association with VEGFR2 at the same extent of full length Sdc2. Conversely, replacement of Sdc2 D1 with Sdc4 D1 (Sdc2$^{D1}$/Sdc4$^{D2}$) abolished Sdc2 ability to form a complex with VEGFR2. **f** A mutant expressing only Sdc2 D1 region (Sdc2$^{D1}$) formed complex with VEGFR2 upon VEGFA$_{165}$ stimulation while Sdc4 D1 did not associate with VEGFR2

Ad-Sdc2$^{D1}$, but not Sdc4$^{D1}$, was sufficient to promote VEGFR2 complex formation (Fig. 5f).

**Core-dependent composition of HS chains.** Since the D1 domain of both syndecans contains all the GAG chains attached to the molecules, we next examined potential composition differences between Sdc2 and Sdc4 chains. To this end, isolated syndecan ED from HUVEC were treated with HepI-III and resulting disaccharides were analyzed using SAX-high

performance liquid chromatography (HPLC) and LC-mass spectrometry (MS). The level of total 6-O-sulfation and relative abundance of the D2S6 disaccharide have been described as major predictors of a HS chain's affinity for VEGFA$_{165}$, while 2-O-sulfation appears less critical[51,52]. In agreement with this observation, disaccharide analysis showed a 33.8% increase in total 6-O-sulfation in Sdc2 compared to Sdc4 HS chains (15.63 ± 1.55% vs. 11.68 ± 0.56%, $P < 0.01$ by unpaired $t$ test, Fig. 6a inset) while no statistically different changes were observed in N- and 2-O-sulfation frequency. Sdc2 D1 domain also appeared sufficient

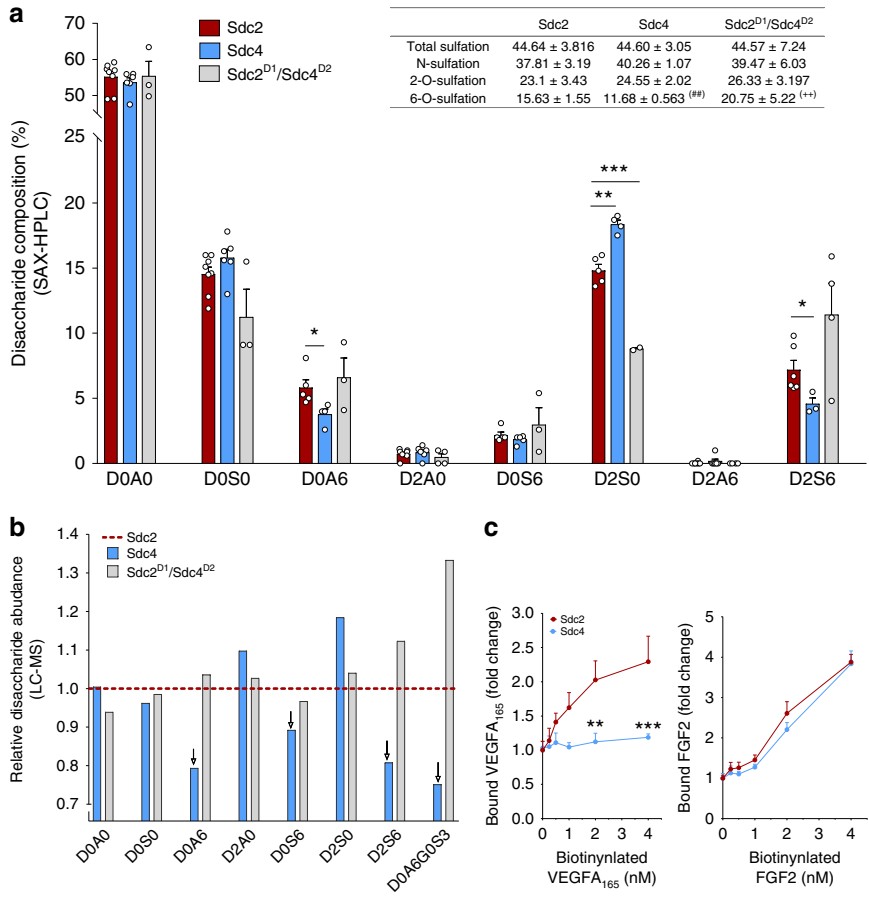

**Fig. 6** Composition of Sdc2-linked HS chains is N-terminal domain-dependent. **a**, **b** HUVEC were transduced with adenovirus expressing the indicated syndecan extracellular domain (ED). Secreted EDs were purified by ionic-exchange chromatography (DEAE) followed by affinity chromatography (anti-HA resin). Sdc-linked HS chains were digested with Heparinase I–III and analyzed by SAX-HPLC (**a**, $n = 4$–8, each dot is an independent HS chain isolation) and LC–MS (**b**, 3 independent isolations). **a** Figure inset, Sdc2 HS chains (Sdc2) showed higher frequency of 6-O sulfation compared to Sdc4 HS chains. A Sdc4 chimera ED expressing Sdc2 D1 region in place of Sdc4 D1 (Sdc2D1/Sdc4D2) showed increased 6-O sulfation that was comparable to that seen in Sdc2 (top right inset, ##$P < 0.01$ vs. Sdc2, ++$P < 0.01$ vs. Sdc4, ±precedes the standard deviation value). **b** Results of LC–MS analysis are presented after normalization to Sdc2 (shown as dotted line in red at $y = 1$). The analysis was repeated three times with similar results (a representative image is shown). Downward arrows highlight a reduction in 6-O sulfated disaccharides. **c** Secreted syndecan EDs were collected, digested with 24 h with pronase and GAG chains were concentrated/enriched with DEAE column chromatography. GAG-binding plate were coated with same amount of purified chains (1 μg) and tested for binding of biotinylated-VEGFA165 (left panel) or FGF2 (right panel). Errors bars represent SEM. Statistical analysis was performed by unpaired $t$ test (N.S. not significant, *$P < 0.05$, **$P < 0.01$, ***$P < 0.001$)

to promote the increase in 6-O sulfation as HS chains isolated from the Sdc2D1/Sdc4D2 chimera showed again a higher 6-O-sulfation compared to Sdc4 (77.7% increase, 20.75 ± 5.22% vs. 11.68 ± 0.56%, $P < 0.01$ by unpaired $t$ test, Fig. 6a inset). In particular, the abundance of D2S6 disaccharides was significantly higher in Sdc2 vs. Sdc4 chains as shown by HPLC (Fig. 6a). LC–MS analysis confirmed HPLC findings and showed that disaccharides carrying 6-O-sulfation (Fig. 6b) existed in higher frequency in Sdc2 vs. Sdc4 HS chains. 3-O-sulfation is a rare modification believed to occur after all other sulfation modifications[53]. LC–MS analysis showed a much higher abundance of the D0A6G0S3 tetrasaccharide carrying the 3-O-sulfation in Sdc2 compared to Sdc4 HS chains, in agreement with larger amounts of its precursor, D0A6G0S0, in Sdc2 HS chains. LC–MS also confirmed that HS composition in Sdc2D1/Sdc4D2 chimera was similar to Sdc2 and in agreement with HPLC findings (Fig. 6b).

To verify these observations and to demonstrate that they are not HUVEC-specific, we isolated Sdc2 and Sdc4 EDs from a mouse endothelial cell line (MS1) and performed disaccharide analysis as described above. Similar to the observed differences in

HUVEC, we found that Sdc2 HS chains derived from MS-1 also showed higher 6-O sulfation compared to Sdc4 HS chains (22.90 ± 1.91 vs. 15.5 ± 2.545, $P < 0.05$ by unpaired $t$ test, Supplementary Figure 6e). In line with these results, co-immunoprecipitation experiments using this cell line showed that VEGFA165 was again able to induce the association between VEGFR2 and Sdc2 but not with Sdc4 (Supplementary Figure 6f).

Finally, we tested if these differences in composition are reflected in the ability of Sdc2 and Sdc4 HS chains to bind VEGFA165 and FGF2. To this end, we measured binding of the two growth factors to DEAE-purified HS chains from pronase-treated Sdc2 and Sdc4 EDs in a plate assay. As expected, there was a significant increase VEGFA165 binding to Sdc2 but not Sdc4 HS chains while FGF2 bound to both sets of HS chain with equal affinity (Fig. 6c).

## Discussion
The results of this study show that Sdc2 engages in formation of Sdc2–VEGFA165–VEGFR2 trimolecular complex which leads to

enhanced VEGFA signaling while Sdc4 does not possess such an ability. Surprisingly, the main reason for this difference is increased specific binding of $VEGFA_{165}$ to Sdc2,HS chains. Analysis of syndecan te HS chains showed higher 6-O sulfation in Sdc2, compared to Sdc4. The extent of 6-O-sulfation was previously correlated with the increased HS chain's affinity for $VEGFA_{165}$[51,52]. In agreement with these data, Sdc2 (but not Sdc4) knockout mice display reduction of VEGFA-driven angiogenic and arteriogenic processes both during development and in adult animals.

The phenotype that we describe in retinal vascular development of mice carrying endothelial-specific Sdc2 deletion, resembles one previously observed in heterozygous VEGFR2 mice, where a single allele deletion in ECs (Pdgfb-iCre) led to a ~15% reduction in the vascular front extension and ~30% reduction in number of branch points[54] suggesting similar reductions of VEGFR2 in vivo signaling between the two strains. We speculate that in vivo, Sdc2 functions as a VEGFR2 sensitizer acting to boost occupation VEGFR2 sites by $VEGFA_{165}$. Modulation of Sdc2 expression in various vascular beds under different physiologic or pathologic conditions may provide the means to modulate signaling of heparin-binding vs. nonheparin-binding VEGFA isoforms. Although we observed a direct role of Sdc2 in regulating VEGFR2 signaling in retinal ECs, it is possible that shed Sdc2 could affect neuronal signaling in a paracrine fashion. Since neuronal VEGFR2 signaling can affect retinal vascular development[37], this effect may contribute to the vascular phenotype of Sdc2iCdh5 mice.

At the cellular level, a reduction in VEGFR2 activation in Sdc2$^{-/-}$ ECs leads to significantly reduced VEGFA-driven cell proliferation and migration. This is important, as to date little attention has been paid to Sdc2/VEGFA interactions. Impaired vascular development was reported in zebrafish following Sdc2 morpholino oligonucleotides knockdown[25] while the addition of glycosylated Sdc2 ED or conditioned medium enriched for shed Sdc2 to aortic explants was shown to inhibit VEGFA-induced sprouting[55].

A number of other studies reported variable results in in vitro when endothelial effects of Sdc2 knockdown were studied in the context of serum stimulation. In particular, while one study reported decreased serum-induced ECs migration and cord formation in the Matrigel assay following Sdc2 knockdown[56], another reported also reduced cord formation in the Matrigel assay but increased single-cell motility in response to 10% fetal bovine serum (FBS)[57]. Finally, no significant differences were detected in a scratch migration assay in response to 10% FBS[57]. A recombinant Sdc2 ED lacking glycosylation increased ECs migration in response to 10% FBS and cord formation in a Matrigel assay[56]. These results are difficult to interpret given HS chains ability to bind multiple heparin-binding growth factors and the variable growth factor composition of different FBS sources. Interestingly, Sdc2 has been shown to regulate migration and adhesion via integrin interaction[58,59] and, since VEGFR2 is known to activate integrins[60–62], it is possible that formation of VEGFA-induced Sdc2–VEGFR2 complex may represent a molecular link between these pathways.

A recent surface plasmon resonance study showed that the presence of heparin strongly enhances occupation of VEGFR2 binding sites by $VEGFA_{165}$[49]. This is in agreement with our data showing that Sdc2 HS chains, enriched with $VEGFA_{165}$-binding sites, increase VEGFR2 activation. Additionally, the same report showed that VEGFR2 can interact with heparin only in the presence of $VEGFA_{165}$. This is also consistent with our data demonstrating that a stable association between Sdc2 and VEGFR2 is not observed when $VEGFA_{165}$ is absent. In this model, $VEGFA_{165}$ appears to behave as a molecular bridge

between Sdc2 and VEGFR2 and promoting formation of a stable trimolecular complex. The exact kinetic and structural parameters of this model remained to be elucidated and may provide novel insights into how HS chains can regulate growth factor signaling.

Importantly, our data show that the increased extent of 6-O sulfation of Sdc2 HS chains requires the presence of an N-terminal 59 aminoacid sequence in the Sdc2 core protein. Indeed, a swap of N-terminal subdomains containing this fragment between Sdc2 and Sdc4 increased 6-O sulfation of the Sdc2$^{D1}$/Sdc4$^{D2}$ chimera to the level seen in Sdc2 demonstrating that this aminoacid segment controls 6-O sulfation of HS chains.

How this N-terminal Sdc2 domain regulates 6-O sulfation levels of HS chains remains to be determined. One possibility is that some, but not other core protein sequences are likely to facilitate the exposure of a growing GAG chains to a particular sulfotransferase by increasing the time of contact. Another is that certain N-terminal core protein sequences can address the newly synthesized core protein to Golgi compartments with a higher concentration of a specific sulfotransferase.

## Methods

**List of antibodies and growth factors**. List of antibodies with application and dilutions (immunohistochemistry (IHC); western blot (WB)): pAKT T308 (WB 1:1000, Cell Signaling #4056), Total AKT (WB 1:1000, Cell Signaling # 4691), ERG (IHC 1:100, Abcam #ab92513), pERK (WB 1:1000, Cell Signaling #9106), total ERK (1:2000, Cell Signaling #9102, WB), FGFR1 (Cell signaling #9740, WB 1:1000), Flag-tag (WB 1:500, Sigma-Aldrich F1804), HA-tag (WB 1:1000, Cell signaling #3724), NRP1 (Cell signaling #3725, WB, 1:1000), pIntegrin (Itg)-β3 Y759 (WB 1:200, Santa Cruz #sc136458), total Integrin-β3 (WB 1:1000, Cell signaling #13166) PECAM1 (IHC 1:200, BD Pharmingen #553370), Sdc2-mouse (IHC 1:200, LSBIO #LS-B2981), Sdc2-mouse (WB 1:200, R&D #AF6585—polyclonal raised against mouse Sdc2 ED), Sdc2-human (WB 1:200, R&D # AF2965, polyclonal raised against mouse human ED), Sdc4 (WB 1:500, Abcam #ab24511—monoclonal against C-terminal domain), pSRC Y416 (WB 1:1000, Cell Signaling #6943), Total Src (WB 1:1000, Cell Signaling #2109), pVEGFR2 Y1175 (WB 1:1000, Cell Signaling #2478), total VEGFR2 (WB 1:1000, Cell Signaling #2479), VE-Cad (WB 1:200, Santa Cruz #sc-6458 C-19 clone), GFP (Santa Cruz #sc-9996, WB, 1:200), and Syntenin (WB 1:500, Abcam #ab19903). List Growth factors: $VEGFA_{165}$ (R&D #293-VE-010), $VEGFA_{121}$ (R&D #4644-VS-010), FGF2 (R&D #233-FB-025).

**Detection of syndecan expression by western blot**. Sdc2 detection in mouse ECs and HUVEC: heparinase pre-treatment is required to unveil epitopes on core protein and remove nonspecific sugar epitopes (Supplementary Figure 7 for uncut WB images). Briefly, confluent ECs on 6 cm tissue culture dishes were rinsed twice with DPBS that was then replaced with 2 ml of serum-free media (Opti-MEM, ThermoFisher) containing 0.5 U/ml of Heparinase I-III (Sigma-Aldrich, #H2519 and #H8891) and 0.2 U/ml of Heparinase II (Sigma-Aldrich, #H6512). After 2 h digestion at 37 °C, cells were washed three times with ice-cold PBS, lysed in RIPA buffer and samples prepared for western blot analysis. Specific band at ~42 kDa is the naked core protein in a dimeric form. Additional specific smears (50–250 kDa) are Sdc2 glycosylation isoforms. Sdc4 detection in mouse ECs and HUVEC: heparinase pretreatment is not necessary. A specific band below 20 kDa is detected corresponding to the naked core in a monomeric form. HA-Tag syndecans: heparinase pretreatment is not required. HA detection reveals naked core proteins in a dimeric form and additional higher molecular weight smear bands due to glycosylation (Fig. 4e for comparison of all syndecans).

**Generation of transgenic mice**. Mouse ES cells carrying a KO-first Sdc2 transgenic allele (Supplementary Fig. 1a) were obtained from KOMP repository (Strain ID: Sdc2$^{tm1a(KOMP)Wtsi}$), injected into a blastocyst and implanted into a pseudo-pregnant mouse. Mice carrying the transgenic allele were established and back-crossed to a pure C57Black6/J for at least 10 generations. These were crossed to a FLPe recombinase to remove the FRT sites and generate a conditional allele with two loxP sites flanking exon 3 (Sdc2$^{flox/flox}$ mice). These mice were finally crossed to various CRE-recombinases to either obtain global null mice (Sdc2$^{-/-}$) or tamoxifen-inducible endothelial deletion (Sdc2$^{iPdfb}$ and Sdc2$^{iCdh5}$). Sdc4 global null mice (Sdc4$^{-/-}$) were previously studied in our laboratory[13]. CMV-Cre exon three deletion was confirmed in tail-isolated DNA (Transnetyx, INC., Cordova, TN). Endothelial-Cre driven deletion was confirmed by qPCR analysis of primary lung ECs as described in detail below. All mouse experimental protocols have been approved by the Institutional Animal Care & Use Committee (IACUC) at Yale University. The authors have complied with all relevant animal testing and research ethical regulations.

**RNA isolation and qPCR**. Cells were washed twice with PBS and homogenized with QIAshredder Kit (Qiagen). Total RNA was extracted with RNeasy Plus Mini Kit (Qiagen) which include a genomic DNA elimination step. cDNA synthesis was performed with iScript cDNA syntesys kit (Biorad). Quantitative real-time PCR (qPCR) was performed in triplicate using iQ SYBR Green Supermix kit and CFX96™ Real-Time System (Biorad). Thermocycling conditions were: 95 °C for 3 min, followed by 45 cycles at 95 °C for 10 s, 60 °C for 30 s. Gene expression was normalized (GAPDH or VE-Cad) and relative expression was calculated using the ΔΔCt method. A complete primer list is reported in Supplementary Table 1.

**Analysis of mouse retinal vascular developments**. Eyes were removed from neonates at postnatal day 5 (P6) and prefixed in 4% paraformaldehyde (4% PFA) for 15 min at room temperature. Dissected retinas were blocked overnight at 4 °C in TNBT (0.1 M Tris-HCl, 150 mM NaCl, 0.2% blocking reagent (PerkinElmer) supplemented with 0.5% TritonX-100). After washing, the retinas were incubated with Isolectin GS-IB₄, Alexa Fluor® 488 Conjugate (ThermoFisher #I21411) in Pblec (1 mM MgCl2, 1 mM CaCl2, 0.1 mM MnCl2, 1% Triton X-100 in PBS) for 2 h at RT. Retinas were washed 6 times, for 10 min in PBS, fixed briefly for 5 min in 4% PFA, washed twice in PBS and mounted in fluorescent mounting medium (DAKO mounting media #CS703). Low- and high-magnification images were acquired using fluorescent (Nikon 80i Nikon Ti-E Eclipse inverted microscope) and confocal (ZEISS LSM710 laser scanning confocal) microscopes. ImageJ was used to measure distance from center to vascular edge (VE) and center to retina edge (RE). Vascular progression is reported as VE/RE ratio. Biological CMM Analyzer software 16 was used for quantification of branch points per image[63].

**Assessment of EC density and proliferation in retina**. P5 mice were injected with a single pulse of 50 μg EdU (Sigma #900584) for 4 h before retina dissection. Retina was stained with Erg to visualize EC nuclei followed by detection of EdU incorporation using Click-iT Edu Imaging Kit (Invitrogen, #C10340). The retinal vasculature was labeled with Isolectin B4 (Invitrogen, #I121411). ImageJ (NIH) was used for quantification of the following parameters in each image: total number of ECs (Erg⁺ cells), total number of proliferating ECs (double positive Erg⁺ Edu⁺) and vascularized area. Cell density corresponds to number of ECs (Erg⁺) per millimeter square (mm2) of vascular area. Fraction of proliferating cells correspond to double positive ECs (Erg⁺ Edu⁺) over total ECs (Erg⁺).

**Cornea pocket assay**. Slow-releasing pellets containing VEGFA₁₆₅ or FGF2 were surgically implanted into the mouse cornea[64]. One week after pellet implantation, eyeballs were collected, corneas dissected and immunostained with PECAM1 (BD Pharmigen #553370) to quantify neovessels formation with ImageJ (NIH).

**Hind limb ischemia**. This was done as previously described by our lab[65]. Briefly, surgical procedures were performed in mice under anesthesia and sterile conditions. A vertical longitudinal incision was made in the right hind limb (10 mm long). The right CFA and its side branches were dissected and ligated with 6-0 silk sutures spaced 5 mm apart, and the arterial segment between the ligatures was excised. Assessment of blood perfusion by Laser-Doppler flow-Imaging (LDI) was done by scanning both rear paws with a LDI analyzer (Moor Infrared Laser Doppler Imager Instrument, Wilmington, Delaware) before and after the surgical procedure (days 0, 3, 7, and 14). The animals were kept under 1% isoflurane anesthesia and body temperature was maintained between 36.5 and 37.5 °C. Low or no perfusion is displayed as dark blue, whereas the highest degree of perfusion is displayed as red. The images obtained were quantitatively converted into histograms with Moor LDI processing software V3.09. Data were reported as the ratio of flow in the right/left (R/L) hind limb and calf regions (not shown). Measurement of blood flow was done before and after the surgical procedure (days 0, 3, 7, and 14).

**Micro-CT angiography**. For microcomputed tomography (mCT) renal and hind limb vasculature, euthanized mice were injected with 0.7 ml solution (bismuth contrast solution) in the descending aorta. The mice were immediately chilled in ice and immersion fixed in 2% paraformaldehyde overnight. The vasculature was imaged and quantified as described previously[65] and in detail as follows: 2D mCT scans were acquired with a GE eXplore Micro-CT System (GE Healthcare), using a 400 cone beam filtered back projection algorithm, set to an 8–27 μm micron slice thickness. Micro-CT quantification was done as previously described. In brief, data were acquired in an axial mode, covering a volume of 2.0 cm in the z direction with a 1.04 cm field of view. During postprocessing, a 40,000 gray scale value was set as a threshold to eliminate noise (air, water, and bone signals) with minimal sacrifice of vessel visualization. The mCT data were processed using real-time 3D volume rendering software (version 3.1, Vital Images, Inc. Plymouth, MN) and microview (version 1.15, GE medical system) software to reconstruct three 2D maximum intensity projection images (x, y,and z axes) from raw data. Quantification was performed using a modified Image ProPlus 5.0 algorithm (Media Cybernetics). The data are expressed as vessel number, representing total number of vessels, of specified diameter counted in 200 z sections from thigh and kidney or in 350–400 z sections from heart images. For analysis of heart and kidney, and hind limb vasculature, 4–5 mutant mice and 4 gender and age matched controls were used.

**Cell culture and mouse ECs isolation**. Human umbilical vein endothelial cells (HUVEC) were obtained from Yale VBT tissue-culture core laboratory at Passage 1 and maintained in complete EGM-2 MV medium (LONZA). HUVEC were used for experiments between P2 and P6. Primary mouse ECs were isolated as previously described[47]. Briefly, 4 hearts or lungs were harvested, finely minced with scissors and digested (37 °C for 45 min under gentle agitation) in 25 ml of 1.4 mg/ml Collagenase/Dispase® solution for Heart (Sigma-Aldrich #10269638001) or 1.5 mg/ml Collagenase Type I (Sigma-Aldrich #C0130). The crude preparation was triturated passing it 10 times through a cannula needle, filtered on a 70-μM sterile cell strainer, and spun at 400g for 10 min. Pellet was resuspended in 2 ml of 0.1% bovine serum albumin (BSA) and 50 μl magnetic dynabeads (ThermoFisher #11035) precoated overnight with anti-mouse CD31 (BD Pharmingen™ #553370) were added for ECs-positive selection. Selection was carried out for 20 min at room temperature under slow rotation. The bead-bound cells were recovered with a magnetic separator and washed five times with DMEM containing 10% FBS. Cells were finally resuspended in 10 ml of complete DMEM medium (20% FBS with ECGS and antibiotics) and seeded onto gelatin-precoated 10 cm plates. HEK 293A cells were maintained in DMEM containing 10% FBS and penicillin/streptomycin. MS1 cell were obtained from ATCC (ATCC® CRL-2279™) and maintained in 5% FBS with penicillin/streptomycin.

**Cloning and adenovirus production**. Adenoviruses expressing various syndecan sequences were generated as previously reported[13,47]. Briefly, presynthetized blunt-end sequences corresponding to wild-type or mutant syndecans (IDT, Coralville, IOWA) were subcloned into a pENTR/D-TOPO (Invitrogen) vector, and then transferred via LR recombination into a pAD/CMV/V5-DEST adenoviral vector (Invitrogen). Adenoviruses were generated by transfection of this plasmid into HEK 293A (Invitrogen). A full list of constructs used in this study can be found in Supplementary Table 2.

**Growth factor stimulation and western blot analysis**. HUVEC or mouse ECs were seeded onto 6 cm plates in a complete medium. Confluent cells were starved (2% FBS, no growth factors added) for 12 h and then stimulated for 5 min with the indicated agent. Rescue experiments were carried out by infecting ECs with an adenovirus (MOI ~1–2) for 16 h followed by starvation for 12 h. Stimulated cells were rapidly washed twice with ice-cold PBS and lysed with 200 μL lysis buffer (1% Triton X100, 150 mM NaCl, 50 mM Tris-HCl, 5 mM EDTA) containing protease/phosphatase inhibitor cocktail. Total lysates were cleared with a 16,000g spin and protein concentration was determined using the bicinchoninic acid assay method. Samples were added with reducing loading buffer, boiled for 5 min and loaded on 4–15% gels for sodium dodecyl sulphate polyacrylamide gel electrophoresis separation. Proteins were then transferred to polyvinylidene fluoride Immobilon-P membranes (Millipore), blocked for 1 h in 5% fat-dry milk TBS-T (0.05% Tween) followed by 4 °C overnight incubation the primary antibody. Protein bands were visualized using horseradish peroxidase (HRP)-conjugated secondary antibodies associated to enhanced chemiluminescence (Immobilon™ Western, Millipore). Signal from chemiluminescence reaction was recorded in a digital acquisition system (G-Box by Syngene) equipped with CCD camera. Linear range is automatically calculated by the software and is displayed as a histogram with each acquired image. Images without band saturation were used for densitometric quantification. Total intensity of each band was determined with ImageJ software[66]. Molecular weight on western blot images is reported in kilodaltons (kDa). All uncropped western blot images are reported in Supplementary Figures 8–12.

**Sdc2 silencing in HUVEC**. HUVEC were seeded onto 6-well plates and transfected at 75% confluency with 2.5 ml Opti-MEM (Thermo-Fisher) containing 40 nM of Sdc2 siRNA (OriGene #SR321721—C) and 2.5 μL Lipofectamine RNAiMAX (ThermoFisher) for 6 h. Transfection mix was replaced with full media (EGM-2 MV) for 60 h, then cell starved in 2% FBS for 12 h before growth factor stimulation.

**Analysis of VEGFR2 phosphorylation in whole retina**. VEGFR2 phosphorylation in whole retina was assed as previously reported[67]. Briefly, P7 mice were euthanized, eyes removed and retina quickly dissected in ice-cold PBS. Whole retinas were homogenized in 80 μl RIPA buffer with a Tissue Lyser (Qiagen). Samples were cleared at 16,000g for 10 min, added with 1× loading buffer and boiled for 5 min. WB analysis was carried out as described above.

**Analysis of VEGFA–VEGFR2 downstream gene expression in retinal ECs**. Expression of VEGFA downstream genes was performed by qPCR analysis of freshly-sorted retinal ECs. Primers of analyzed genes (RCAN, ANGPT2, EGR3, NR4A2, VEGFR2, and NRP1) are reported on Supplementary Table 1. Sorting of retina ECs was performed as previously reported[68]. In brief, retinas from same mouse (P7) were pooled together and digested in 1.5 ml with a collagenase type I (I (Worthington #LS004196, 1 mg/ml) for 40 min. Digested suspension was triturated 4–5 times with a 1 ml tips, filtered through a 70 μm strainer and spun down at 400g for 10 min. The resulting pellet was resuspended in 1.5 ml of a 0.1% BSA/Dulbecco's phosphate-buffered saline (DPBS) solution and allow to bind CD31-precoated magnetic beads (30 μl) for 20 min at room temperature. Beads were washed five times with DPBS then lysis buffer added and RNA quickly isolated.

**Primary endothelial cell proliferation and migration**. Proliferation was performed using xCELLigence RTCA System (Acea Biosciences). Briefly, 2000 primary mouse ECs per well were seeded in gelatin-coated E16 plates in 2% FBS, allowed to adhere for 7 h and then stimulated with growth factors (VEGFA165 200 ng/ml or FGF2 100 ng/ml) or vehicle (PBS). Proliferation curves show ~60 h of monitoring with readings of cell index performed every 15 min. Cell index values at 24 h were used to calculate fold changes in proliferation of indicated growth factor versus vehicle. ECs migration was assessed using an in vitro wound healing assay as previously reported[47]. Briefly, cells were seeded on Ibidi-culture inserts (Ibidi, #80,209) to create a wound between two adjacent EC monolayers. At confluency, insets were removed, and cells allowed to migrate. Pictures of wound width were taken before and after stimulation (8 h) and % closure was calculated.

**Co-immunoprecipitation**. Confluent HUVEC or mouse ECs were transduced for 16 h in complete media (EGM-2 MV) using adenovirus (MOI ~1–2) expressing the indicated syndecan constructs. Except where indicated otherwise, all constructs carried an N-terminal HA (human influenza hemagglutinin) epitope tag after the signal peptide sequence. Following transduction cells were starved for 8 h and then stimulated with the indicated agent for 5 min. Cells were quickly washed twice with ice-cold PBS, lysed in 1.6 ml in 1% Triton lysis buffer and spun at 16,000$g$ for 10 min. Totally, 600 μg (~1.4 ml) of cleared lysate was immunoprecipitated with 30 μl/ sample of anti-HA magnetic beads (ThermoFisher #88836;) for 2 h at 4 °C under gentle rotation. Beads were washed 3 times with 1.5 ml lysis buffer, resuspended in 80 μl of 1× loading buffer and boiled for 5 min. Samples were analyzed by western blot as described above.

**Purification of Sdc2 and Sdc4 EDs**. Secreted syndecan EDs were purified from HUVEC as in previously described protocols[69,70]. Briefly, 10–12 tissue culture dishes (10 cm diameter) of confluent HUVEC were transduced with adenovirus expressing the indicated syndecan ED (MOI = 5–10) in serum-free, growth factor-added media (EGM-2 MV). Media plus one PBS wash were collected at 36 and 72 h post-transduction. Each fraction was kept frozen until purification. Syndecan-rich media (~300 ml) was filtered on 0.45 μm filter unit (NALGENE) and then pass through a 1 ml DEAE column (GE Healthcare HiTrap™ DEAE FF) using a peristaltic pump (flow ~1.5 ml/min). The column was then washed with 10 column volumes of PBS, 10 column volumes PBS (0.25 NaCl), followed by elution with 20 column volumes of phosphate-buffered 2 M NaCl. Samples were buffer exchanged to 150 mM NaCl, concentrated to ~6 ml and adjusted to pH ~7.2. To this solution 250 μL of anti-HA conjugate agarose resin (ThermoFisher #26181) was added and incubated for 16–18 h at 4 °C under gentle rotation. The resin was then washed 4 times with PBS and batch-eluted 4 times with 500 μl of a 3 N NaSCN solution. Buffer was exchanged to low PBS (20 mM NaCl, 10 mM phosphate buffer, pH = 7.4) and concentrated to ~150 μl. Purified EDs were used for HS compositional analysis.

**HS chain compositional analysis (SAX-HPLC and LC–MS)**. Syndecan EDs were digested with 1 μL Heparinases I–II–III (New England Biolabs) to break down any potential Heparin/HS into disaccharides. The enzymatic products were then separated with SAX-HPLC (4.6 × 250 mm Waters Spherisorb analytical column with 5 μm particle at 1.0 ml/min flow rate) coupled to fluorescence detection via postcolumn derivatization. The separations of sample disaccharides were compared to the separation of standard disaccharides (Dextra Laboratories) for identification and quantification. LC–MS analysis was performed on a Dionex Ultimate 3000 LC system interface with Thermo Scientific Orbitrap Elite; Separation was carried out on a 2.1 × 150 mm Waters Acquity UPLC BEH C18 column with 1.7 μm particle size at 30 °C. The analytes were monitored by Orbitrap Elite with spray voltage of 3.6 kv and capillary temperature of 275 °C. Mobile phase (A): [20 mM tributylamine acetate in water, pH 4.1], mobile phase (B): [20 mM tributylamine acetate in 80% acetonitrile].

**HS chains quantification (carbazole micro-assay)**. HS amount in purified EDs or isolated HS chains was quantified as previously described in detail[71]. Briefly, 150 μl/well of a 25 mM sodium tetraborate (Sigma-Aldrich #S9640) solution in 98% sulfuric acid were added to a 96-microplate kept on an ice-bed. To this, 40 μl/well of 1:10 H$_2$O-diluted samples of isolated syndecan EDs were added. The plate was transferred to a 100 °C oven for 15 min. After incubation, the plate was returned to the ice-bed, allowed to cool down and then 4 μl/well of a 0.125% w/v carbazole (Sigma Aldrich # 442506) solution in ethanol were added. The plate was then incubated at 100 °C for 10 additional minutes followed by absorbance measurement at 525 nm. Heparin (Sigma Aldrich # H3393) was used to build a standard curve for quantification.

**VEGFA-GAG binding**. Secreted syndecan EDs were collected from HUVEC as above and digested with pronase (0.5 mg/ml) for 18 h at 37 °C under gentle agitation. Afterwards, the solution was passed through a DEAE column to eliminate any remained protein fragment and obtain a concentrated GAG solution (PBS as final buffer). Binding of VEGFA165 to isolated chains was quantified using a GAG binding 96-well plate (Galen # H/G plates) following manufacturer's instructions and as previously reported[72]. Briefly, wells were coated overnight at room

temperature with 1 μg/well of total chains, blocked with 1% BSA (1 h at 37 °C) and then incubated for an additional 1 h with increasing concentration of biotinylated-VEGFA165 or FGF2 (AcroBiosystems #VE5-H8210 and #BFF-H8117). Plate wells were washed three times and bound factor was detected with HRP-conjugated Neutravidin followed by colorimetric detection.

**Reporting Summary**. Further information on experimental design is available in the Nature Research Reporting Summary linked to this article.

## Data availability

All data presented in this manuscript are available from corresponding author upon reasonable request. The source data underlying Fig. 2e are provided as a Source Data file.

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

## Acknowledgements

We thank Mary Edgington for helping with the initial cloning of syndecan constructs and Georgia Zarkada for technical help in retinas analysis. This research was supported in part by the "National Institutes of Health (NIH)-funded Research Resource for Integrated Glycotechnology" (NIH grant No. 5P41GM10339024) to Parastoo Azadi at the Complex Carbohydrate Research Center and R01 HL062289 (M.S.).

## Author contributions

F.C. designed and performed the experiments, analyzed the data and wrote the paper. Y. W. designed/performed the experiments and analyzed the data. J.M.R. designed the experiments and generated the initial Sdc2 transgenic mice. D.A. designed and performed the experiments, S.A.-H., J.Z., Z.W.Z, D.C., T.W, and Z.W. performed the experiments, P.A. analyzed the data. M.S. designed the experiments, analyzed the data and wrote the paper.

## Additional information

**Competing interests:** The authors declare no competing interests.

