## [Peer Review File · Nature Communications]

Reviewers' comments:

Reviewer #1 - expert in syndecans (Remarks to the Author):

Heparan sulfate (HS) glycosaminoglycan chains have very important roles in development and disease, and a numerous studies indicate that these roles are dictated by the sulfation pattern within the HS chain. The question of whether or not the sulfation pattern is distinctive of the core protein on which they are synthesized, which implies the presence of regulatory determinants in the protein itself, rather than distinctive of a given cell type and the enzymes that it expresses in the Golgi apparatus, has been debated now for several decades. But the studies investigating the presence of potential regulatory elements in the core protein have been few and mixed, leaving it still a very open-ended question after years of progress on understanding the biology of the HS chains themselves.

The Simons group has now shown that a determinant (although a rather large protein of the core protein extracellular domain) in syndecan-2 (Sdc2) appears to dictate the composition of the HS chain in endothelial cells. Furthermore, the sulfate composition of the chains favors the binding of VEGF-A165, one of the several VEGF angiogenic factors that is known to bind HS. What this determinant actually does is not addressed, which is a little disappointing, but the authors have established an assay system that should allow them to get at this. If true, this is the first meaningful demonstration of such a determinant, and I suspect it will lead to the discovery of others. The work is very important.

Nonetheless, there are some nagging concerns with the work. One concern is an inadequate description of the angiogenic response of the Sdc2^{-/-} "global" knockout mouse. The authors indicate that they saw no profound pathological effect in the global knockout, but then go on to describe a defect using an endothelial cell-specific inducible knockout activated after birth of the pups. Another concern is the use of epitope-tagged syndecans that are likely overexpressed when introduced in adenoviral vectors. Either condition (epitope tag or overexpression) can interfere with heparan sulfate synthesis on the core protein. These pesky concerns detract from the excellence of the finding, especially as it is being considered for publication in a top journal. Despite these concerns, however, the authors arrive at a consistent conclusion by the end of the manuscript, and can demonstrate that Sdc2 and Sdc4 isolated from human umbilical vein endothelial cells do indeed appear to bear distinct HS that differs in its capacity to bind VEGF-A165.

Specific comments:

1. The authors generate an ES cell "global" Sdc2^{-/-} knockout and demonstrate that Sdc2 is lost in the vasculature. But they state (line 107): " However, unlike the report of profound developmental vascular defects following Sdc2 knockdown in zebrafish, we observed no gross vascular pathology in developing Sdc2^{-/-} embryos. " They then go on to generate and use a conditional knockout specific only for endothelial cells, and induced at P0-P5, to describe vascular defects. The global Sdc2^{-/-} is never mentioned again, nor used to corroborate these defects. This is raised as a concern because global knockouts often suffer from compensation, and this may be especially true for the syndecans. The authors need to be more clear regarding the phenotype of the Sdc2^{-/-} global knockout mouse for the benefit of the reader.

2. It is a bit of a concern that the authors use a conditional Sdc2^{-/-} knockout and compare it to a global Sdc4^{-/-} knockout to question if the two syndecans have shared or distinct roles in these cells. This is a concern because of Point #1 that there is concern that global syndecan knockouts display compensation and do not reveal the true knockout phenotype. A more acceptable comparison would have been an equivalent conditional knockout for both.

3. It is not clear what control is used for the Sdc2^{-/-} and Sdc4^{-/-} mice in Fig. 1. Are these different

mouse stains? Although the Sdc2^{-/-} mouse shows reduced retinal outgrowth, the same can be said for the control used for the Sdc4^{-/-} mouse. A comparison of the two controls would show a very distinct difference. More information should be provided to account for this difference. Are these different mouse strains, and, if so, is that also a factor in the differences seen between Sdc2 and Sdc4?

4. Given that the authors have chosen to compare Sdc2 and Sdc4 in their opening figures, it is a little surprising that they do not show the expression of both syndecans in the cells examined in Fig. 3, particularly as readers may assume that loss of one syndecan causes upregulation of another. Although not strictly required, it would be a definite improvement to the manuscript if the authors showed that there is no deficit or enhancement of Sdc4 expression in the Sdc2^{-/-} endothelial cells, despite the lessened activation of VEGFR2 (Fig. 3A), and that the sustained VEGFR2 signaling in the Sdc4^{-/-} cells is accompanied by sustained Sdc2 expression (Fig. 3D).

4. It is a concern that all of the studies use cells from knockout animals. The paper would be strengthened if siRNA/shRNA to Sdc2 and Sdc4 were introduced into the endothelial cells from a wt mouse and showed the same phenotype.

5. It is also a concern that all rescue experiments are performed with epitope-tagged syndecans. Although a common practice for many proteins, it was suggested some years ago by the Rosenberg lab that introducing epitope tags into syndecans altered the HS decoration of the proteins. Given the subject matter of the current work, it seems essential that the authors should introduce Sdc2 lacking an epitope tag and show identical rescue. If using Sdc2^{-/-} cells, there should be no need for the tag.

6. It is unfortunate that the authors do not make a stronger attempt to show HS-decorated syndecan in their blots, rather than just a narrow size range of HA-tagged protein, often without a size marker. It is presumed that the size shown represents the size of nascent Sdc2 or Sdc4 (MW markers and anticipated syndecan core protein sizes should be shown). But the nascent core protein, according to the findings in this body of work, is not the entity that engages VEGF-A165 and VEGFR2; thus, this is not a meaningful depiction of syndecan expression. It is not apparent from the Material and Methods or Figure Legends that the syndecan-containing samples were treated with heparinases prior to PAGE and western blot analysis, which means the amount of the syndecan shown is only a fraction of the total, and does not inform the reader about whether it is decorated with HS or not. In one example (Fig. 4E), a large portion of the blot is shown that reveals bands of many sizes, with the larger sizes presumably the HS-decorated protein. Frankly, the authors should show the larger sizes in all figures, if not actually showing how much core protein shows up on the blot with and without treatment with heparinases. Just showing a narrow size range of HA staining may be very misleading.

7. The syndecan (HA) blots shown in Fig. 4E leads to other questions: Are Sdc1 and Sdc3 decorated with HS? There seems to be a preponderance of undecorated core protein for Sdc1, Sdc2 and Sdc3. It is difficult to say for Sdc4. Is this a function of adenoviral overexpression? It is difficult to accept expression of protein that lacks HS because this is not a meaningful test for ability to bind VEGF. The authors need to take steps to show that these proteins are in fact HS-decorated. This is especially true when analyzing the Sdc2/Sdc4 chimeras and epitope tagged syndecans. In Fig. 3G, where the Sdc2^{ex}/Sdc4ⁱⁿ chimera rescues pVEGFR2, the HA tag band is much darker than the Sdc4^{ex}/Sdc2ⁱⁿ. This major difference in expression would seem to invalidate the finding, and, if the HA staining is to be accepted, indicates that the syndecan is decorated poorly, if at all, with HS.

Minor suggestions:

L57-59 "The process of chain modification is thought to be similar for different core proteins so that specific functional differences among various proteoglycans are attributed solely to core

protein diversity" This is such an important point, yet it risks not being clear to the reader. Re-wording and simplification is suggested.

Reviewer #2 - expert in proteoglycans and GAGs (Remarks to the Author):

It is significant that the authors demonstrate that different Sdc-2 constructs have different HS sulfation.

1. Line 28, The statement is too general. State specifically what aspects of syndecans-2 structure or functions remain unstudied.
2. Lines 34-35. It is not clear what is meant by this sentence. It is not correct to say that the N-terminal domain drives HS chain sulfation. The sulfation pattern differs according to the protein sequence but the driver is not clear.
3. Lines 87-88. The logic is very difficult to follow. The statement that the VEGF signaling defect resulted from reduction of HS 6-O sulfation on Sdc-2 in Sdc-2^{-/-} endothelial cells is nonsensical. It is nonsensical because Sdc-2 has been knocked out. We find out later that the cells were transduced with Sdc constructs. It would be a good idea to specify this here.
4. Line 270. It does not seem appropriate to say that the Sdc-2 N-terminal increases sulfation of its HS chains. Nor does it seem appropriate to say that Sdc-2 directs its HS chain sulfation. It would be more appropriate to say that the HS chain sulfation differs for different Sdc-2 constructs. The difference is the result of the passage of the Sdc-2 through the secretory pathway. This is a very complex process with many interactions with biosynthetic enzymes. To say that the protein directs HS sulfation is an overstatement.

Reviewer #3 - expert in angiogenesis (Remarks to the Author):

In the manuscript by Federico et al, the authors study the role of Sdc-2 in angiogenesis using the retinal and hindlimb ischemia models, concluding that it impairs VEGFA-165-driven angiogenesis. Mechanistically, and resorting exclusively to in vitro studies, the authors propose that the increased levels of 6-O sulfation of Sdc2 HS chains confer it the unique ability to form a ternary complex Sdc2-VEGFA165-VEGFR2, which is proposed to enhance VEGFR2 signalling.

The paper is Ok when it comes to demonstrating how different biochemical modifications may change the modulation of VEGFR2 signalling by Sdc2 in vitro. However, the in vivo analysis performed was very superficial, poorly quantified and there is no clear evidence that the molecular mechanism identified in vitro is at the source of the observed phenotype in vivo.

The manuscript needs a significant amount of new in vivo data to further validate the in vitro findings. In addition, the weak regulation of Vegfr2 signalling identified in Sdc2KO cells (-35% signaling found in ECs in vitro) and the very mild defects seen in Sdc2Pdgfb mutant retinas, shows that the regulation is not of enough biological significance.

Major concerns:

1. The current manuscript requires substantial new in vivo data to validate the in vitro results:
 - Analysis in Sdc2 mutants of Vegfr2 signalling levels and downstream targets.
 - Analysis of other vascular parameters in Sdc2 mutants, such as endothelial proliferation, density and sprouting.
 - Comparison of Sdc2 mutant phenotypes with Vegfr2^{+/-} or Vegfr2 Pdgfb/Cdh5-CreERT2 inducible heterozygous deletion phenotypes. These have a 50% reduction in VEGF signaling and allow for a

better phenotypic comparison.

- Mutate the endogenous Sdc2 gene, to analyse the impact on VEGfr signaling and angiogenesis of the same Sdc2 mutant forms characterized in vitro.
- If possible modulate Sdc2 glycosylation and sulfation in vivo, or its binding to VEGF.

2. The authors have not presented and discussed all existing bibliographic data on the other published work regarding the role of Sdc2 in angiogenesis. Papers from Rossi et al (2014) *J. Cell Science* 127, 4788 and Noguer et al (2009) *Exp Cell Res* 315, 795-808 are two such examples. These papers propose opposite or similar roles of Sdc2, to the ones presented in this manuscript. This data has to be carefully contrasted and discussed to reveal what is novel and different in this manuscript.

3. The authors present that Sdc4 is important in FGF-dependent signal transduction however no significant effect is obtained in the cornea pocket assay with FGF2 pellets. Surprisingly, the authors claim that "Sdc4^{-/-}, but not Sdc2iCdh5 showed a reduction in angiogenesis in response to FGF2 (Fig. 1g-h)". The indicated figures show Sdc4 mutants present no difference in both VEGF- and FGF2-induced angiogenesis when compared to control mice.

4. Analysis of the EC-specific Sdc2 LOF leads to delayed vessel outgrowth. Retinas in Fig1A appear to have different sizes, ie., the Sdc2 mutant whole retina tissue seems smaller. Instead of presenting the absolute values of vascular outgrowth, it would be better to normalize the vascular outgrowth to the total size of the retina. The control and mutants do not seem stage matched.

Minor Concerns:

1. Authors show by immunostaining that Sdc2 is expressed in ECs of the aorta and in the retina, where it seems that it is mostly enriched in arteries and arterioles. Images representative of all regions of the retina (vein and venous plexus, angiogenic front and sprouts) are necessary.
2. None of the images presented in the manuscript have scale bars.
3. No data has been presented regarding the endothelial expression of Sdc4.
4. Authors do not attribute the decrease in vessel density to enhanced vascular pruning in the Sdc2 mutants because "collagen IV staining did not detect empty sleeves". Since collagen empty sleeves are indeed detectable in Fig S2A, we infer that the authors meant that there were no differences in the frequency of empty sleeves between control and mutant retinas. However, no quantification data is presented.
5. Quantifications on the density of vascular networks in the skin and kidney are not presented.
6. Although the authors claim the the role of Sdc2 on angiogenesis is mediated through the VEGFa165-VEGFR2 axis, there is also evidence that Sdc2 also interacts with integrins (*Exp. Cell. Res* (2000), Vol.256, 434 and *Biophysics Res Commun* (2009), Vol385, 231), which is important for cell migration and adhesion. In fact, there are several papers that show VEGFA-mediated functions dependent on integrins (*Blood* 2012 120:4892-4902; *J Biol Chem.* 2011 286: 1083–1092; *JBC* 2007 282, 15187-15196; *Embo J* 1999;18:882–892; *Mol Cell* 2000;6:851–860; *Circ Res* 2007;101:570–580). To assess integrin signaling, immunostaining for p-FAK or active-integrinB1 could be used.
7. In vivo validation of the in vitro co-IP data would also be interesting, using for example the tissues from animals.
8. It would be important to check by western blot if Sdc4 is regulated in the absence of Sdc2 and vice-versa.
9. In Fig3H and Fig 4G, it appears that Sdc2 and Sdc4 have opposite effects on p-VEGFR2 (comparing to cells transduced with only Ad-GFP). Could the authors comment on this?
10. All interaction assays were performed in cells transduced with adenovirus and expressing different mutated forms of Sdc2/4. It is known that over-expression of proteins can sometimes lead to unspecific protein interactions. How do the total levels of these proteins compare to the endogenous levels? Can the same experiments be performed with CRISPR/Cas9 induced mutations in Sdc2?

11. Although the protein sequence of the D1 domains of Sdc2 are well conserved among species, I wonder whether the glycosylation events are also well conserved, since this is known to vary depending on cell context. It would be important to confirm that glycosylation pattern in mouse endothelial cells are identical to those described in the manuscript and ideally in the absence of overexpression of the proteins.

Reviewer #1 - expert in syndecans (Remarks to the Author):

Heparan sulfate (HS) glycosaminoglycan chains have very important roles in development and disease, and numerous studies indicate that these roles are dictated by the sulfation pattern within the HS chain. **The question of whether or not the sulfation pattern is distinctive of the core protein on which they are synthesized, which implies the presence of regulatory determinants in the protein itself, rather than distinctive of a given cell type and the enzymes that it expresses in the Golgi apparatus, has been debated now for several decades.** But the studies investigating the presence of potential regulatory elements in the core protein have been few and mixed, leaving it still a very open-ended question after years of progress on understanding the biology of the HS chains themselves.

The Simons group has now shown that a determinant (although a rather large protein of the core protein extracellular domain) in syndecan-2 (Sdc2) appears to dictate the composition of the HS chain in endothelial cells. Furthermore, the sulfate composition of the chains favors the binding of VEGF-A165, one of the several VEGF angiogenic factors that is known to bind HS. What this determinant actually does is not addressed, which is a little disappointing, but the authors have established an assay system that should allow them to get at this. **If true, this is the first meaningful demonstration of such a determinant, and I suspect it will lead to the discovery of others. The work is very important.**

Reply: *We thank the reviewer for kind comments and high opinion of the work's importance.*

Nonetheless, there are some nagging concerns with the work. One concern is an inadequate description of the angiogenic response of the Sdc2-/- "global" knockout mouse. The authors indicate that they saw no profound pathological effect in the global knockout, but then go on to describe a defect using an endothelial cell-specific inducible knockout activated after birth of the pups. Another concern is the use of epitope-tagged syndecans that are likely overexpressed when introduced in adenoviral vectors. Either condition (epitope tag or overexpression) can interfere with heparan sulfate synthesis on the core protein. These pesky concerns detract from the excellence of the finding, especially as it is being considered for publication in a top journal. Despite these concerns, however, the authors arrive at a consistent conclusion by the end of the manuscript, and can demonstrate that Sdc2 and Sdc4 isolated from human umbilical vein endothelial cells do indeed appear to bear distinct HS that differs in its capacity to bind VEGF-A165.

Specific comments:

1. The authors generate an ES cell “global” Sdc2^{-/-} knockout and demonstrate that Sdc2 is lost in the vasculature. But they state (line 107): “However, unlike the report of profound developmental vascular defects following Sdc2 knockdown in zebrafish, we observed no gross vascular pathology in developing Sdc2^{-/-} embryos. “They then go on to generate and use a conditional knockout specific only for endothelial cells, and induced at P0-P5, to describe vascular defects. The global Sdc2^{-/-} is never mentioned again, nor used to corroborate these defects. This is raised as a concern because global knockouts often suffer from compensation, and this may be especially true for the syndecans. The authors need to be more clear regarding the phenotype of the Sdc2^{-/-} global knockout mouse for the benefit of the reader.

Reply: *We are sorry for the confusion and poor choice of wording. What we meant to say is that although global Sdc2 KD led to profound suppression of vascular development, surprisingly global Sdc2^{-/-} mice did not reveal embryonic lethality as one might have expected from zebrafish studies.*

We did observe a delay in retina angiogenic development similar to that we described in Sdc2 endothelial-specific KO. This information is now expanded in the text (page 5, lines 115-118) and global Sdc2^{-/-} retina’s phenotype is shown in Supplementary Figure 2A-C.

Furthermore, similar to previous reports for Sdc1^{-/-} and Sdc4^{-/-} mice, we also observed a delay in skin wound healing in global Sdc2^{-/-} mice. This information has been added to text and data are now shown in Supplementary Figure S2D-E

2. It is a bit of a concern that the authors use a conditional Sdc2^{-/-} knockout and compare it to a global Sdc4^{-/-} knockout to question if the two syndecans have shared or distinct roles in these cells. This is a concern because of Point #1 that there is concern that global syndecan knockouts display compensation and do not reveal the true knockout phenotype. A more acceptable comparison would have been an equivalent conditional knockout for both.

Reply: *We agree that it would be nice to have an endothelial-specific Sdc4 KO mouse. Such a line, however, was never generated, to the best of our knowledge. The reason is that given the absence of a vascular phenotype in Sdc4 global KO, it seems highly unlikely that one would be observed after an endothelial-specific knockout. Indeed, we are not aware of any instances when a phenotype was seen in a cell-type specific KO when a global KO shows no abnormalities. Nevertheless, to better address this point, we have now added a description of vascular abnormalities in global Sdc2^{-/-} mice. When compared to the global Sdc4^{-/-} KO, one can see that vascular abnormalities are limited to Sdc2 null mice. Please also see the reply to point 1 above.*

Since any mechanistic interpretation of vascular phenotypes in global Sdc2^{-/-} mice is rather difficult since Sdc2 is expressed in smooth muscle cells and may play a role in vessel wall

maturation (which is also critical for proper angiogenesis/arteriogenesis). Thus, to confirm ECs importance in the observed phenotype we generated and characterized *Sdc2* endothelial-specific KO mice. As shown in the additional data presented throughout the revision, we do not see an evident compensatory mechanism in *Sdc2*^{-/-} mice:

1) Constitutive global *Sdc2*^{-/-} mice show an angiogenic delay similar to that seen in *Sdc2* endothelial-specific KO which where *Cre* is activated at P1.

2) VEGFA signaling defect is observed in both EC derived from mice with a constitutive deletion of *Sdc2* (Fig.3A-C) and in WT ECs following *Sdc2* silencing (supplementary Figure 5A-C)

3) We have previously published that *Sdc4* silencing in endothelial cells does not affect VEGFA signaling nor its biological effects in vitro (Corti et al, JBC). In line with these observations, we did not find angiogenesis defects in *Sdc4*^{-/-} mice. Information regarding *Sdc4* silencing and VEGFA has been added to text (page 8, lines 175-177)

3. It is not clear what control is used for the *Sdc2*^{-/-} and *Sdc4*^{-/-} mice in Fig. 1. Are these different mouse stains? Although the *Sdc2*^{-/-} mouse shows reduced retinal outgrowth, the same can be said for the control used for the *Sdc4*^{-/-} mouse. A comparison of the two controls would show a very distinct difference. More information should be provided to account for this difference. Are these different mouse strains, and, if so, is that also a factor in the differences seen between *Sdc2* and *Sdc4*?

Reply: For conditional *Sdc2*^{iPdgfb}, *Sdc2*^{fl^{ox}/fl^{ox}} *Cre* negative littermates were used as controls (both groups were equally exposed to tamoxifen). For *Sdc4*^{-/-} mice, littermates wild-type (*Sdc4*^{+/+}) mice were used as controls. This information is now spelled out in all figure panels.

The quantification of retina outgrowth measures the distance between the center of the retina (entrance point of the retinal artery) and the vascular front. Differences in sizes of retinas in the avascular parts are due technical variability during eye opening and sample flattening needed for retina staining. We have not observed any differences in retinal sizes across genotypes.

To further clarify this issue the progression of vascular retinal development for each mouse strain is now shown as normalized to *Sdc2*^{fl^{ox}/fl^{ox}} (new quantification in Fig. 1B).

4. Given that the authors have chosen to compare *Sdc2* and *Sdc4* in their opening figures, it is a little surprising that they do not show the expression of both syndecans in the cells examined in Fig. 3, particularly as readers may assume that loss of one syndecan causes upregulation of another. Although not strictly required, it would be a definite improvement to the manuscript if

the authors showed that there is no deficit or enhancement of Sdc4 expression in the Sdc2^{-/-} endothelial cells, despite the lessened activation of VEGFR2 (Fig. 3A), and that the sustained VEGFR2 signaling in the Sdc4^{-/-} cells is accompanied by sustained Sdc2 expression (Fig. 3D).

Reply: We now show both Sdc4 in Figure 3A (Sdc2^{-/-} ECs) and Sdc2 in Figure 3D (Sdc4^{-/-} ECs). We did not detect any evident upregulation.

4. It is a concern that all of the studies use cells from knockout animals. The paper would be strengthened if siRNA/shRNA to Sdc2 and Sdc4 were introduced into the endothelial cells from a wt mouse and showed the same phenotype.

Reply: Effect of Sdc2 silencing in WT ECs on VEGFR2 activation is now shown in Figure S5. Similar to Sdc2^{-/-} ECs, we detected reduced VEGFR2 activation after Sdc2 KD.

We have previously published that Sdc4 silencing does not alter VEGFA-VEGFR2 signaling or VEGFA biological effects in vitro (Corti et al. JBC 2013). This information is added to and cited in the text (page 8, lines 175-177)

5. It is also a concern that all rescue experiments are performed with epitope-tagged syndecans. Although a common practice for many proteins, it was suggested some years ago by the Rosenberg lab that introducing epitope tags into syndecans altered the HS decoration of the proteins. Given the subject matter of the current work, it seems essential that the authors should introduce Sdc2 lacking an epitope tag and show identical rescue. If using Sdc2^{-/-} cells, there should be no need for the tag.

Reply: This is an interesting point. The Rosenberg studies (performed while the senior author was a postdoc there) used in vitro microsomal GAG synthesis assay. A massive (>100 fold) increase in expression of one core protein did affect the extent of glycosylation on others. To our knowledge, this has never been seen in vivo or in any physiological settings. Nevertheless, we have tried to address this concern by performing a number new studies:

1) As suggested by the reviewer, we show rescue of VEGFA-VEGFR2 signaling activation by re-introduction of a native mouse Sdc2 (without epitope-tag) in Sdc2^{-/-} ECs (Supplementary Figure 5F-G).

2) We generated syndecan constructs with a different tag (Flag instead of HA) or the same HA-tag in different position (C-instead of N-terminal) and show that specificity of the Sdc2-VEGFR2 association is conserved (supplementary figure 6C-D).

3) We show that our IP experiments with adenoviruses are titrated to achieve expression level that is similar to syndecans endogenous level (supplementary figures 6A-B)

6. It is unfortunate that the authors do not make a stronger attempt to show HS-decorated syndecan in their blots, rather than just a narrow size range of HA-tagged protein, often without a size marker. It is presumed that the size shown represents the size of nascent Sdc2 or Sdc4 (MW markers and anticipated syndecan core protein sizes should be shown). But the nascent core protein, according to the findings in this body of work, is not the entity that engages VEGF-A165 and VEGFR2; thus, this is not a meaningful depiction of syndecan expression. It is not apparent from the Material and Methods or Figure Legends that the syndecan-containing samples were treated with heparinases prior to PAGE and western blot analysis, which means the amount of the syndecan shown is only a fraction of the total and does not inform the reader about whether it is decorated with HS or not. In one example (Fig. 4E), a large portion of the blot is shown that reveals bands of many sizes, with the larger sizes presumably the HS-decorated protein. **Frankly, the authors should show the larger sizes in all figures, if not actually showing how much core protein shows up on the blot with and without treatment with heparinases.** Just showing a narrow size range of HA staining may be very misleading.

Reply: All full blots showing either syndecans or HA expression (with appropriate size marker and core MW) are now provided either in main figures (for figures 3H, 4E-F-G, 5E, 5F) or Supplementary Figure 7 (for figures 3A, 3D, 4A-B-C-D)

Material and methods section dealing with this issue has been rewritten to clarify this (page 16 – added specific paragraph to method: “detection of syndecan expression by western blot”).

HA antibody detect both nascent core protein (dimers) and higher MW glycosylation-isoforms smears. HA detection is done without heparinase treatment of syndecans samples.

Sdc4 antibody recognize a specific band below ~20 KDa (Supplementary Figure 7: Fig 3D -Sdc4) without heparinase treatment. The band is believed to be the naked core protein in monomeric form. The Sdc4 antibody used in these studies is a monoclonal Ab against a peptide-sequence that encompass C-terminal domain of Sdc4 (abcam #ab24511).

The Sdc2 antibody (for mouse Sdc2 RD#AF6585) detects specific bands (core at ~37 KDa MW ladder and higher MV isoforms) only after heparinase treatment (Supplementary Figure 7: Fig 3A-Sdcc2). This a polyclonal antibody raised against recombinant Sdc2 ED. Heparinase treatment appear necessary to unveil epitopes on the core protein and to remove most of non-specific sugar epitopes. Before (-heparinase) and after heparinase (+ heparinase) treatment Sdc2 blots are now shown in Supplementary figure 7: 3A-Sdc2 and 3D-Sdc2

7. The syndecan (HA) blots shown in Fig. 4E leads to other questions: Are Sdc1 and Sdc3 decorated with HS? There seems to be a preponderance of undecorated core protein for Sdc1, Sdc2 and Sdc3. It is difficult to say for Sdc4. Is this a function of adenoviral overexpression? It is difficult to accept expression of protein that lacks HS because this is not a meaningful test for ability to bind VEGF. The authors need to take steps to show that these proteins are in fact HS-decorated. This is especially true when analyzing the Sdc2/Sdc4 chimeras and epitope tagged syndecans. In Fig. 3G, where the Sdc2ex/Sdc4in chimera rescues pVEGFR2, the HA tag band is much darker than the Sdc4ex/Sdc2in. This major difference in expression would seem to invalidate the finding, and, if the HA staining is to be accepted, indicates that the syndecan is decorated poorly, if at all, with HS.

Reply: *Regarding Figure 4E, we now provide a full blot that show core and decoration in all syndecans. There seem to be quite comparable core amount and HS decoration level across all isoforms. For figure 4G, we have expanded the observation and provide a better representative picture for the blot. We have also added quantification of the rescue with chimeras (in a similar manner as shown for Figure 3I)*

Minor suggestions:

L57-59 “The process of chain modification is thought to be similar for different core proteins so that specific functional differences among various proteoglycans are attributed solely to core protein diversity” This is such an important point, yet it risks not being clear to the reader. Re-wording and simplification is suggested.

Reply: *Thank you. We have attempted to do this.*

Reviewer #2 - expert in proteoglycans and GAGs (Remarks to the Author):

It is significant that the authors demonstrate that different Sdc-2 constructs have different HS sulfation.

1. Line 28, The statement is too general. State specifically what aspects of syndecans-2 structure or functions remain unstudied.

Reply: *This has been done as requested (page 2, lines 29-31).*

2. Lines 34-35. It is not clear what is meant by this sentence. It is not correct to say that the N-terminal domain drives HS chain sulfation. The sulfation pattern differs according to the protein sequence but the driver is not clear.

Reply: *The sentence has been re-written it now states that: The latter finding was accounted for by a 59 amino acid sequence in the N-terminal domain of Sdc2 that conferred the ability to specifically enhance 6-O sulfation” (Page 5, line 96-97).*

3. Lines 87-88. The logic is very difficult to follow. The statement that the VEGF signaling defect resulted from reduction of HS 6-O sulfation on Sdc-2 in Sdc-2^{-/-} endothelial cells is nonsensical. It is nonsensical because Sdc-2 has been knocked out. We find out later that the cells were transduced with Sdc constructs. It would be a good idea to specify this here.

Reply: *We apologize for the confusion. The text has been amended to fix this issue. What we meant to say in that Sdc2 chains binds VEGFA more efficiently than Scc4 chains because of higher 6-O sulfation (page 4, line 92-96).*

4. Line 270. It does not seem appropriate to say that the Sdc-2 N-terminal increases sulfation of its HS chains. Nor does it seem appropriate to say that Sdc-2 directs its HS chain sulfation. It would be more appropriate to say that the HS chain sulfation differs for different Sdc-2 constructs. The difference is the result of the passage of the Sdc-2 through the secretory pathway. This is a very complex process with many interactions with biosynthetic enzymes. To say that the protein directs HS sulfation is an overstatement.

Reply: *Thank you. This has been re-written (page 15, line 334-338)*

Reviewer #3 - expert in angiogenesis (Remarks to the Author):

In the manuscript by Federico et al, the authors study the role of Sdc-2 in angiogenesis using the retinal and hindlimb ischemia models, concluding that it impairs VEGFA-165-driven angiogenesis. Mechanistically, and resorting exclusively to in vitro studies, the authors propose that the increased levels of 6-O sulfation of Sdc2 HS chains confer it the unique ability to form a ternary complex Sdc2-VEGFA165-VEGFR2, which is proposed to enhance VEGFR2 signaling.

The paper is Ok when it comes to demonstrating how different biochemical modifications may change the modulation of VEGFR2 signaling by Sdc2 in vitro. However, the in vivo analysis performed was very superficial, poorly quantified and there is no clear evidence that the

molecular mechanism identified in vitro is at the source of the observed phenotype in vivo.

The manuscript needs a significant amount of new in vivo data to further validate the in vitro findings. In addition, the weak regulation of Vegfr2 signaling identified in Sdc2KO cells (-35% signaling found in ECs in vitro) and the very mild defects seen in Sdc2Pdgfb mutant retinas, shows that the regulation is not of enough biological significance.

*We thank the reviewer suggestions to improve the manuscript. Importantly, we now show strong **in vivo evidence** that a 35% reduction in VEGFR2 signaling activation has biological relevance and it is involved in determining the vascular phenotype shown in Sdc2 endothelial-specific KO.*

Major concerns:

1. The current manuscript requires substantial new in vivo data to validate the in vitro results:
- Analysis in Sdc2 mutants of Vegfr2 signalling levels and downstream targets.

Reply: *Similar to our in vitro data, we now show significantly reduced VEGFR2 activation in whole-retinal lysate (P6) after Tamoxifen-induced Sdc2 endothelial deletion (P1 to P5) (Supplementary Figure 5D-E)*

- Analysis of other vascular parameters in Sdc2 mutants, such as endothelial proliferation, density and sprouting.

Reply: *Characterization of the retinal vascular phenotype in Sdc2 ECKO mice has been expanded. We now show reduced number of tip cells (Supplementary figure 3C-D). Reductions in vascular density and proliferation are also observed in Sdc2 ECKO (supp. Figure 3E-F)*

Additionally, we have performed analysis of VEGFA-induced biological effects in vitro. We show that Sdc2^{-/-} ECs have reduced VEGFA-induced proliferation and migration, while FGF2 responses are not affected (Supplementary Figure S4)

- Comparison of Sdc2 mutant phenotypes with Vegfr2^{+/-} or Vegfr2 Pdgfb/Cdh5-CreERT2 inducible heterozygous deletion phenotypes. These have a 50% reduction in VEGF signaling and allow for a better phenotypic comparison.

Reply: There is a large body of literature dealing with these phenotypes. Given the number of published studies, it seems hardly reasonable to carry out yet another VEGFR2 het study. The information below has been added to the Discussion (page 13, lines 297-311):

- VEGFA concentration is the key limiting factor regulating VEGFR2 signaling. That is, VEGFR2 signaling is ligand-dependent and the amount of ligand available to bind VEGFR2 is far less than the number of VEGFR2 binding sites available. Thus, a deletion of a single VEGFR2 allele DOES NOT lead to a 50% reduction in VEGFA signaling while a single allele deletion of VEGFA is lethal.
- Shalaby et al (Nature, 1996) show no measurable phenotypes during embryonic development in VEGFR2 hets.
- Lars Jakobsson (Gerhardt lab, Nat Cell Biol 2010) showed in competition assays that VEGFR2^{+/-} ECs move slower than VEGFR2^{+/+} ECs. This is really a migration and not a proliferation assay. In agreement with this, we also see decreased migration of Sdc2^{-/-} ECs in response to VEGFA.
- Silvaraj et al (Dev Cell 2013) show 15% reduction in the retinal angiogenesis extent and ~35% reduction in the number of branch points in VEGFR2 hets generated with Pdgfb-*iCre* (in Suppl Fig 2F, G). These numbers are consistent with our observations.

In summary, 35% reduction in VEGF signaling here is highly biologically meaningful given that 50% reduction is lethal. VEGF signaling is driven by ligand availability. receptor levels are of secondary importance.

Sdc2 role in facilitating VEGFA-VEGFR2 complex formation is critical as the absence of Sdc2 leads to a 35% reduction in signaling with measurable *in vivo* biological effects.

- Mutate the endogenous Sdc2 gene, to analyze the impact on Vegfr signaling and angiogenesis of the same Sdc2 mutant forms characterized *in vitro*.

Reply: *This is an interesting idea that, we believe, is outside of the scope of this manuscript that deals with the effect of protein sequence on a sulfation pattern. Especially given how long it would take to do these studies and how much this would cost.*

*We clearly show that reduced VEGFA signaling in Sdc2^{-/-} EC (due to specific loss of Sdc2 HS-chains) is involved in determining vascular phenotype *in vivo* and phenotypic effects *in vitro*. This does not exclude that Sdc2 deletion may also have other HS-independent consequences (i.e. due to loss of Sdc2 core protein)*

- If possible modulate Sdc2 glycosylation and sulfation *in vivo*, or its binding to VEGF.

Reply: *Currently, there is not feasible way to selectively modulate glycosylation or sulfation *in vivo* or *in vitro*.*

2. The authors have not presented and discussed all existing bibliographic data on the other published work regarding the role of Sdc2 in angiogenesis. Papers from Rossi et al (2014) J. Cell Science 127, 4788 and Noguer et al (2009) Exp Cell Res 315, 795-808 are two such examples. These papers propose opposite or similar roles of Sdc2, to the ones presented in this manuscript. This data has to be carefully contrasted and discussed to reveal what is novel and different in this manuscript.

***Reply:** These have been added to Discussion (page 14, lines 310-316):*

3. The authors present that Sdc4 is important in FGF-dependent signal transduction however no significant effect is obtained in the cornea pocket assay with FGF2 pellets. Surprisingly, the authors claim that “Sdc4^{-/-}, but not Sdc2iCdh5 showed a reduction in angiogenesis in response to FGF2 (Fig. 1g-h)”. The indicated figures show Sdc4 mutants present no difference in both VEGF- and FGF2-induced angiogenesis when compared to control mice.

***Reply:** We apologize for the writing error. Indeed, we intended to write that both VEGFA- and FGF2 -induced angiogenesis in cornea-pocket assay WAS NOT affected in Sdc4^{-/-} mice.*

Sdc4 controls other biological aspects related to FGF-signaling, particularly mTOR/AKT/eNOS pathway which modulate systemic blood pressure.

4. Analysis of the EC-specific Sdc2 LOF leads to delayed vessel outgrowth. Retinas in Fig1A appear to have different sizes, ie., the Sdc2 mutant whole retina tissue seems smaller. Instead of presenting the absolute values of vascular outgrowth, it would be better to normalize the vascular outgrowth to the total size of the retina. The control and mutants do not seem stage matched.

***Reply:** The quantification of retina outgrowth measures the distance between the center and vascular front. Difference size of retinas in the avascular parts are due technical variability during eye opening and flattening for retinal staining. We have not observed significant differences between total size of retinas across genotypes.*

Additionally, for reader's confidence, the vascular progression of each strain is now shown as normalized to Sdc2^{fl/fl} (new quantification in Fig. 1B).

Minor Concerns:

1. Authors show by immunostaining that Sdc2 is expressed in ECs of the aorta and in the retina, where it seems that it is mostly enriched in arteries and arterioles. Images representative of all regions of the retina (vein and venous plexus, angiogenic front and sprouts) are necessary.

Reply: *Additional staining of retinas showing Sdc2 expression have been added (Supplementary figure 1e). Syndecan-2 is expressed in arteries, veins and capillaries. It may be true that Sdc2 is enriched in arterial versus venous EC, however the fact that it is expressed in both SMCs and ECs does not allow us to conclude this.*

2. None of the images presented in the manuscript have scale bars.

Reply: *scale bars have been added*

3. No data has been presented regarding the endothelial expression of Sdc4.

Reply: *There are numerous publications demonstrating high Sdc4 expression in ECs, including in the retinal vasculature. This information, along with citations to most recent papers, has been added to the text (page 6, lines 132-133).*

4. Authors do not attribute the decrease in vessel density to enhanced vascular pruning in the Sdc2 mutants because “collagen IV staining did not detect empty sleeves”. Since collagen empty sleeves are indeed detectable in Fig S2A, we infer that the authors meant that there were no differences in the frequency of empty sleeves between control and mutant retinas. However, no quantification data is presented.

Reply: *Quantification has been added (Supplementary Figure 3B)*

5. Quantifications on the density of vascular networks in the skin and kidney are not presented.

Reply: *We have decided to remove these data and to focus on retina vascular development and hindlimb ischemia, two settings in which relative contribution of VEGFA versus other factors is extremely well characterized.*

6. Although the authors claim the role of Sdc2 on angiogenesis is mediated through the VEGFa165-VEGFR2 axis, there is also evidence that Sdc2 also interacts with integrins (Exp. Cell. Res (2000), Vol.256, 434 and Biophysics Res Commun (2009), Vol385, 231), which is important for cell migration and adhesion. In fact, there are several papers that show VEGFA-mediated functions dependent on integrins (Blood 2012 120:4892-4902; J Biol Chem. 2011 286: 1083–

1092; JBC 2007 282, 15187-15196; Embo J 1999;18:882–892; Mol Cell 2000;6:851–860; Circ Res 2007;101:570–580). To assess integrin signaling, immunostaining for p-FAK or active-integrinB1 could be used.

Reply: We agree that integrins are important in VEGFr2 signaling. How that integrates with Sdc2 is less clear and is the subject for further studies. Our data clearly show that Sdc2 is essential for formation of VEGFA165-VEGFR2-Sdc2 complex which, in turn, is required for maximal VEGFR2 phosphorylation. Whether an integrin, such as beta-2 is involved is not certain. Integrins activation is one of the VEGFR2 downstream target and likely to be affected as the other downstream targets. This possibility and relative references are added to discussion as suggested. (pages 14, lines 317-323).

7. In vivo validation of the in vitro co-IP data would also be interesting, using for example the tissues from animals.

Reply: In the proteoglycans world, IP and WB experiments are severely limited by lack of working antibodies against endogenous proteins. Sdc2 is no exceptions. This is the reason why most studies in this field rely on the use of epitope-tags and overexpression systems. There is no possibility of getting a Sdc2-specific IP from a tissue lysate.

8. It would be important to check by western blot if Sdc4 is regulated in the absence of Sdc2 and vice-versa.

Reply: As requested, we now show Sdc4 blot in Figure 3A (Sdc2-/- ECs) and Sdc2 blot in Figure 3D (Sdc4-/- ECs). We did not detect any evident upregulation.

9. In Fig3H and Fig 4G, it appears that Sdc2 and Sdc4 have opposite effects on p-VEGFR2 (comparing to cells transduced with only Ad-GFP). Could the authors comment on this?

Reply: The only statistically significant effect we see in these studies is when Sdc2 is reintroduced into Sdc2-/- ECs.

10. All interaction assays were performed in cells transduced with adenovirus and expressing different mutated forms of Sdc2/4. It is known that over-expression of proteins can sometimes lead to unspecific protein interactions. How do the total levels of these proteins compare to the endogenous levels? Can the same experiments be performed with CRISPR/Cas9 induced mutations in Sdc2?

Reply: We agree this is an important point. IP experiments following adenovirus transduction (at MOI = ~ 1-2) were performed with minimal overexpression which was similar to syndecans endogenous level (supplementary figures 6A-B)

CRISPR/Cas9 mutagenesis would be a whole new study and approach would still be limited by lack of good antibody against endogenous protein without epitope-tag.

11. Although the protein sequence of the D1 domains of Sdc2 are well conserved among species, I wonder whether the glycosylation events are also well conserved, since this is known to vary depending on cell context. It would be important to confirm that glycosylation pattern in mouse endothelial cells are identical to those described in the manuscript and ideally in the absence of overexpression of the proteins.

Reply: That would be a good study to do. Unfortunately, high-quality antibody for affinity purification of endogenous syndecans do not exist. Nevertheless, the point of testing a different species is well taken and we have repeated HS chain analysis of both *sdc2* and *sdc4* isolated from a mouse endothelial cell line. We now show that also mouse ECs generate higher level of 6-O-sulfation in *Sdc2* HS chains than in *Sdc4* HS chains (Supplementary Figure 6E). This is similar to what we observed in HUVECs

Reviewers' comments:

Reviewer #1 (Remarks to the Author):

The authors have satisfactorily answered the questions from the prior review in part by correcting misconceptions by clarifying their writing, and by performing additional experimentation. The original claims remain true and are compelling. The most important finding in this work is that a determinant in the core protein of syndecan-2 potentially regulates the sulfation pattern of the heparan sulfate chains synthesized on the protein. The second important finding is that sulfation pattern is not only different, but it supports a syndecan-2-specific function that is not shared with its closest homologue, syndecan-4, nor apparently with syndecans 1 and 3 which are less homologous. This is a "first of its kind" demonstration that could well be paradigm changing.

The work appears to be statistically significant and is conducted in such a manner, e.g., supplementing the animal models with cultured cells and comparing Sdc2 versus Sdc4, that the findings are convincingly reproducible.

Minor point:

L 155-160 The authors describe a comparison between Sdc2^{-/-} and Sdc4^{-/-} endothelial cells, yet the actual comparison as shown in Supplementary Figure 4a-f appears to be between wild-type and Sdc2^{-/-}. The data as shown are fine, but the text needs to be corrected to fit the figure.

Reviewer: Alan C. Rapraeger

Reviewer #2 (Remarks to the Author):

The extent to which different proteoglycans expressed by the same cell differ in HS chain structure has been a long simmering controversy. The authors have performed careful and convincing studies that show differences in 6-O-sulfation for syndecan-2 versus syndecan-4 that cause differences in VEGFA-driven angiogenic processes. This work is significant in correlating differences in HS structure with angiogenic function.

Line 542, define HA in the text. The reader may confuse the HA epitope from hemagglutinin with the abbreviation commonly used for hyaluronan.

Reviewer #3 (Remarks to the Author):

Reviewer #3 - expert in angiogenesis (Remarks to the Author):

In the manuscript by Federico et al, the authors study the role of Sdc-2 in angiogenesis using the retinal and hindlimb ischemia models, concluding that it impairs VEGFA-165-driven angiogenesis. Mechanistically, and resorting exclusively to in vitro studies, the authors propose that the increased levels of 6-O sulfation of Sdc2 HS chains confer it the unique ability to form a ternary complex Sdc2-VEGFA165-VEGFR2, which is proposed to enhance VEGFR2 signaling.

The paper is Ok when it comes to demonstrating how different biochemical modifications may change the modulation of VEGFR2 signaling by Sdc2 in vitro. However, the in vivo analysis performed was very superficial, poorly quantified and there is no clear evidence that the molecular mechanism identified in vitro is at the source of the observed phenotype in vivo.

The manuscript needs a significant amount of new in vivo data to further validate the in vitro findings. In addition, the weak regulation of Vegfr2 signaling identified in Sdc2KO cells (-35% signaling found in ECs in vitro) and the very mild defects seen in Sdc2Pdgb mutant retinas, shows that the regulation is not of enough biological significance.

Reply: We thank the reviewer suggestions to improve the manuscript. Importantly, we now show strong **in vivo evidence** that a 35% reduction in VEGFR2 signaling activation has biological relevance and it is involved in determining the vascular phenotype shown in Sdc2 endothelial- specific KO.

R#3 Reply: The authors still do not provide clear evidence that VEGFR2 signalling is decreased in endothelial cells *in vivo* (see below), and how the estimated 35% reduction in VEGFR2 signalling causes the observed *in vivo* vascular/angiogenesis phenotype, which is one of the most important aspects of this work.

Major concerns:

1. The current manuscript requires substantial new in vivo data to validate the in vitro results:
- Analysis in Sdc2 mutants of Vegfr2 signalling levels and downstream targets.

Reply: Similar to our in vitro data, we now show significantly reduced VEGFR2 activation in whole-retinal lysate (P6) after Tamoxifen-induced Sdc2 endothelial deletion (P1 to P5) (Supplementary Figure 5D-E)

R#3 Reply: Unfortunately, the authors' choice to assess VEGFR2 signalling in lysates of whole postnatal retinas was not the best. Okabe et al (2014) Cell, 159, 584-96 and other papers have

shown that in the retina, VEGFR2 is expressed in neurons at levels that are much higher than those present in ECs. Moreover, ECs are only a very minor fraction (less than 1%) of the entire postnatal retina tissue. Thus, western blot of whole retinal lysates cannot be used to evaluate VEGFR2 activity in endothelial cells. Authors should have instead performed immunostaining to profile the protein or phosphorylation levels of Vegfr2 or several published canonical downstream targets (ERK, Esm1, Vegfr3, etc...). The authors can also FACS sort ECs from different tissues of their mutants, and perform Western/qRT-PCR analysis for Vegfr2 signalling or downstream targets.

In the discussion line 318, the authors mention “we have not investigated whether Sdc2 deletion differentially affects various VEGFR2 downstream signaling pathways (e.g. PLC γ /ERK, AKT/Src, integrin activation, etc...)”. This reviewer believes it is of high relevance to do this, in order to support the paper’s main message.

Moreover, Okabe et al (2014) Cell, 159, 584-96 have also shown that a reduction in neuronal VEGFR2 signalling alone leads to alterations in retinal angiogenesis, with retinas displaying “delays in the radial outgrowth and decreased vascular density in the superficial plexus when compared to control mice”. Given the authors whole retina lysate western blot results, do the authors conclude that VEGFR2 signalling is also impaired in the neurons of their Sdc2 EC-specific mutants? The authors should comment/address this issue.

- Analysis of other vascular parameters in Sdc2 mutants, such as endothelial proliferation, density and sprouting.

Reply: *Characterization of the retinal vascular phenotype in Sdc2 ECKO mice has been expanded. We now show reduced number of tip cells (Supplementary figure 3C-D). Reductions in vascular density and proliferation are also observed in Sdc2 ECKO (supp. Figure 3E-F)*

Additionally, we have performed analysis of VEGFA-induced biological effects in vitro. We show that Sdc2^{-/-} ECs have reduced VEGFA-induced proliferation and migration, while FGF2 responses are not affected (Supplementary Figure S4).

R#3 Reply: The authors have now improved the *in vivo* analysis by providing quantifications of images that were not previously quantified and by analysing other vascular parameters such as endothelial proliferation, density and sprouting. However, the quantifications/charts provided do not follow the latest Nature communications editorial policy. Specifically they should present dot-plots, instead of just bars, and should have indication of what the dots and error bars represent. For most data shown, the error bars are very small, particularly for the datasets obtained *in vivo*. It is also not clear in most figure legends or in the methods, how the numbers presented in the charts were obtained (i.e. how many animals/retinas or how many pictures

per animal/retina were used for the quantifications shown on the charts).

- Comparison of Sdc2 mutant phenotypes with Vegfr2^{+/-} or Vegfr2 Pdgfb/Cdh5-CreERT2 inducible heterozygous deletion phenotypes. These have a 50% reduction in VEGF signaling and allow for a better phenotypic comparison.

Reply: There is a large body of literature dealing with these phenotypes. Given the number of published studies, it seems hardly reasonable to carry out yet another VEGFR2 het study. The information below has been added to the Discussion (page 13, lines 297-311):

VEGFA concentration is the key limiting factor regulating VEGFR2 signaling. That is, VEGFR2 signaling is ligand-dependent and the amount of ligand available to bind VEGFR2 is far less than the number of VEGFR2 binding sites available. Thus, a deletion of a single VEGFR2 allele DOES NOT lead to a 50% reduction in VEGFA signaling while a single allele deletion of VEGFA is lethal.

- Shalaby et al (Nature, 1996) show no measurable phenotypes during embryonic development in VEGFR2 hets.
- Lars Jakobsson (Gerhardt lab, Nat Cell Biol 2010) showed in competition assays that VEGFR2^{+/-} ECs move slower than VEGFR2^{+/+} ECs. This is really a migration and not a proliferation assay. In agreement with this, we also see decreased migration of Sdc2^{-/-} ECs in response to VEGFA.
- Silvaraj et al (Dev Cell 2013) show 15% reduction in the retinal angiogenesis extent and ~35% reduction in the number of branch points in VEGFR2 hets generated with Pdgfb-iCre (in Suppl Fig 2F, G). These numbers are consistent with our observations.

In summary, 35% reduction in VEGF signaling here is highly biologically meaningful given that 50% reduction is lethal. VEGF signaling is driven by ligand availability. Receptor levels are of secondary importance.

Sdc2 role in facilitating VEGFA-VEGFR2 complex formation is critical as the absence of Sdc2 leads to a 35% reduction in signaling with measurable in vivo biological effects.

R#3 Reply: What is the published evidence for the authors sentence “deletion of a single VEGFR2 allele DOES NOT lead to a 50% reduction in VEGFA signaling” ? The authors must cite and discuss it to validate their arguments/hypothesis. The fact that most VEGFA KO/wt mutants die during embryonic development, whereas most VEGFR2 KO/Wt mutants survive, cannot be exclusively correlated with Vegfr2 signalling dose, for several reasons. The first being that VEGFA can bind to several other canonical and non-canonical VEGF receptors/adaptor molecules. The second, and in agreement with the ligand vs receptor

relative signalling dynamics proposed by the authors, Vegfr2 heterozygous (KO/wt) animals can still have a 50% decrease in Vegfr2 signaling dose, whereas VEGFA heterozygous (KO/Wt) mutants may have an even more pronounced decrease in Vegfr2 signalling, which would explain why the latter have much more pronounced vascular/angiogenesis defects.

Oladipupo et al (2018), Sci Rep., 8, 14724 -> analyse heterozygous mice for VEGFR2 (Vegfr2Cre/+ or Vegfr2LacZ/+ knockin/knockout mice) and observe a decrease above 50% in tumor angiogenesis (Fig1D, Fig2D, Fig3C) and in p-VEGFR2 staining in tumor vessels (Fig3C).

Regarding the citation Shalaby et al., Nature 1996, I believe the authors are referring to Shalaby et al (1995), Nature, 376, 62-66. In this paper, VEGFR2 KO/+ embryos are only compared with VEGFR2 KO/KO littermates and not with wild-type embryos. Therefore it sheds no light onto the phenotype of VEGFR2 het mice. I do not understand the authors point here, except that Vegfr2 hets do not die during development, like Sdc2 full KO animals.

Regarding Jakobsson et al., NCB 2010 paper, once again the authors cite it incorrectly. Jakobsson et al paper state "Cell tracking demonstrated that wild-type cells (expressing DsRed or yellow fluorescent protein; YFP) and Vegfr2^{-/+} egfp cells migrated at roughly similar velocity and with similarly persistent directionality (Fig. 5f, g)." Thus, a true 50% reduction in VEGFR2 protein levels (Fig. S2A of the same cited paper) does not seem to lead to endothelial cell migration defects, as the authors wrongly cite in their text and associate with their Sdc2 mutant phenotypes, with an estimated 35% reduction in Vegfr2 signalling.

Regarding the citation Sivaraj et al (2013), Dev Cell, 25, 427-434 -> they indeed show a 15% reduction in the retinal angiogenesis extent and ~35% reduction in the number of branch points in VEGFR2 hets generated with Pdgfb- iCreERT2 (in Suppl Fig 2F, G). However, special attention has to be made to the fact that the retinas analysed in the cited paper are with inducible deletion of Vegfr2, which was not validated. Zarkada et al (2015), PNAS, 112, 761-766, shows in Figure 1D that the deletion efficiency of VEGFR2 varies from 20% to 80% (Figure 1E-J).

In summary, regardless of the relative importance for signalling of ligand vs receptor dose, VEGFR2 heterozygous mice/cells do have a 50% decrease in VEGFR2 expression/protein, and presumably signalling, which would be the best comparison to justify that a 35% decrease in Vegfr2 signalling in Sdc2 mutants is biologically relevant. Several papers on Vegfr2 signalling, including the ones cited by the authors, show that the authors' following argument is not correct "Receptor levels are of secondary importance".

- Mutate the endogenous Sdc2 gene, to analyze the impact on Vegfr signaling and angiogenesis of the same Sdc2 mutant forms characterized in vitro.

Reply: This is an interesting idea that, we believe, is outside of the scope of this manuscript that deals with the effect of protein sequence on a sulfation pattern. Especially given how long it would take to do these studies and how much this would cost.

We clearly show that reduced VEGFA signaling in Sdc2^{-/-} EC (due to specific loss of Sdc2 HS-chains) is involved in determining vascular phenotype in vivo and phenotypic effects in vitro. This does not exclude that Sdc2 deletion may also have other HS-independent consequences (i.e. due to loss of Sdc2 core protein)

- If possible modulate Sdc2 glycosylation and sulfation in vivo, or its binding to VEGF.

Reply: *Currently, there is not feasible way to selectively modulate glycosylation or sulfation in vivo or in vitro.*

2. The authors have not presented and discussed all existing bibliographic data on the other published work regarding the role of Sdc2 in angiogenesis. Papers from Rossi et al (2014) J. Cell Science 127, 4788 and Noguer et al (2009) Exp Cell Res 315, 795-808 are two such examples. These papers propose opposite or similar roles of Sdc2, to the ones presented in this manuscript. This data has to be carefully contrasted and discussed to reveal what is novel and different in this manuscript.

Reply: *These have been added to Discussion (page 14, lines 310-316):*

R#3 Reply: Bibliographic data that had not been previously presented and discussed, namely Rossi et al (2014) J. Cell Science 127, 4788 and Noguer et al (2009) Exp Cell Res 315, 795-808, has now been introduced in the current manuscript. However, Noguer et al has been incorrectly referred to. The authors cite Noguer et al., saying that the data presented in this manuscript is “in agreement with early reports of in vitro Sdc2 silencing resulting in reduced ECs migration and cord formation in the Matrigel assay^{55, 56}, however Noguer et al shows that downregulation of Sdc2 enhances EC migration (also written in their paper abstract).

Rossi et al., has also shown that shed Sdc2 acts as a negative (not positive) regulator of VEGF signalling by sequestering it. In this context, it is important for the authors to better discuss the contradicting data or try to assess the relative levels of shed versus non-shed Sdc2 in endothelial cells. How much of the endothelial Sdc2 exists at the membrane to form a complex with VEGFR2-VEGFA165, versus the one that is shed and may be inhibitory ? There should exist feasible ways of quantifying the different forms of Sdc2 produced by ECs and characterize their distinct activities on signalling.

3. The authors present that Sdc4 is important in FGF-dependent signal transduction however no significant effect is obtained in the cornea pocket assay with FGF2 pellets. Surprisingly, the authors claim that “Sdc4^{-/-}, but not Sdc2iCdh5 showed a reduction in angiogenesis in response to FGF2 (Fig. 1g-h)”. The indicated figures show Sdc4 mutants present no difference in both VEGF- and FGF2-induced angiogenesis when compared to control mice.

Reply: *We apologize for the writing error. Indeed, we intended to write that both VEGFA- and*

FGF2 -induced angiogenesis in cornea-pocket assay WAS NOT affected in Sdc4-/- mice.

Sdc4 controls other biological aspects related to FGF-signaling, particularly mTOR/AKT/eNOS pathway which modulate systemic blood pressure.

4. Analysis of the EC-specific Sdc2 LOF leads to delayed vessel outgrowth. Retinas in Fig1A appear to have different sizes, ie., the Sdc2 mutant whole retina tissue seems smaller. Instead of presenting the absolute values of vascular outgrowth, it would be better to normalize the vascular outgrowth to the total size of the retina. The control and mutants do not seem stage matched.

Reply: *The quantification of retina outgrowth measures the distance between the center and vascular front. Difference size of retinas in the avascular parts are due technical variability during eye opening and flattening for retinal staining. We have not observed significant differences between total size of retinas across genotypes.*

Additionally, for reader's confidence, the vascular progression of each strain is now shown as normalized to Sdc2^{fl/fl} (new quantification in Fig. 1B).

R#3 Reply: It is precisely because of technical variability that the retinal outgrowth should be measured as a ratio that reflects the amount of retina that is vascularized. Thus retina outgrowth should be VD/PD, being VD = distance between the center and vascular front and PD = distance between the center and the periphery of the retina. This should be addressed appropriately, as Reviewer 1 expressed similar concerns.

Minor Concerns:

1. Authors show by immunostaining that Sdc2 is expressed in ECs of the aorta and in the retina, where it seems that it is mostly enriched in arteries and arterioles. Images representative of all regions of the retina (vein and venous plexus, angiogenic front and sprouts) are necessary.

Reply: *Additional staining of retinas showing Sdc2 expression have been added (Supplementary figure 1e). Syndecan-2 is expressed in arteries, veins and capillaries. It may be true that Sdc2 is enriched in arterial versus venous EC, however the fact that it is expressed in both SMCs and ECs does not allow us to conclude this.*

R#3 Reply: The authors now present data on the expression of Sdc2 in veins and capillaries, in addition to retinal arteries. Unfortunately, the authors have not provided any data on the expression of Sdc2 at the angiogenic front (where sprouting and proliferating cells locate), which would be of interest as they propose that the observed phenotype is related with the

abrogation of VEGF-VEGFR2-Sdc2 interaction that is needed for optimal VEGFR2 signalling, and EC migration and proliferation. Given that VEGF levels are highest at the angiogenic front/leading edge of the retina, this is where one would expect a more relevant role of Sdc2 (and perhaps expression). Ideally, an image of a whole retinal flank with immunostaining for Sdc2 and higher magnification insets of the different regions should be provided.

2. None of the images presented in the manuscript have scale bars.

Reply: *scale bars have been added*

R#3 Reply: images S1D;E and S2A, D still do not have scale bars.

3. No data has been presented regarding the endothelial expression of Sdc4.

Reply: *There are numerous publications demonstrating high Sdc4 expression in ECs, including in the retinal vasculature. This information, along with citations to most recent papers, has been added to the text (page 6, lines 132-133).*

4. Authors do not attribute the decrease in vessel density to enhanced vascular pruning in the Sdc2 mutants because “collagen IV staining did not detect empty sleeves”. Since collagen empty sleeves are indeed detectable in Fig S2A, we infer that the authors meant that there were no differences in the frequency of empty sleeves between control and mutant retinas. However, no quantification data is presented.

Reply: *Quantification has been added (Supplementary Figure 3B)*

5. Quantifications on the density of vascular networks in the skin and kidney are not presented.

Reply: *We have decided to remove these data and to focus on retina vascular development and hindlimb ischemia, two settings in which relative contribution of VEGFA versus other factors is extremely well characterized.*

6. Although the authors claim the role of Sdc2 on angiogenesis is mediated through the VEGFa165-VEGFR2 axis, there is also evidence that Sdc2 also interacts with integrins (Exp. Cell. Res (2000), Vol.256, 434 and Biophysics Res Commun (2009), Vol385, 231), which is important for cell migration and adhesion. In fact, there are several papers that show VEGFA-mediated functions dependent on integrins (Blood 2012 120:4892-4902; J Biol Chem. 2011 286:1083/1092; JBC 2007 282, 15187-15196; Embo J 1999;18:882–892; Mol Cell 2000;6:851–860; Circ Res 2007;101:570–580). To assess integrin signaling, immunostaining for p-FAK or active-integrinB1 could be used.

Reply: We agree that integrins are important in VEGFr2 signaling. How that integrates with Sdc2 is less clear and is the subject for further studies. Our data clearly show that Sdc2 is essential for formation of VEGFA165-VEGFR2-Sdc2 complex which, in turn, is required for maximal VEGFR2 phosphorylation. Whether an integrin, such as beta-2 is involved is not certain. Integrins activation is one of the VEGFR2 downstream target and likely to be affected as the other downstream targets. This possibility and relative references are added to discussion as suggested. (pages 14, lines 317-323).

R#3 Reply: AS mentioned in the answer to major concern 1, since the decrease in endothelial Vegfr2 signalling *in vivo* is still not clear, and as the authors also mention in the discussion (“we have not investigated whether Sdc2 deletion differentially affects various VEGFR2 downstream signaling pathways (e.g. PLC γ /ERK, AKT/Src, integrin activation, etc....”), it should be analysed how ERK/AKT/Src/Integrin signalling pathways are affected in Sdc2 mutants, to provide a better mechanistic explanation for the defects observed in EC migration/sprouting/proliferation.

7. In vivo validation of the in vitro co-IP data would also be interesting, using for example the tissues from animals.

Reply: In the proteoglycans world, IP and WB experiments are severely limited by lack of working antibodies against endogenous proteins. Sdc2 is no exceptions. This is the reason why most studies in this field rely on the use of epitope-tags and overexpression systems. There is no possibility of getting a Sdc2-specific IP from a tissue lysate.

8. It would be important to check by western blot if Sdc4 is regulated in the absence of Sdc2 and vice-versa.

Reply: As requested, we now show Sdc4 blot in Figure 3A (Sdc2^{-/-} ECs) and Sdc2 blot in Figure 3D (Sdc4^{-/-} ECs). We did not detect any evident upregulation.

9. In Fig3H and Fig 4G, it appears that Sdc2 and Sdc4 have opposite effects on p-VEGFR2 (comparing to cells transduced with only Ad-GFP). Could the authors comment on this?

Reply: The only statistically significant effect we see in these studies is when Sdc2 is reintroduced into Sdc2^{-/-} ECs.

10. All interaction assays were performed in cells transduced with adenovirus and expressing different mutated forms of Sdc2/4. It is known that over-expression of proteins can sometimes lead to unspecific protein interactions. How do the total levels of these proteins compare to the

endogenous levels? Can the same experiments be performed with CRISPR/Cas9 induced mutations in Sdc2?

Reply: We agree this is an important point. IP experiments following adenovirus transduction (at MOI = ~ 1-2) were performed with minimal overexpression which was similar to syndecans endogenous level (supplementary figures 6A-B)

CRISPR/Cas9 mutagenesis would be a whole new study and approach would still be limited by lack of good antibody against endogenous protein without epitope-tag.

11. Although the protein sequence of the D1 domains of Sdc2 are well conserved among species, I wonder whether the glycosylation events are also well conserved, since this is known to vary depending on cell context. It would be important to confirm that glycosylation pattern in mouse endothelial cells are identical to those described in the manuscript and ideally in the absence of overexpression of the proteins.

Reply: That would be a good study to do. Unfortunately, high-quality antibody for affinity purification of endogenous syndecans do not exist. Nevertheless, the point of testing a different species is well taken and we have repeated HS chain analysis of both sdc2 and sdc4 isolated from a mouse endothelial cell line. We now show that also mouse ECs generate higher level of 6-O-sulfation in Sdc2 HS chains than in Sdc4 HS chains (Supplementary Figure 6E). This is similar to what we observed in HUVECs

Overview of manuscript changes:

- Quantification of vascular progression in retina has been corrected (Fig 1B, and lines 427-248 in method section)
- ERK activation (previous Fig.3A,C) merged with other VEGFR2 downstream effectors (Supp. Fig. 5A-B)
- Vascular parameters in retinas are now reported as dot-blot (Supp. Fig 3C-G). Number animal/retinas added to legends
- Effect of Sdc2 silencing in HUVEC (previous Supp. Fig. 5A-C) moved to Supp. Fig 4G-I
- Analysis of VEGFA target genes expression in retina ECs has been added (Supp. Fig. 5E)
- Sdc2 blot in Supp. Fig. 5F showed a wrong image (last lane indicate "Sdc2^{-/-} + Ad-Sdc4 " but it was showing expression of Sdc2 in wild-type cells). A representative picture that match legend is now shown.
- Primer sequences are now indicated in a separate table (Table 2)

Reviewers' comments:

Reviewer #1 (Remarks to the Author):

The authors have satisfactorily answered the questions from the prior review in part by correcting misconceptions by clarifying their writing, and by performing additional experimentation. The original claims remain true and are compelling. The most important finding in this work is that a determinant in the core protein of syndecan-2 potentially regulates the sulfation pattern of the heparan sulfate chains synthesized on the protein. The second important finding is that sulfation pattern is not only different, but it supports a syndecan-2-specific function that is not shared with its closest homologue, syndecan-4, nor apparently with syndecans 1 and 3 which are less homologous. This is a "first of its kind" demonstration that could well be paradigm changing.

The work appears to be statistically significant and is conducted in such a manner, e.g., supplementing the animal models with cultured cells and comparing Sdc2 versus Sdc4, that the findings are convincingly reproducible.

We thank the reviewer for recognizing the importance of this work and for comments/suggestions that have greatly improved the quality of this work

Minor point:

L 155-160 The authors describe a comparison between Sdc2^{-/-} and Sdc4^{-/-} endothelial cells, yet the actual comparison as shown in Supplementary Figure 4a-f appears to be between wild-type and Sdc2^{-/-}. The data as shown are fine, but the text needs to be corrected to fit the figure.

Reply: Yes, we meant wild-type. This has been corrected

Reviewer: Alan C. Rapraeger

Reviewer #2 (Remarks to the Author):

The extent to which different proteoglycans expressed by the same cell differ in HS chain structure has been a long simmering controversy. The authors have performed careful and convincing studies that show differences in 6-O-sulfation for syndecan-2 versus syndecan-4 that cause differences in VEGFA-driven angiogenic processes. This work is significant in correlating differences in HS structure with angiogenic function.

Thank you for the comments and recognizing the importance of this work!

Line 542, define HA in the text. The reader may confuse the HA epitope from hemagglutinin with the abbreviation commonly used for hyaluronan.

HA has been spelled out (line 197)

Reviewer #3 - expert in angiogenesis (Remarks to the Author):

In the manuscript by Federico et al, the authors study the role of Sdc-2 in angiogenesis using the retinal and hindlimb ischemia models, concluding that it impairs VEGFA-165-driven angiogenesis. Mechanistically, and resorting exclusively to in vitro studies, the authors propose that the increased levels of 6-O sulfation of Sdc2 HS chains confer it the unique ability to form a ternary complex Sdc2-VEGFA165-VEGFR2, which is proposed to enhance VEGFR2 signaling. The paper is Ok when it comes to demonstrating how different biochemical modifications may change the modulation of VEGFR2 signaling by Sdc2 in vitro. However, the in vivo analysis performed was very superficial, poorly quantified and there is no clear evidence that the molecular mechanism identified in vitro is at the source of the observed phenotype in vivo. The manuscript needs a significant amount of new in vivo data to further validate the in vitro findings. In addition, the weak regulation of Vegfr2 signaling identified in Sdc2KO cells (-35% signaling found in ECs in vitro) and the very mild defects seen in Sdc2Pdgfb mutant retinas, shows that the regulation is not of enough biological significance.

Reply: We thank the reviewer suggestions to improve the manuscript. Importantly, we now show strong **in vivo evidence** that the observed 35% reduction in VEGFR2 signaling activation has biological relevance and it is involved in determining the vascular phenotype shown in Sdc2 endothelial- specific KO.

R#3 Reply: The authors still do not provide clear evidence that VEGFR2 signalling is decreased in endothelial cells in vivo (see below), and how the estimated 35% reduction in VEGFR2

signalling causes the observed *in vivo* vascular/angiogenesis phenotype, which is one of the most important aspects of this work.

Reply 2: As suggested, we isolated retinal ECs from WT and *Sdc2*ECKO mice retinas and measured expression of multiple VEGFA/VEGFR2-induced genes in freshly-sorted cells (no *in vitro* culture). These data clearly indicate reduced VEGFR2 signaling also *in vivo*. (Fig. S5E, lines 176-186, see below also)

Taken together with the data already in the manuscript that show angiogenic, VEGF-A-dependent defects in 3 different highly VEGFA-dependent models in mice with endothelial specific *Sdc2* deletion (i.e. retinal angiogenesis, VEGFA-induced corneal angiogenesis, and hind limb ischemia), these data clearly establish reduced VEGF/VEGFR2 signaling *in vivo*.

Major concerns:

1. The current manuscript requires substantial new *in vivo* data to validate the *in vitro* results:
 - Analysis in *Sdc2* mutants of Vegfr2 signalling levels and downstream targets.

Reply: Similar to our *in vitro* data, we now show significantly reduced VEGFR2 activation in whole-retinal lysate (P6) after Tamoxifen-induced *Sdc2* endothelial deletion (P1 to P5) (Supplementary Figure 5D-E)

R#3 Reply: Unfortunately, the authors' choice to assess VEGFR2 signaling in lysates of whole postnatal retinas was not the best. Okabe et al (2014) Cell, 159, 584-96 and other papers have shown that in the retina, VEGFR2 is expressed in neurons at levels that are much higher than those present in ECs. Moreover, ECs are only a very minor fraction (less than 1%) of the entire postnatal retina tissue. Thus, western blot of whole retinal lysates cannot be used to evaluate VEGFR2 activity in endothelial cells. Authors should have instead performed immunostaining to profile the protein or phosphorylation levels of Vegfr2 or several published canonical downstream targets (ERK, Esm1, Vegfr3, etc...). The authors can also FACS sort ECs from different tissues of their mutants, and perform Western/qRT-PCR analysis for Vegfr2 signalling or downstream targets.

Reply 2: While it is true that VEGFR2 is expressed in non-ECs in the retina as indicated by the reviewer, our data show a significant reduction in VEGFR2 phosphorylation in *Sdc2*ECKO retinas. Since there is no reason to think that an endothelial *Sdc2* KO would affect VEGFR2 phosphorylation in neuronal cells, these data are conclusive.

The suggested experiment by the reviewer- to assess VEGFR2 phosphorylation in the retina is not feasible as we are not aware of any commercial antibodies that reliably detect VEGFR2 phosphorylation on tissue sections.

Nevertheless, we provide a new set of data to further settle this issue: we sorted retinal ECs from Sdc2^{ECKO} and WT mice and used qPCR to measure expression of a number known downstream target of the VEGFA-PLC γ /IP3 pathway (Fig. S5E, lines 176-186). These genes are strongly induced by VEGFA and thus allow for best sensitivity to detect expression changes. All of them show reduced expression in Sdc2^{ECKO} mice endothelial cells.

In the discussion line 318, the authors mention “we have not investigated whether Sdc2 deletion differentially affects various VEGFR2 downstream signaling pathways (e.g. PLC γ /ERK, AKT/Src, integrin activation, etc....” . This reviewer believes it is of high relevance to do this, in order to support the paper’s main message.

Reply 2: *Analysis of all major VEGFA-VEGFR2 downstream effectors (ERK, AKT, SRC, Integrin β 3) have been added (Lines 169-172, Fig. S5A-B). All show reduced signaling as would be expected.*

Moreover, Okabe et al (2014) Cell, 159, 584-96 have also shown that a reduction in neuronal VEGFR2 signalling alone leads to alterations in retinal angiogenesis, with retinas displaying “delays in the radial outgrowth and decreased vascular density in the superficial plexus when compared to control mice”. Given the authors whole retina lysate western blot results, do the authors conclude that VEGFR2 signalling is also impaired in the neurons of their Sdc2 EC-specific mutants? The authors should comment/address this issue.

Reply 2: *We now show a reduced expression of multiple VEGFR2 downstream genes in freshly-sorted ECs from Sdc2^{ECKO} retinas (Fig. S5E, lines 176-186). This show that VEGFR2 signaling is directly affected in ECs.*

- Analysis of other vascular parameters in Sdc2 mutants, such as endothelial proliferation, density and sprouting.

Reply: *Characterization of the retinal vascular phenotype in Sdc2 ECKO mice has been expanded. We now show reduced number of tip cells (Supplementary figure 3C-D). Reductions in vascular density and proliferation are also observed in Sdc2 ECKO (supp. Figure 3E-F) Additionally, we have performed analysis of VEGFA-induced biological effects in vitro. We show that Sdc2^{-/-} ECs have reduced VEGFA-induced proliferation and migration, while FGF2 responses*

are not affected (Supplementary Figure S4).

R#3 Reply: The authors have now improved the *in vivo* analysis by providing quantifications of images that were not previously quantified and by analysing other vascular parameters such as endothelial proliferation, density and sprouting. However, the quantifications/charts provided do not follow the latest Nature communications editorial policy. Specifically they should present dot-plots, instead of just bars, and should have indication of what the dots and error bars represent. For most data shown, the error bars are very small, particularly for the datasets obtained *in vivo*. It is also not clear in most figure legends or in the methods, how the numbers presented in the charts were obtained (i.e. how many animals/retinas or how many pictures per animal/retina were used for the quantifications shown on the charts).

Reply 2: These data are now presented as dot-blots (Fig. S3D, F, G) and animal/retinas are clarified in supplementary legend 3

- Comparison of *Sdc2* mutant phenotypes with *Vegfr2*^{+/-} or *Vegfr2* *Pdgfb/Cdh5-CreERT2* inducible heterozygous deletion phenotypes. These have a 50% reduction in VEGF signaling and allow for a better phenotypic comparison.

Reply: There is a large body of literature dealing with these phenotypes. Given the number of published studies, it seems hardly reasonable to carry out yet another VEGFR2 het study. The information below has been added to the Discussion (page 13, lines 297-311):

VEGFA concentration is the key limiting factor regulating VEGFR2 signaling. That is, VEGFR2 signaling is ligand-dependent and the amount of ligand available to bind VEGFR2 is far less than the number of VEGFR2 binding sites available. Thus, a deletion of a single VEGFR2 allele DOES NOT lead to a 50% reduction in VEGFA signaling while a single allele deletion of VEGFA is lethal.

- Shalaby et al (Nature, 1996) show no measurable phenotypes during embryonic development in VEGFR2 hets.

- Lars Jakobsson (Gerhardt lab, Nat Cell Biol 2010) showed in competition assays that VEGFR2^{+/-} ECs move slower than VEGFR2^{+/+} ECs. This is really a migration and not a proliferation assay. In agreement with this, we also see decreased migration of *Sdc2*^{-/-} ECs in response to VEGFA.

- Silvaraj et al (Dev Cell 2013) show 15% reduction in the retinal angiogenesis extent and ~35% reduction in the number of branch points in VEGFR2 hets generated with *Pdgfb*Cre (in Suppl Fig 2F, G). These numbers are consistent with our observations.

In summary, 35% reduction in VEGF signaling here is highly biologically meaningful given that 50% reduction is lethal. VEGF signaling is driven by ligand availability. Receptor levels are of secondary importance.

Sdc2 role in facilitating VEGFA-VEGFR2 complex formation is critical as the absence of Sdc2 leads to a 35% reduction in signaling with measurable in vivo biological effects.

R#3 Reply: What is the published evidence for the authors sentence “deletion of a single VEGFR2 allele DOES NOT lead to a 50% reduction in VEGFA signaling” ? The authors must cite and discuss it to validate their arguments/hypothesis. The fact that most VEGFA KO/wt mutants die during embryonic development, whereas most VEGFR2 KO/Wt mutants survive, cannot be exclusively correlated with Vegfr2 signalling dose, for several reasons. The first being that VEGFA can bind to several other canonical and non-canonical VEGF receptors/adaptor molecules. The second, and in agreement with the ligand vs receptor relative signalling dynamics proposed by the authors, Vegfr2 heterozygous (KO/wt) animals can still have a 50% decrease in Vegfr2 signaling dose, whereas VEGFA heterozygous (KO/Wt) mutants may have an even more pronounced decrease in Vegfr2 signalling, which would explain why the latter have much more pronounced vascular/angiogenesis defects.

Oladipupo et al (2018), Sci Rep., 8, 14724 -> analyse heterozygous mice for VEGFR2 (Vegfr2Cre/+ or Vegfr2LacZ/+ knockin/knockout mice) and observe a decrease above 50% in tumor angiogenesis (Fig1D, Fig2D, Fig3C) and in p-VEGFR2 staining in tumor vessels (Fig3C). Regarding the citation Shalaby et al., Nature 1996, I believe the authors are referring to Shalaby et al (1995), Nature, 376, 62-66. In this paper, VEGFR2 KO/+ embryos are only compared with VEGFR2 KO/KO littermates and not with wild-type embryos. Therefore it sheds no light onto the phenotype of VEGFR2 het mice. I do not understand the authors point here, except that Vegfr2 hets do not die during development, like Sdc2 full KO animals.

Regarding Jakobsson et al., NCB 2010 paper, once again the authors cite it incorrectly. Jakobsson et al paper state “Cell tracking demonstrated that wild-type cells (expressing DsRed or yellow fluorescent protein; YFP) and *Vegfr2*^{-/+}*egfp* cells migrated at roughly similar velocity and with similarly persistent directionality (Fig. 5f, g).” Thus, a true 50% reduction in VEGFR2 protein levels (Fig. S2A of the same cited paper) does not seem to lead to endothelial cell migration defects, as the authors wrongly cite in their text and associate with their Sdc2 mutant phenotypes, with an estimated 35% reduction in Vegfr2 signalling.

Regarding the citation Sivaraj et al (2013), Dev Cell, 25, 427-434 -> they indeed show a 15%

reduction in the retinal angiogenesis extent and ~35% reduction in the number of branch points in VEGFR2 hets generated with Pdgfb- iCreERT2 (in Suppl Fig 2F, G). However, special attention has to be made to the fact that the retinas analysed in the cited paper are with inducible deletion of Vegfr2, which was not validated. Zarkada et al (2015), PNAS, 112, 761-766, shows in Figure 1D that the deletion efficiency of VEGFR2 varies from 20% to 80% (Figure 1E-J).

In summary, regardless of the relative importance for signalling of ligand vs receptor dose, VEGFR2 heterozygous mice/cells do have a 50% decrease in VEGFR2 expression/protein, and presumably signalling, which would be the best comparison to justify that a 35% decrease in Vegfr2 signalling in Sdc2 mutants is biologically relevant. Several papers on Vegfr2 signalling, including the ones cited by the authors, show that the authors' following argument is not correct "Receptor levels are of secondary importance".

Reply 2: *While this is an interesting topic for debate, it seems to have little relevance to the study that examines the effect of amino acid sequence on HS sulfation pattern. We should be beyond a shadow of doubt that VEGF-A binding is decreased in vitro and VEGF-dependent process are impaired in vivo. Whether the extent of VEGF signaling reduction in these mice is similar or not to VEGFR2 hets is simply not relevant to this study.*

- Mutate the endogenous Sdc2 gene, to analyze the impact on Vegfr signaling and angiogenesis of the same Sdc2 mutant forms characterized in vitro.

Reply: *This is an interesting idea that, we believe, is outside of the scope of this manuscript that deals with the effect of protein sequence on a sulfation pattern. Especially given how long it would take to do these studies and how much this would cost.*

We clearly show that reduced VEGFA signaling in Sdc2^{-/-} EC (due to specific loss of Sdc2 HS chains) is involved in determining vascular phenotype in vivo and phenotypic effects in vitro. This does not exclude that Sdc2 deletion may also have other HS-independent consequences (i.e. due to loss of Sdc2 core protein)

- If possible modulate Sdc2 glycosylation and sulfation in vivo, or its binding to VEGF.

Reply: *Currently, there is not feasible way to selectively modulate glycosylation or sulfation in vivo or in vitro.*

2. The authors have not presented and discussed all existing bibliographic data on the other

published work regarding the role of Sdc2 in angiogenesis. Papers from Rossi et al (2014) J. Cell Science 127, 4788 and Noguer et al (2009) Exp Cell Res 315, 795-808 are two such examples. These papers propose opposite or similar roles of Sdc2, to the ones presented in this manuscript. This data has to be carefully contrasted and discussed to reveal what is novel and different in this manuscript.

Reply: These have been added to Discussion (page 14, lines 310-316):

R#3 Reply: Bibliographic data that had not been previously presented and discussed, namely Rossi et al (2014) J. Cell Science 127, 4788 and Noguer et al (2009) Exp Cell Res 315, 795-808, has now been introduced in the current manuscript. However, Noguer et al has been incorrectly referred to. The authors cite Noguer et al., saying that the data presented in this manuscript is “in agreement with early reports of in vitro Sdc2 silencing resulting in reduced ECs migration and cord formation in the Matrigel assay^{55, 56}, however Noguer et al shows that downregulation of Sdc2 enhances EC migration (also written in their paper abstract). Rossi et al., has also shown that shed Sdc2 acts as a negative (not positive) regulator of VEGF signalling by sequestering it. In this context, it is important for the authors to better discuss the contradicting data or try to assess the relative levels of shed versus non-shed Sdc2 in endothelial cells. How much of the endothelial Sdc2 exists at the membrane to form a complex with VEGFR2-VEGFA165, versus the one that is shed and may be inhibitory? There should exist feasible ways of quantifying the different forms of Sdc2 produced by ECs and characterize their distinct activities on signalling.

Reply 2: *More extensive discussion of previous data has been added (lines 326-338).*

3. The authors present that Sdc4 is important in FGF-dependent signal transduction however no significant effect is obtained in the cornea pocket assay with FGF2 pellets. Surprisingly, the authors claim that “Sdc4^{-/-}, but not Sdc2iCdh5 showed a reduction in angiogenesis in response to FGF2 (Fig. 1g-h)”. The indicated figures show Sdc4 mutants present no difference in both VEGF- and FGF2-induced angiogenesis when compared to control mice.

Reply: *We apologize for the writing error. Indeed, we intended to write that both VEGFA- and FGF2 -induced angiogenesis in cornea-pocket assay WAS NOT affected in Sdc4^{-/-} mice. Sdc4 controls other biological aspects related to FGF-signaling, particularly mTOR/AKT/eNOS pathway which modulate systemic blood pressure.*

4. Analysis of the EC-specific Sdc2 LOF leads to delayed vessel outgrowth. Retinas in Fig1A appear to have different sizes, ie., the Sdc2 mutant whole retina tissue seems smaller. Instead

of presenting the absolute values of vascular outgrowth, it would be better to normalize the vascular outgrowth to the total size of the retina. The control and mutants do not seem stage matched.

Reply: *The quantification of retina outgrowth measures the distance between the center and vascular front. Difference size of retinas in the avascular parts are due technical variability during eye opening and flattening for retinal staining. We have not observed significant differences between total size of retinas across genotypes.*

Additionally, for reader's confidence, the vascular progression of each strain is now shown as normalized to $Sdc2_{fl/fl}$ (new quantification in Fig. 1B).

R#3 Reply: **It is precisely because of technical variability that the retinal outgrowth should be measured as a ratio that reflects the amount of retina that is vascularized. Thus retina outgrowth should be VD/PD, being VD = distance between the center and vascular front and PD = distance between the center and the periphery of the retina. This should be addressed appropriately, as Reviewer 1 expressed similar concerns.**

Reply 2: *Quantification has been changed as suggested by the reviewer (Fig 1B, and lines 423-424 in method section)*

Minor Concerns:

1. Authors show by immunostaining that Sdc2 is expressed in ECs of the aorta and in the retina, where it seems that it is mostly enriched in arteries and arterioles. Images representative of all regions of the retina (vein and venous plexus, angiogenic front and sprouts) are necessary.

Reply: *Additional staining of retinas showing Sdc2 expression have been added (Supplementary figure 1e). Syndecan-2 is expressed in arteries, veins and capillaries. It may be true that Sdc2 is enriched in arterial versus venous EC, however the fact that it is expressed in both SMCs and ECs does not allow us to conclude this.*

R#3 Reply: **The authors now present data on the expression of Sdc2 in veins and capillaries, in addition to retinal arteries. Unfortunately, the authors have not provided any data on the expression of Sdc2 at the angiogenic front (where sprouting and proliferating cells locate), which would be of interest as they propose that the observed phenotype is related with the abrogation of VEGF-VEGFR2-Sdc2 interaction that is needed for optimal VEGFR2 signalling, and EC migration and proliferation. Given that VEGF levels are highest at the angiogenic**

front/leading edge of the retina, this is where one would expect a more relevant role of Sdc2 (and perhaps expression). Ideally, an image of a whole retinal flank with immunostaining for Sdc2 and higher magnification insets of the different regions should be provided.

REPLY 2: We have added expression of Sdc2 in tip cells (Fig. S1E-lower left inset). Unfortunately, quality of all available Sdc2 antibodies is poor (not surprisingly) and we cannot have the highest staining quality. We are developing a custom-made antibody against mouse Sdc2 that should permit better staining and WB/IP analysis in our future works.

2. None of the images presented in the manuscript have scale bars.

Reply: scale bars have been added

R#3 Reply: images S1D; E and S2A, D still do not have scale bars.

Reply 2: Scale bars have been added.

3. No data has been presented regarding the endothelial expression of Sdc4.

Reply: There are numerous publications demonstrating high Sdc4 expression in ECs, including in the retinal vasculature. This information, along with citations to most recent papers, has been added to the text (page 6, lines 132-133).

4. Authors do not attribute the decrease in vessel density to enhanced vascular pruning in the Sdc2 mutants because “collagen IV staining did not detect empty sleeves”. Since collagen empty sleeves are indeed detectable in Fig S2A, we infer that the authors meant that there were no differences in the frequency of empty sleeves between control and mutant retinas. However, no quantification data is presented.

Reply: Quantification has been added (Supplementary Figure 3B)

5. Quantifications on the density of vascular networks in the skin and kidney are not presented.

Reply: We have decided to remove these data and to focus on retina vascular development and hindlimb ischemia, two settings in which relative contribution of VEGFA versus other factors is extremely well characterized.

6. Although the authors claim the role of Sdc2 on angiogenesis is mediated through the VEGFa165-VEGFR2 axis, there is also evidence that Sdc2 also interacts with integrins (Exp. Cell. Res (2000), Vol.256, 434 and Biophysics Res Commun (2009), Vol385, 231), which is important for cell migration and adhesion. In fact, there are several papers that show VEGFA-mediated functions dependent on integrins (Blood 2012 120:4892-4902; J Biol Chem. 2011 286: 1083/1092; JBC 2007 282, 15187-15196; Embo J 1999;18:882–892; Mol Cell 2000;6:851–860; Circ Res 2007;101:570–580). To assess integrin signaling, immunostaining for p-FAk or active-integrinB1 could be used.

Reply: We agree that integrins are important in VEGFr2 signaling. How that integrates with Sdc2 is less clear and is the subject for further studies. Our data clearly show that Sdc2 is essential for formation of VEGFA165-VEGFR2-Sdc2 complex which, in turn, is required for maximal VEGFR2 phosphorylation. Whether an integrin, such as beta-2 is involved is not certain. Integrins activation is one of the VEGFR2 downstream target and likely to be affected as the other downstream targets. This possibility and relative references are added to discussion as suggested. (pages 14, lines 317-323).

R#3 Reply: AS mentioned in the answer to major concern 1, since the decrease in endothelial Vegfr2 signaling *in vivo* is still not clear, and as the authors also mention in the discussion (“we have not investigated whether Sdc2 deletion differentially affects various VEGFR2 downstream signaling pathways (e.g. PLCγ/ERK, AKT/Src, integrin activation, etc....”), it should be analyzed how ERK/AKT/Src/Integrin signaling pathways are affected in Sdc2 mutants, to provide a better mechanistic explanation for the defects observed in EC migration/sprouting/proliferation.

Reply: Analysis of all major VEGFA-VEGFR2 downstream effectors (ERK, AKT, SRC, Integrin-β3) have been added (Lines 169-172, Fig. S5A-B).

7. In vivo validation of the in vitro co-IP data would also be interesting, using for example the tissues from animals.

Reply: In the proteoglycans world, IP and WB experiments are severely limited by lack of working antibodies against endogenous proteins. Sdc2 is no exceptions. This is the reason why most studies in this field rely on the use of epitope-tags and overexpression systems. There is no possibility of getting a Sdc2-specific IP from a tissue lysate.

8. It would be important to check by western blot if Sdc4 is regulated in the absence of Sdc2 and vice-versa.

Reply: As requested, we now show Sdc4 blot in Figure 3A (Sdc2^{-/-} ECs) and Sdc2 blot in Figure 3D (Sdc4^{-/-} ECs). We did not detect any evident upregulation.

9. In Fig3H and Fig 4G, it appears that Sdc2 and Sdc4 have opposite effects on p-VEGFR2 (comparing to cells transduced with only Ad-GFP). Could the authors comment on this?

Reply: The only statistically significant effect we see in these studies is when Sdc2 is reintroduced into Sdc2^{-/-} ECs.

10. All interaction assays were performed in cells transduced with adenovirus and expressing different mutated forms of Sdc2/4. It is known that over-expression of proteins can sometimes lead to unspecific protein interactions. How do the total levels of these proteins compare to the endogenous levels? Can the same experiments be performed with CRISPR/Cas9 induced mutations in Sdc2?

Reply: We agree this is an important point. IP experiments following adenovirus transduction (at MOI = ~ 1-2) were performed with minimal overexpression which was similar to syndecans endogenous level (supplementary figures 6A-B) CRISPR/Cas9 mutagenesis would be a whole new study and approach would still be limited by lack of good antibody against endogenous protein without epitope-tag.

11. Although the protein sequence of the D1 domains of Sdc2 are well conserved among species, I wonder whether the glycosylation events are also well conserved, since this is known to vary depending on cell context. It would be important to confirm that glycosylation pattern in mouse endothelial cells are identical to those described in the manuscript and ideally in the absence of overexpression of the proteins.

Reply: That would be a good study to do. Unfortunately, high-quality antibody for affinity purification of endogenous syndecans do not exist. Nevertheless, the point of testing a different species is well taken and we have repeated HS chain analysis of both sdc2 and sdc4 isolated from a mouse endothelial cell line. We now show that also mouse ECs generate higher level of 6-O-sulfation in Sdc2 HS chains than in Sdc4 HS chains (Supplementary Figure 6E). This is similar to what we observed in HUVECs

REVIEWERS' COMMENTS:

Reviewer #3 (Remarks to the Author):

Overview of manuscript changes:

- Quantification of vascular progression in retina has been corrected (Fig 1B, and lines 427-248 in method section)
- ERK activation (previous Fig.3A,C) merged with other VEGFR2 downstream effectors (Supp. Fig. 5A-B)
- Vascular parameters in retinas are now reported as dot-blot (Supp. Fig 3C-G). Number animal/retinas added to legends
- Effect of Sdc2 silencing in HUVEC (previous Supp. Fig. 5A-C) moved to Supp. Fig 4G-I
- Analysis of VEGFA target genes expression in retina ECs has been added (Supp. Fig. 5E)
- Sdc2 blot in Supp. Fig. 5F showed a wrong image (last lane indicate "Sdc2-/- + Ad-Sdc4 " but it was showing expression of Sdc2 in wild-type cells). A representative picture that match legend is now shown.
- Primer sequences are now indicated in a separate table (Table 2)

Reviewers' comments:

R#3 Reply 2: General Remarks: The authors have done most of the requested experiments and changes. They also corrected some of the previously wrong citations. The in vivo angiogenesis analysis could still be improved further. However, the paper is stronger on in vitro Sdc/glycobiology and VEGFR signalling studies, some of which are difficult to be fully validated by in vivo experiments.

Reviewer #3 - expert in angiogenesis (Remarks to the Author):

In the manuscript by Federico et al, the authors study the role of Sdc-2 in angiogenesis using the retinal and hindlimb ischemia models, concluding that it impairs VEGFA-165-driven angiogenesis. Mechanistically, and resorting exclusively to in vitro studies, the authors propose that the increased levels of 6-O sulfation of Sdc2 HS chains confer it the unique ability to form a ternary complex Sdc2-VEGFA165-VEGFR2, which is proposed to enhance VEGFR2 signaling. The paper is Ok when it comes to demonstrating how different biochemical modifications may change the modulation of VEGFR2 signaling by Sdc2 in vitro. However, the in vivo analysis performed was very superficial, poorly quantified and there is no clear evidence that the molecular mechanism identified in vitro is at the source of the observed phenotype in vivo. The manuscript needs a significant amount of new in vivo data to further validate the in vitro findings. In addition, the weak regulation of Vegfr2 signaling identified in Sdc2KO cells (-35% signaling found in ECs in vitro) and the very mild defects seen in Sdc2Pdgfb mutant retinas, shows that the regulation is not of enough biological significance.

Reply: We thank the reviewer suggestions to improve the manuscript. Importantly, we now show strong in vivo evidence that the observed 35% reduction in VEGFR2 signaling activation has biological relevance and it is involved in determining the vascular phenotype shown in Sdc2 endothelial- specific KO.

R#3 Reply: The authors still do not provide clear evidence that VEGFR2 signalling is decreased in endothelial cells in vivo (see below), and how the estimated 35% reduction in VEGFR2 signalling causes the observed in vivo vascular/angiogenesis phenotype, which is one of the most important aspects of this work.

Reply 2: As suggested, we isolated retinal ECs from WT and Sdc2ECKO mice retinas and measured expression of multiple VEGFA/VEGFR2-induced genes in freshly-sorted cells (no in vitro culture). These data clearly indicate reduced VEGFR2 signaling also in vivo. (Fig. S5E, lines 176-186, see below also) Taken together with the data already in the manuscript that show angiogenic, VEGF-A

dependent defects in 3 different highly VEGFA-dependent models in mice with endothelial specific Sdc2 deletion (i.e. retinal angiogenesis, VEGFA-induced corneal angiogenesis, and hind limb ischemia), these data clearly establish reduced VEGF/VEGFR2 signaling in vivo.

R#3 Reply 2: Ok

Major concerns:

1. The current manuscript requires substantial new in vivo data to validate the in vitro results:
- Analysis in Sdc2 mutants of Vegfr2 signalling levels and downstream targets.

Reply: Similar to our in vitro data, we now show significantly reduced VEGFR2 activation in whole-retinal lysate (P6) after Tamoxifen-induced Sdc2 endothelial deletion (P1 to P5) (Supplementary Figure 5D-E)

R#3 Reply: Unfortunately, the authors' choice to assess VEGFR2 signaling in lysates of whole postnatal retinas was not the best. Okabe et al (2014) Cell, 159, 584-96 and other papers have shown that in the retina, VEGFR2 is expressed in neurons at levels that are much higher than those present in ECs. Moreover, ECs are only a very minor fraction (less than 1%) of the entire postnatal retina tissue. Thus, western blot of whole retinal lysates cannot be used to evaluate VEGFR2 activity in endothelial cells. Authors should have instead performed immunostaining to profile the protein or phosphorylation levels of Vegfr2 or several published canonical downstream targets (ERK, Esm1, Vegfr3, etc...). The authors can also FACS sort ECs from different tissues of their mutants, and perform Western/qRT-PCR analysis for Vegfr2 signalling or downstream targets.

Reply 2: While it is true that VEGFR2 is expressed in non-ECs in the retina as indicated by the reviewer, our data show a significant reduction in VEGFR2 phosphorylation in Sdc2ECKO retinas. Since there is no reason to think that an endothelial Sdc2 KO would affect VEGFR2 phosphorylation in neuronal cells, these data are conclusive. The suggested experiment by the reviewer- to assess VEGFR2 phosphorylation in the retina is not feasible as we are not aware of any commercial antibodies that reliably detect VEGFR2 phosphorylation on tissue sections. Nevertheless, we provide a new set of data to further settle this issue: we sorted retinal ECs from Sdc2ECKO and WT mice and used qPCR to measure expression of a number known downstream target of the VEGFA-PLC γ /IP3 pathway (Fig. S5E, lines 176-186). These genes are strongly induced by VEGFA and thus allow for best sensitivity to detect expression changes. All of them show reduced expression in Sdc2ECKO mice endothelial cells.

R#3 Reply 2: Ok

In the discussion line 318, the authors mention "we have not investigated whether Sdc2 deletion differentially affects various VEGFR2 downstream signaling pathways (e.g. PLC γ /ERK, AKT/Src, integrin activation, etc...)". This reviewer believes it is of high relevance to do this, in order to support the paper's main message.

Reply 2: Analysis of all major VEGFA-VEGFR2 downstream effectors (ERK, AKT, SRC, Integrin β 3) have been added (Lines 169-172, Fig. S5A-B). All show reduced signaling as would be expected.

R#3 Reply 2: Ok

Moreover, Okabe et al (2014) Cell, 159, 584-96 have also shown that a reduction in neuronal VEGFR2 signalling alone leads to alterations in retinal angiogenesis, with retinas displaying "delays in the radial outgrowth and decreased vascular density in the superficial plexus when compared to

control mice". Given the authors whole retina lysate western blot results, do the authors conclude that VEGFR2 signalling is also impaired in the neurons of their Sdc2 ECspecific mutants? The authors should comment/address this issue.

Reply 2: We now show a reduced expression of multiple VEGFR2 downstream genes in freshly sorted ECs from Sdc2ECKO retinas (Fig. S5E, lines 176-186). This show that VEGFR2 signaling is directly affected in ECs.

R#3 Reply 2: Agree. However, it does not exclude the possibility of endothelial Sdc2, which also exists in a shedded form, having a paracrine effect on neuronal VEGFR2 signalling. Given that ECs constitute less than 1% of all retinal cells, it seems unlikely that the observed decrease in pVEGFR2 in whole retinal lysates is quantitatively robust or solely due to reduced endothelial VEGFR2 activation.

- Analysis of other vascular parameters in Sdc2 mutants, such as endothelial proliferation, density and sprouting.

Reply: Characterization of the retinal vascular phenotype in Sdc2 ECKO mice has been expanded. We now show reduced number of tip cells (Supplementary figure 3C-D). Reductions in vascular density and proliferation are also observed in Sdc2 ECKO (supp. Figure 3E-F) Additionally, we have performed analysis of VEGFA-induced biological effects in vitro. We show that Sdc2^{-/-} ECs have reduced VEGFA-induced proliferation and migration, while FGF2 responses are not affected (Supplementary Figure S4).

R#3 Reply: The authors have now improved the in vivo analysis by providing quantifications of images that were not previously quantified and by analysing other vascular parameters such as endothelial proliferation, density and sprouting. However, the quantifications/charts provided do not follow the latest Nature communications editorial policy. Specifically they should present dot-plots, instead of just bars, and should have indication of what the dots and error bars represent. For most data shown, the error bars are very small, particularly for the datasets obtained in vivo. It is also not clear in most figure legends or in the methods, how the numbers presented in the charts were obtained (i.e. how many animals/retinas or how many pictures per animal/retina were used for the quantifications shown on the charts).

Reply 2: These data are now presented as dot-blots (Fig. S3D, F, G) and animal/retinas are clarified in supplementary legend 3

R#3 Reply 2: My comments related with ALL figures and datasets. The authors show Dot-plots only in Sup. Fig. 3. I leave this to the editor and the Nature Communications policy.

- Comparison of Sdc2 mutant phenotypes with Vegfr2^{+/-} or Vegfr2 Pdgfb/Cdh5-CreERT2 inducible heterozygous deletion phenotypes. These have a 50% reduction in VEGF signaling and allow for a better phenotypic comparison.

Reply: There is a large body of literature dealing with these phenotypes. Given the number of published studies, it seems hardly reasonable to carry out yet another VEGFR2 het study. The information below has been added to the Discussion (page 13, lines 297-311): VEGFA concentration is the key limiting factor regulating VEGFR2 signaling. That is, VEGFR2 signaling is ligand-dependent and the amount of ligand available to bind VEGFR2 is far less than the number of VEGFR2 binding sites available. Thus, a deletion of a single VEGFR2 allele DOES NOT lead to a 50% reduction in VEGFA signaling while a single allele deletion of VEGFA is lethal.

- Shalaby et al (Nature, 1996) show no measurable phenotypes during embryonic development in VEGFR2 hets.

- Lars Jakobsson (Gerhardt lab, Nat Cell Biol 2010) showed in competition assays that VEGFR2^{+/-}

ECs move slower than VEGFR2^{+/+} ECs. This is really a migration and not a proliferation assay. In agreement with this, we also see decreased migration of Sdc2^{-/-} ECs in response to VEGFA.

- Silvaraj et al (Dev Cell 2013) show 15% reduction in the retinal angiogenesis extent and ~35% reduction in the number of branch points in VEGFR2 hets generated with PdgfbⁱCre (in Suppl Fig 2F, G). These numbers are consistent with our observations.

In summary, 35% reduction in VEGF signaling here is highly biologically meaningful given that 50% reduction is lethal. VEGF signaling is driven by ligand availability. Receptor levels are of secondary importance. Sdc2 role in facilitating VEGFA-VEGFR2 complex formation is critical as the absence of Sdc2 leads to a 35% reduction in signaling with measurable in vivo biological effects.

R#3 Reply: What is the published evidence for the authors sentence "deletion of a single VEGFR2 allele DOES NOT lead to a 50% reduction in VEGFA signaling" ? The authors must cite and discuss it to validate their arguments/hypothesis. The fact that most VEGFA KO/wt mutants die during embryonic development, whereas most VEGFR2 KO/Wt mutants survive, cannot be exclusively correlated with Vegfr2 signalling dose, for several reasons. The first being that VEGFA can bind to several other canonical and non-canonical VEGF receptors/adaptor molecules. The second, and in agreement with the ligand vs receptor relative signalling dynamics proposed by the authors, Vegfr2 heterozygous (KO/wt) animals can still have a 50% decrease in Vegfr2 signalling dose, whereas VEGFA heterozygous (KO/Wt) mutants may have an even more pronounced decrease in Vegfr2 signalling, which would explain why the latter have much more pronounced vascular/angiogenesis defects. Oladipupo et al (2018), Sci Rep., 8, 14724 -> analyse heterozygous mice for VEGFR2 (Vegfr2^{Cre/+} or Vegfr2^{LacZ/+} knockin/knockout mice) and observe a decrease above 50% in tumor angiogenesis (Fig1D, Fig2D, Fig3C) and in p-VEGFR2 staining in tumor vessels (Fig3C). Regarding the citation Shalaby et al., Nature 1996, I believe the authors are referring to Shalaby et al (1995), Nature, 376, 62-66. In this paper, VEGFR2 KO/+ embryos are only compared with VEGFR2 KO/KO littermates and not with wild-type embryos. Therefore it sheds no light onto the phenotype of VEGFR2 het mice. I do not understand the authors point here, except that Vegfr2 hets do not die during development, like Sdc2 full KO animals. Regarding Jakobsson et al., NCB 2010 paper, once again the authors cite it incorrectly. Jakobsson et al paper state "Cell tracking demonstrated that wild-type cells (expressing DsRed or yellow fluorescent protein; YFP) and Vegfr2^{-/+} egfp cells migrated at roughly similar velocity and with similarly persistent directionality (Fig. 5f, g)." Thus, a true 50% reduction in VEGFR2 protein levels (Fig. S2A of the same cited paper) does not seem to lead to endothelial cell migration defects, as the authors wrongly cite in their text and associate with their Sdc2 mutant phenotypes, with an estimated 35% reduction in Vegfr2 signalling. Regarding the citation Sivaraj et al (2013), Dev Cell, 25, 427-434 -> they indeed show a 15% reduction in the retinal angiogenesis extent and ~35% reduction in the number of branch points in VEGFR2 hets generated with PdgfbⁱCreERT2 (in Suppl Fig 2F, G). However, special attention has to be made to the fact that the retinas analysed in the cited paper are with inducible deletion of Vegfr2, which was not validated. Zarkada et al (2015), PNAS, 112, 761- 766, shows in Figure 1D that the deletion efficiency of VEGFR2 varies from 20% to 80% (Figure 1E-J).

In summary, regardless of the relative importance for signalling of ligand vs receptor dose, VEGFR2 heterozygous mice/cells do have a 50% decrease in VEGFR2 expression/protein, and presumably signalling, which would be the best comparison to justify that a 35% decrease in Vegfr2 signalling in Sdc2 mutants is biologically relevant. Several papers on Vegfr2 signalling, including the ones cited by the authors, show that the authors' following argument is not correct "Receptor levels are of secondary importance".

Reply 2: While this is an interesting topic for debate, it seems to have little relevance to the study that examines the effect of amino acid sequence on HS sulfation pattern. We should beyond a shadow of doubt that VEGF-A binding is decreased in vitro and VEGF-dependent process are impaired in vivo. Whether the extent of VEGF signaling reduction in these mice is similar or not to VEGFR2 hets is simply not relevant to this study.

R#3 Reply 2: The authors removed some of the wrongly cited papers, and corrected their

interpretation and discussion of others. The authors still do not show any in vivo data supporting that a 35% decrease in VEGFR2 activity, or a 2-3 fold decrease in VEGF target genes expression (that has cited are deregulated 10 to 600 fold by VEGF) in their mutants, has any biological relevance, as compared to 50% Vegfr2 signalling decrease (Vegfr2+/- animals). They also do not fully discuss it.

- Mutate the endogenous Sdc2 gene, to analyze the impact on Vegfr signaling and angiogenesis of the same Sdc2 mutant forms characterized in vitro.

Reply: This is an interesting idea that, we believe, is outside of the scope of this manuscript that deals with the effect of protein sequence on a sulfation pattern. Especially given how long it would take to do these studies and how much this would cost. We clearly show that reduced VEGFA signaling in Sdc2-/- EC (due to specific loss of Sdc2 HS chains) is involved in determining vascular phenotype in vivo and phenotypic effects in vitro. This does not exclude that Sdc2 deletion may also have other HS-independent consequences (i.e. due to loss of Sdc2 core protein)

- If possible modulate Sdc2 glycosylation and sulfation in vivo, or its binding to VEGF.

Reply: Currently, there is not feasible way to selectively modulate glycosylation or sulfation in vivo or in vitro.

2. The authors have not presented and discussed all existing bibliographic data on the other published work regarding the role of Sdc2 in angiogenesis. Papers from Rossi et al (2014) J. Cell Science 127, 4788 and Noguera et al (2009) Exp Cell Res 315, 795-808 are two such examples. These papers propose opposite or similar roles of Sdc2, to the ones presented in this manuscript. This data has to be carefully contrasted and discussed to reveal what is novel and different in this manuscript.

Reply: These have been added to Discussion (page 14, lines 310-316):

R#3 Reply: Bibliographic data that had not been previously presented and discussed, namely Rossi et al (2014) J. Cell Science 127, 4788 and Noguera et al (2009) Exp Cell Res 315, 795-808, has now been introduced in the current manuscript. However, Noguera et al has been incorrectly referred to. The authors cite Noguera et al., saying that the data presented in this manuscript is "in agreement with early reports of in vitro Sdc2 silencing resulting in reduced ECs migration and cord formation in the Matrigel assay 55, 56, however Noguera et al shows that downregulation of Sdc2 enhances EC migration (also written in their paper abstract). Rossi et al., has also shown that shed Sdc2 acts as a negative (not positive) regulator of VEGF signalling by sequestering it. In this context, it is important for the authors to better discuss the contradicting data or try to assess the relative levels of shed versus non-shed Sdc2 in endothelial cells. How much of the endothelial Sdc2 exists at the membrane to form a complex with VEGFR2-VEGFA165, versus the one that is shed and may be inhibitory? There should exist feasible ways of quantifying the different forms of Sdc2 produced by ECs and characterize their distinct activities on signalling.

Reply 2: More extensive discussion of previous data has been added (lines 326-338).

R#3 Reply 2: Rossi et al (2014) J. Cell Science 127, 4788 should also be cited again in this paragraph, as it contradicts the results reported by Fears et al (2006) J. Biol. Chem. 281, 14533-14536. Moreover the results obtained by Rossi et al are not exclusively based on the addition of extracellular Sdc2 to cell culture medium but also on TNF α -induced shedding of Sdc2.

3. The authors present that Sdc4 is important in FGF-dependent signal transduction however no significant effect is obtained in the cornea pocket assay with FGF2 pellets. Surprisingly, the authors claim that "Sdc4-/-, but not Sdc2iCdh5 showed a reduction in angiogenesis in response to

FGF2 (Fig. 1g-h)". The indicated figures show Sdc4 mutants present no difference in both VEGF- and FGF2-induced angiogenesis when compared to control mice.

Reply: We apologize for the writing error. Indeed, we intended to write that both VEGFA- and FGF2-induced angiogenesis in cornea-pocket assay WAS NOT affected in Sdc4-/- mice. Sdc4 controls other biological aspects related to FGF-signaling, particularly mTOR/AKT/eNOS pathway which modulate systemic blood pressure.

4. Analysis of the EC-specific Sdc2 LOF leads to delayed vessel outgrowth. Retinas in Fig1A appear to have different sizes, ie., the Sdc2 mutant whole retina tissue seems smaller. Instead of presenting the absolute values of vascular outgrowth, it would be better to normalize the vascular outgrowth to the total size of the retina. The control and mutants do not seem stage matched.

Reply: The quantification of retina outgrowth measures the distance between the center and vascular front. Difference size of retinas in the avascular parts are due technical variability during eye opening and flattening for retinal staining. We have not observed significant differences between total size of retinas across genotypes. Additionally, for reader's confidence, the vascular progression of each strain is now shown as normalized to Sdc2fl/fl (new quantification in Fig. 1B).

R#3 Reply: It is precisely because of technical variability that the retinal outgrowth should be measured as a ratio that reflects the amount of retina that is vascularized. Thus retina outgrowth should be VD/PD, being VD = distance between the center and vascular front and PD = distance between the center and the periphery of the retina. This should be addressed appropriately, as Reviewer 1 expressed similar concerns.

Reply 2: Quantification has been changed as suggested by the reviewer (Fig 1B, and lines 423-424 in method section)

R#3 Reply 2: Ok

Minor Concerns:

1. Authors show by immunostaining that Sdc2 is expressed in ECs of the aorta and in the retina, where it seems that it is mostly enriched in arteries and arterioles. Images representative of all regions of the retina (vein and venous plexus, angiogenic front and sprouts) are necessary.

Reply: Additional staining of retinas showing Sdc2 expression have been added (Supplementary figure 1e). Syndecan-2 is expressed in arteries, veins and capillaries. It may be true that Sdc2 is enriched in arterial versus venous EC, however the fact that it is expressed in both SMCs and ECs does not allow us to conclude this.

R#3 Reply: The authors now present data on the expression of Sdc2 in veins and capillaries, in addition to retinal arteries. Unfortunately, the authors have not provided any data on the expression of Sdc2 at the angiogenic front (where sprouting and proliferating cells locate), which would be of interest as they propose that the observed phenotype is related with the abrogation of VEGF-VEGFR2-Sdc2 interaction that is needed for optimal VEGFR2 signalling, and EC migration and proliferation. Given that VEGF levels are highest at the angiogenic front/leading edge of the retina, this is where one would expect a more relevant role of Sdc2 (and perhaps expression). Ideally, an image of a whole retinal flank with immunostaining for Sdc2 and higher magnification insets of the different regions should be provided.

REPLY 2: We have added expression of Sdc2 in tip cells (Fig. S1E-lower left inset). Unfortunately, quality of all available Sdc2 antibodies is poor (not surprisingly) and we cannot have the highest staining quality. We are developing a custom-made antibody against mouse Sdc2 that should permit better staining and WB/IP analysis in our future works.

R#3 Reply 2: The authors have plenty of space to show the full size angiogenic front and tip cells figures. Why a small inset is provided ? The quality of the signal is good enough.

2. None of the images presented in the manuscript have scale bars.

Reply: scale bars have been added

R#3 Reply: images S1D; E and S2A, D still do not have scale bars.

Reply 2: Scale bars have been added.

R#3 Reply 2: Ok

3.No data has been presented regarding the endothelial expression of Sdc4.

Reply: There are numerous publications demonstrating high Sdc4 expression in ECs, including in the retinal vasculature. This information, along with citations to most recent papers, has been added to the text (page 6, lines 132-133).

4.Authors do not attribute the decrease in vessel density to enhanced vascular pruning in the Sdc2 mutants because "collagen IV staining did not detect empty sleeves". Since collagen empty sleeves are indeed detectable in Fig S2A, we infer that the authors meant that there were no differences in the frequency of empty sleeves between control and mutant retinas. However, no quantification data is presented.

Reply: Quantification has been added (Supplementary Figure 3B)

5.Quantifications on the density of vascular networks in the skin and kidney are not presented.

Reply: We have decided to remove these data and to focus on retina vascular development and hindlimb ischemia, two settings in which relative contribution of VEGFA versus other factors is extremely well characterized.

6. Although the authors claim the role of Sdc2 on angiogenesis is mediated through the VEGFA165-VEGFR2 axis, there is also evidence that Sdc2 also interacts with integrins (Exp. Cell. Res (2000), Vol.256, 434 and Biophysics Res Commun (2009), Vol385, 231), which is important for cell migration and adhesion. In fact, there are several papers that show VEGFA-mediated functions dependent on integrins (Blood 2012 120:4892-4902; J Biol Chem. 2011 286: 1083/1092; JBC 2007 282, 15187-15196; Embo J 1999; 18:882-892; Mol Cell 2000; 6:851-860; Circ Res 2007; 101:570-580). To assess integrin signaling, immunostaining for p-FAK or active-integrinB1 could be used.

Reply: We agree that integrins are important in VEGFr2 signaling. How that integrates with Sdc2 is less clear and is the subject for further studies. Our data clearly show that Sdc2 is essential for formation of VEGFA165-VEGFR2-Sdc2 complex which, in turn, is required for maximal VEGFR2 phosphorylation. Whether an integrin, such as beta-2 is involved is not certain. Integrins activation is one of the VEGFR2 downstream target and likely to be affected as the other downstream targets. This possibility and relative references are added to discussion as suggested. (pages 14, lines 317-323).

R#3 Reply: AS mentioned in the answer to major concern 1, since the decrease in endothelial Vegfr2 signaling in vivo is still not clear, and as the authors also mention in the discussion ("we have not investigated whether Sdc2 deletion differentially affects various VEGFR2 downstream signaling pathways (e.g. PLCγ/ERK, AKT/Src, integrin activation, etc...."), it should be analyzed how ERK/AKT/Src/Integrin signaling pathways are affected in Sdc2 mutants, to provide a better

mechanistic explanation for the defects observed in EC migration/sprouting/proliferation.

Reply: Analysis of all major VEGFA-VEGFR2 downstream effectors (ERK, AKT, SRC, Integrin- β 3) have been added (Lines 169-172, Fig. S5A-B).

R#3 Reply 2: Ok

7. In vivo validation of the in vitro co-IP data would also be interesting, using for example the tissues from animals.

Reply: In the proteoglycans world, IP and WB experiments are severely limited by lack of working antibodies against endogenous proteins. Sdc2 is no exceptions. This is the reason why most studies in this field rely on the use of epitope-tags and overexpression systems. There is no possibility of getting a Sdc2-specific IP from a tissue lysate.

8. It would be important to check by western blot if Sdc4 is regulated in the absence of Sdc2 and vice-versa.

Reply: As requested, we now show Sdc4 blot in Figure 3A (Sdc2-/- ECs) and Sdc2 blot in Figure 3D (Sdc4-/- ECs). We did not detect any evident upregulation.

9. In Fig3H and Fig 4G, it appears that Sdc2 and Sdc4 have opposite effects on p-VEGFR2 (comparing to cells transduced with only Ad-GFP). Could the authors comment on this?

Reply: The only statistically significant effect we see in these studies is when Sdc2 is reintroduced into Sdc2-/- ECs.

10. All interaction assays were performed in cells transduced with adenovirus and expressing different mutated forms of Sdc2/4. It is known that over-expression of proteins can sometimes lead to unspecific protein interactions. How do the total levels of these proteins compare to the endogenous levels? Can the same experiments be performed with CRISPR/Cas9 induced mutations in Sdc2?

Reply: We agree this is an important point. IP experiments following adenovirus transduction (at MOI = ~ 1-2) were performed with minimal overexpression which was similar to syndecans endogenous level (supplementary figures 6A-B) CRISPR/Cas9 mutagenesis would be a whole new study and approach would still be limited by lack of good antibody against endogenous protein without epitope-tag.

11. Although the protein sequence of the D1 domains of Sdc2 are well conserved among species, I wonder whether the glycosylation events are also well conserved, since this is known to vary depending on cell context. It would be important to confirm that glycosylation pattern in mouse endothelial cells are identical to those described in the manuscript and ideally in the absence of overexpression of the proteins.

Reply: That would be a good study to do. Unfortunately, high-quality antibody for affinity purification of endogenous syndecans do not exist. Nevertheless, the point of testing a different species is well taken and we have repeated HS chain analysis of both sdc2 and sdc4 isolated from a mouse endothelial cell line. We now show that also mouse ECs generate higher level of 6-O-sulfation in Sdc2 HS chains than in Sdc4 HS chains (Supplementary Figure 6E). This is similar to what we observed in HUVECs

Manuscript changes from previous version

- Discussed possibility of paracrine effect by shed Sdc2 on neuronal VEGFR2 signaling (lines discussion (line 330-333))
- Dot-blot overlapping bar graphs have been added whenever possible
- Added molecular weight markers to western blot images
- Figure 1f expressed as fold changes for clarity
- For clarity and better comparison, labelling of Figure 6b x-axis is presented as in 6a.
- Added uncropped western blot images in supplementary figures

REVIEWERS' COMMENTS:

Reviewer #3 (Remarks to the Author):

Overview of manuscript changes:

- Quantification of vascular progression in retina has been corrected (Fig 1B, and lines 427-248 in method section)
- ERK activation (previous Fig.3A,C) merged with other VEGFR2 downstream effectors (Supp. Fig. 5A-B)
- Vascular parameters in retinas are now reported as dot-blot (Supp. Fig 3C-G). Number animal/retinas added to legends
- Effect of Sdc2 silencing in HUVEC (previous Supp. Fig. 5A-C) moved to Supp. Fig 4G-I
- Analysis of VEGFA target genes expression in retina ECs has been added (Supp. Fig. 5E)
- Sdc2 blot in Supp. Fig. 5F showed a wrong image (last lane indicate "Sdc2-/- + Ad-Sdc4 " but it was showing expression of Sdc2 in wild-type cells). A representative picture that match legend is now shown.
- Primer sequences are now indicated in a separate table (Table 2)

Reviewers' comments:

R#3 Reply 2: General Remarks: The authors have done most of the requested experiments and changes. They also corrected some of the previously wrong citations. The in vivo angiogenesis analysis could still be improved further. However, the paper is stronger on in vitro Sdc/glycobiology and VEGFR signalling studies, some of which are difficult to be fully validated by in vivo experiments.

Reviewer #3 - expert in angiogenesis (Remarks to the Author):

In the manuscript by Federico et al, the authors study the role of Sdc-2 in angiogenesis using the retinal and hindlimb ischemia models, concluding that it impairs VEGFA-165-driven angiogenesis. Mechanistically, and resorting exclusively to in vitro studies, the authors propose that the increased levels of 6-O sulfation of Sdc2 HS chains confer it the unique ability to form a ternary complex Sdc2-VEGFA165-VEGFR2, which is proposed to enhance VEGFR2 signaling. The paper is Ok when it comes to demonstrating how different biochemical modifications may change the modulation of VEGFR2 signaling by Sdc2 in vitro. However, the in vivo analysis performed was very superficial, poorly quantified and there is no clear evidence that the molecular mechanism identified in vitro is at the source of the observed phenotype in vivo. The manuscript needs a significant amount of new in vivo data to further validate the in vitro findings. In addition, the weak regulation of Vegfr2 signaling identified in Sdc2KO cells (-35% signaling found in ECs in vitro) and the very mild defects seen in Sdc2Pdgfb mutant retinas, shows that the regulation is not of enough biological significance.

Reply: We thank the reviewer suggestions to improve the manuscript. Importantly, we now show strong *in vivo* evidence that the observed 35% reduction in VEGFR2 signaling activation has biological relevance and it is involved in determining the vascular phenotype shown in Sdc2 endothelial-specific KO.

R#3 Reply: The authors still do not provide clear evidence that VEGFR2 signalling is decreased in endothelial cells *in vivo* (see below), and how the estimated 35% reduction in VEGFR2 signalling causes the observed *in vivo* vascular/angiogenesis phenotype, which is one of the most important aspects of this work.

Reply 2: As suggested, we isolated retinal ECs from WT and Sdc2ECKO mice retinas and measured expression of multiple VEGFA/VEGFR2-induced genes in freshly-sorted cells (no *in vitro* culture). These data clearly indicate reduced VEGFR2 signaling also *in vivo*. (Fig. S5E, lines 176-186, see below also) Taken together with the data already in the manuscript that show angiogenic, VEGF-A dependent defects in 3 different highly VEGFA-dependent models in mice with endothelial specific Sdc2 deletion (i.e. retinal angiogenesis, VEGFA-induced corneal angiogenesis, and hind limb ischemia), these data clearly establish reduced VEGF/VEGFR2 signaling *in vivo*.

R#3 Reply 2: Ok

Major concerns:

1. The current manuscript requires substantial new *in vivo* data to validate the *in vitro* results:
 - Analysis in Sdc2 mutants of Vegfr2 signalling levels and downstream targets.

Reply: Similar to our *in vitro* data, we now show significantly reduced VEGFR2 activation in whole-retinal lysate (P6) after Tamoxifen-induced Sdc2 endothelial deletion (P1 to P5) (Supplementary Figure 5D-E)

R#3 Reply: Unfortunately, the authors' choice to assess VEGFR2 signaling in lysates of whole postnatal retinas was not the best. Okabe et al (2014) Cell, 159, 584-96 and other papers have shown that in the retina, VEGFR2 is expressed in neurons at levels that are much higher than those present in ECs. Moreover, ECs are only a very minor fraction (less than 1%) of the entire postnatal retina tissue. Thus, western blot of whole retinal lysates cannot be used to evaluate VEGFR2 activity in endothelial cells. Authors should have instead performed immunostaining to profile the protein or phosphorylation levels of Vegfr2 or several published canonical downstream targets (ERK, Esm1, Vegfr3, etc...). The authors can also FACS sort ECs from different tissues of their mutants, and perform Western/qRT-PCR analysis for Vegfr2 signalling or downstream targets.

Reply 2: While it is true that VEGFR2 is expressed in non-ECs in the retina as indicated by the reviewer, our data show a significant reduction in VEGFR2 phosphorylation in Sdc2ECKO retinas. Since there is no reason to think that an endothelial Sdc2 KO would affect VEGFR2 phosphorylation in neuronal cells, these data are conclusive. The suggested experiment by the reviewer- to assess VEGFR2 phosphorylation in the retina is not feasible as we are not aware of any commercial antibodies that reliably detect VEGFR2 phosphorylation on tissue sections. Nevertheless, we provide a new set of data to further settle this issue: we sorted retinal ECs from Sdc2ECKO and WT mice and used qPCR to measure expression of a number known downstream target of the VEGFA-PLC γ /IP3 pathway (Fig. S5E, lines 176-186). These genes are strongly induced by VEGFA and thus allow for best sensitivity to detect expression changes. All of them show reduced expression in Sdc2ECKO mice endothelial cells.

R#3 Reply 2: Ok

In the discussion line 318, the authors mention “we have not investigated whether Sdc2 deletion differentially affects various VEGFR2 downstream signaling pathways (e.g. PLC γ /ERK, AKT/Src, integrin activation, etc....” . This reviewer believes it is of high relevance to do this, in order to support the paper’s main message.

Reply 2: Analysis of all major VEGFA-VEGFR2 downstream effectors (ERK, AKT, SRC, Integrin β 3) have been added (Lines 169-172, Fig. S5A-B). All show reduced signaling as would be expected.

R#3 Reply 2: Ok

Moreover, Okabe et al (2014) Cell, 159, 584-96 have also shown that a reduction in neuronal VEGFR2 signalling alone leads to alterations in retinal angiogenesis, with retinas displaying “delays in the radial outgrowth and decreased vascular density in the superficial plexus when compared to control mice”. Given the authors whole retina lysate western blot results, do the authors conclude that VEGFR2 signalling is also impaired in the neurons of their Sdc2 ECspecific mutants? The authors should comment/address this issue.

Reply 2: We now show a reduced expression of multiple VEGFR2 downstream genes in freshly sorted ECs from Sdc2ECKO retinas (Fig. S5E, lines 176-186). This show that VEGFR2 signaling is directly affected in ECs.

R#3 Reply 2: Agree. However, it does not exclude the possibility of endothelial Sdc2, which also exists in a shedded form, having a paracrine effect on neuronal VEGFR2 signalling. Given that ECs constitute less than 1% of all retinal cells, it seems unlikely that the observed decrease in pVEGFR2 in whole retinal lysates is quantitatively robust or solely due to reduced endothelial VEGFR2 activation.

Reply 3: Yes, this possibility has been added to discussion (page 15, line 330-333)

- Analysis of other vascular parameters in Sdc2 mutants, such as endothelial proliferation, density and sprouting.

Reply: Characterization of the retinal vascular phenotype in Sdc2 ECKO mice has been expanded. We now show reduced number of tip cells (Supplementary figure 3C-D). Reductions in vascular density and proliferation are also observed in Sdc2 ECKO (supp. Figure 3E-F) Additionally, we have performed analysis of VEGFA-induced biological effects in vitro. We show that Sdc2 $^{-/-}$ ECs have reduced VEGFA-induced proliferation and migration, while FGF2 responses are not affected (Supplementary Figure S4).

R#3 Reply: The authors have now improved the in vivo analysis by providing quantifications of images that were not previously quantified and by analyzing other vascular parameters such as endothelial proliferation, density and sprouting. However, the quantifications/charts provided do not follow the latest Nature communications editorial policy. Specifically, they should present dot-plots, instead of just bars, and should have indication of what the dots and error bars represent. For most data shown, the error bars are very small, particularly for the datasets obtained in vivo. It is also not clear in most figure legends or in the methods, how the numbers presented in the charts were obtained (i.e. how many animals/retinas or how many pictures per animal/retina were used for the quantifications shown

on the charts).

Reply 2: These data are now presented as dot-blots (Fig. S3D, F, G) and animal/retinas are clarified in supplementary legend 3

R#3 Reply 2: My comments related with ALL figures and datasets. The authors show Dot-plots only in Sup. Fig. 3. I leave this to the editor and the Nature Communications policy.

- Comparison of Sdc2 mutant phenotypes with Vegfr2^{+/-} or Vegfr2 Pdgfb/Cdh5-CreERT2 inducible heterozygous deletion phenotypes. These have a 50% reduction in VEGF signaling and allow for a better phenotypic comparison.

Reply: There is a large body of literature dealing with these phenotypes. Given the number of published studies, it seems hardly reasonable to carry out yet another VEGFR2 het study. The information below has been added to the Discussion (page 13, lines 297-311): VEGFA concentration is the key limiting factor regulating VEGFR2 signaling. That is, VEGFR2 signaling is ligand-dependent and the amount of ligand available to bind VEGFR2 is far less than the number of VEGFR2 binding sites available. Thus, a deletion of a single VEGFR2 allele DOES NOT lead to a 50% reduction in VEGFA signaling while a single allele deletion of VEGFA is lethal.

- Shalaby et al (Nature, 1996) show no measurable phenotypes during embryonic development in VEGFR2 hets.

- Lars Jakobsson (Gerhardt lab, Nat Cell Biol 2010) showed in competition assays that VEGFR2^{+/-} ECs move slower than VEGFR2^{+/+} ECs. This is really a migration and not a proliferation assay. In agreement with this, we also see decreased migration of Sdc2^{-/-} ECs in response to VEGFA.

- Silvaraj et al (Dev Cell 2013) show 15% reduction in the retinal angiogenesis extent and ~35% reduction in the number of branch points in VEGFR2 hets generated with PdgfbCre (in Suppl Fig 2F, G). These numbers are consistent with our observations.

In summary, 35% reduction in VEGF signaling here is highly biologically meaningful given that 50% reduction is lethal. VEGF signaling is driven by ligand availability. Receptor levels are of secondary importance. Sdc2 role in facilitating VEGFA-VEGFR2 complex formation is critical as the absence of Sdc2 leads to a 35% reduction in signaling with measurable *in vivo* biological effects.

R#3 Reply: What is the published evidence for the authors sentence "deletion of a single VEGFR2 allele DOES NOT lead to a 50% reduction in VEGFA signaling" ? The authors must cite and discuss it to validate their arguments/hypothesis. The fact that most VEGFA KO/wt mutants die during embryonic development, whereas most VEGFR2 KO/Wt mutants survive, cannot be exclusively correlated with Vegfr2 signalling dose, for several reasons. The first being that VEGFA can bind to several other canonical and non-canonical VEGF receptors/adaptor molecules. The second, and in agreement with the ligand vs receptor relative signalling dynamics proposed by the authors, Vegfr2 heterozygous (KO/wt) animals can still have a 50% decrease in Vegfr2 signaling dose, whereas VEGFA heterozygous (KO/Wt) mutants may have an even more pronounced decrease in Vegfr2 signalling, which would explain why the latter have much more pronounced vascular/angiogenesis defects. Oladipupo et al (2018), Sci Rep., 8, 14724 -> analyse heterozygous mice for VEGFR2 (Vegfr2Cre/+ or Vegfr2LacZ/+ knockin/knockout mice) and observe a decrease above 50% in tumor angiogenesis (Fig1D, Fig2D, Fig3C) and in p-VEGFR2 staining in tumor vessels (Fig3C). Regarding the citation Shalaby et al., Nature 1996, I believe the authors are referring to Shalaby et al (1995), Nature, 376, 62-66. In this paper, VEGFR2 KO/+ embryos are only compared with VEGFR2 KO/KO littermates and not with wild-type embryos. Therefore it sheds no light onto the phenotype of VEGFR2 het mice. I do not understand the authors point here, except that Vegfr2 hets do not die during development, like Sdc2 full KO animals. Regarding Jakobsson et al., NCB 2010 paper, once again the authors cite it incorrectly. Jakobsson et al paper state "Cell tracking demonstrated that wild-type cells (expressing

DsRed or yellow fluorescent protein; YFP) and Vegfr2 +/- egfp cells migrated at roughly similar velocity and with similarly persistent directionality (Fig. 5f, g).” Thus, a true 50% reduction in VEGFR2 protein levels (Fig. S2A of the same cited paper) does not seem to lead to endothelial cell migration defects, as the authors wrongly cite in their text and associate with their Sdc2 mutant phenotypes, with an estimated 35% reduction in Vegfr2 signalling. Regarding the citation Sivaraj et al (2013), Dev Cell, 25, 427-434 -> they indeed show a 15% reduction in the retinal angiogenesis extent and ~35% reduction in the number of branch points in VEGFR2 hets generated with Pdgfb- iCreERT2 (in Suppl Fig 2F, G). However, special attention has to be made to the fact that the retinas analysed in the cited paper are with inducible deletion of Vegfr2, which was not validated. Zarkada et al (2015), PNAS, 112, 761- 766, shows in Figure 1D that the deletion efficiency of VEGFR2 varies from 20% to 80% (Figure 1E-J).

In summary, regardless of the relative importance for signalling of ligand vs receptor dose, VEGFR2 heterozygous mice/cells do have a 50% decrease in VEGFR2 expression/protein, and presumably signalling, which would be the best comparison to justify that a 35% decrease in Vegfr2 signalling in Sdc2 mutants is biologically relevant. Several papers on Vegfr2 signalling, including the ones cited by the authors, show that the authors’ following argument is not correct “Receptor levels are of secondary importance”.

Reply 2: While this is an interesting topic for debate, it seems to have little relevance to the study that examines the effect of amino acid sequence on HS sulfation pattern. We should beyond a shadow of doubt that VEGF-A binding is decreased in vitro and VEGF-dependent process are impaired in vivo. Whether the extent of VEGF signaling reduction in these mice is similar or not to VEGFR2 hets is simply not relevant to this study.

R#3 Reply 2: The authors removed some of the wrongly cited papers, and corrected their interpretation and discussion of others. The authors still do not show any in vivo data supporting that a 35% decrease in VEGFR2 activity, or a 2-3 fold decrease in VEGF target genes expression (that has cited are deregulated 10 to 600 fold by VEGF) in their mutants, has any biological relevance, as compared to 50% Vegfr2 signalling decrease (Vegfr2+/- animals). They also do not fully discuss it.

- Mutate the endogenous Sdc2 gene, to analyze the impact on Vegfr signaling and angiogenesis of the same Sdc2 mutant forms characterized in vitro.

Reply: This is an interesting idea that, we believe, is outside of the scope of this manuscript that deals with the effect of protein sequence on a sulfation pattern. Especially given how long it would take to do these studies and how much this would cost. We clearly show that reduced VEGFA signaling in Sdc2-/- EC (due to specific loss of Sdc2 HS chains) is involved in determining vascular phenotype in vivo and phenotypic effects in vitro. This does not exclude that Sdc2 deletion may also have other HS-independent consequences (i.e. due to loss of Sdc2 core protein)

- If possible modulate Sdc2 glycosylation and sulfation in vivo, or its binding to VEGF.

Reply: Currently, there is not feasible way to selectively modulate glycosylation or sulfation in vivo or in vitro.

2. The authors have not presented and discussed all existing bibliographic data on the other published work regarding the role of Sdc2 in angiogenesis. Papers from Rossi et al (2014) J. Cell Science 127, 4788 and Noguera et al (2009) Exp Cell Res 315, 795-808 are two such examples. These papers propose opposite or similar roles of Sdc2, to the ones presented in this manuscript. This data has to be carefully contrasted and discussed to reveal what is novel and different in this manuscript.

Reply: These have been added to Discussion (page 14, lines 310-316):

R#3 Reply: Bibliographic data that had not been previously presented and discussed, namely Rossi et al (2014) J. Cell Science 127, 4788 and Noguier et al (2009) Exp Cell Res 315, 795-808, has now been introduced in the current manuscript. However, Noguier et al has been incorrectly referred to. The authors cite Noguier et al., saying that the data presented in this manuscript is "in agreement with early reports of in vitro Sdc2 silencing resulting in reduced ECs migration and cord formation in the Matrigel assay 55, 56, however Noguier et al shows that downregulation of Sdc2 enhances EC migration (also written in their paper abstract). Rossi et al., has also shown that shed Sdc2 acts as a negative (not positive) regulator of VEGF signalling by sequestering it. In this context, it is important for the authors to better discuss the contradicting data or try to assess the relative levels of shed versus non-shed Sdc2 in endothelial cells. How much of the endothelial Sdc2 exists at the membrane to form a complex with VEGFR2-VEGFA165, versus the one that is shed and may be inhibitory? There should exist feasible ways of quantifying the different forms of Sdc2 produced by ECs and characterize their distinct activities on signalling.

Reply 2: More extensive discussion of previous data has been added (lines 326-338).

R#3 Reply 2: Rossi et al (2014) J. Cell Science 127, 4788 should also be cited again in this paragraph, as it contradicts the results reported by Fears et al (2006) J. Biol. Chem. 281, 14533-14536. Moreover the results obtained by Rossi et al are not exclusively based on the addition of extracellular Sdc2 to cell culture medium but also on TNF α -induced shedding of Sdc2.

Reply 3: This has been added to discussion

3. The authors present that Sdc4 is important in FGF-dependent signal transduction however no significant effect is obtained in the cornea pocket assay with FGF2 pellets. Surprisingly, the authors claim that "Sdc4^{-/-}, but not Sdc2iCdh5 showed a reduction in angiogenesis in response to FGF2 (Fig. 1g-h)". The indicated figures show Sdc4 mutants present no difference in both VEGF- and FGF2-induced angiogenesis when compared to control mice.

Reply: We apologize for the writing error. Indeed, we intended to write that both VEGFA- and FGF2 - induced angiogenesis in cornea-pocket assay WAS NOT affected in Sdc4^{-/-} mice. Sdc4 controls other biological aspects related to FGF-signaling, particularly mTOR/AKT/eNOS pathway which modulate systemic blood pressure.

4. Analysis of the EC-specific Sdc2 LOF leads to delayed vessel outgrowth. Retinas in Fig1A appear to have different sizes, ie., the Sdc2 mutant whole retina tissue seems smaller. Instead of presenting the absolute values of vascular outgrowth, it would be better to normalize the vascular outgrowth to the total size of the retina. The control and mutants do not seem stage matched.

Reply: The quantification of retina outgrowth measures the distance between the center and vascular front. Difference size of retinas in the avascular parts are due technical variability during eye opening and flattening for retinal staining. We have not observed significant differences between total size of retinas across genotypes. Additionally, for reader's confidence, the vascular progression of each strain is now shown as normalized to Sdc2^{fl/fl} (new quantification in Fig. 1B).

R#3 Reply: It is precisely because of technical variability that the retinal outgrowth should be measured as a ratio that reflects the amount of retina that is vascularized. Thus retina outgrowth should be VD/PD, being VD = distance between the center and vascular front and PD = distance

between the center and the periphery of the retina. This should be addressed appropriately, as Reviewer 1 expressed similar concerns.

Reply 2: Quantification has been changed as suggested by the reviewer (Fig 1B, and lines 423-424 in method section)

R#3 Reply 2: Ok

Minor Concerns:

1. Authors show by immunostaining that Sdc2 is expressed in ECs of the aorta and in the retina, where it seems that it is mostly enriched in arteries and arterioles. Images representative of all regions of the retina (vein and venous plexus, angiogenic front and sprouts) are necessary.

Reply: Additional staining of retinas showing Sdc2 expression have been added (Supplementary figure 1e). Syndecan-2 is expressed in arteries, veins and capillaries. It may be true that Sdc2 is enriched in arterial versus venous EC, however the fact that it is expressed in both SMCs and ECs does not allow us to conclude this.

R#3 Reply: The authors now present data on the expression of Sdc2 in veins and capillaries, in addition to retinal arteries. Unfortunately, the authors have not provided any data on the expression of Sdc2 at the angiogenic front (where sprouting and proliferating cells locate), which would be of interest as they propose that the observed phenotype is related with the abrogation of VEGF-VEGFR2-Sdc2 interaction that is needed for optimal VEGFR2 signalling, and EC migration and proliferation. Given that VEGF levels are highest at the angiogenic front/leading edge of the retina, this is where one would expect a more relevant role of Sdc2 (and perhaps expression). Ideally, an image of a whole retinal flank with immunostaining for Sdc2 and higher magnification insets of the different regions should be provided.

REPLY 2: We have added expression of Sdc2 in tip cells (Fig. S1E-lower left inset). Unfortunately, quality of all available Sdc2 antibodies is poor (not surprisingly) and we cannot have the highest staining quality. We are developing a custom-made antibody against mouse Sdc2 that should permit better staining and WB/IP analysis in our future works.

R#3 Reply 2: The authors have plenty of space to show the full size angiogenic front and tip cells figures. Why a small inset is provided? The quality of the signal is good enough.

Reply 3: A larger field has been added

2. None of the images presented in the manuscript have scale bars.

Reply: scale bars have been added

R#3 Reply: images S1D; E and S2A, D still do not have scale bars.

Reply 2: Scale bars have been added.

R#3 Reply 2: Ok

3.No data has been presented regarding the endothelial expression of Sdc4.

Reply: There are numerous publications demonstrating high Sdc4 expression in ECs, including in the retinal vasculature. This information, along with citations to most recent papers, has been added to the text (page 6, lines 132-133).

4. Authors do not attribute the decrease in vessel density to enhanced vascular pruning in the Sdc2 mutants because "collagen IV staining did not detect empty sleeves". Since collagen empty sleeves are indeed detectable in Fig S2A, we infer that the authors meant that there were no differences in the frequency of empty sleeves between control and mutant retinas. However, no quantification data is presented.

Reply: Quantification has been added (Supplementary Figure 3B)

5. Quantifications on the density of vascular networks in the skin and kidney are not presented.

Reply: We have decided to remove these data and to focus on retina vascular development and hindlimb ischemia, two settings in which relative contribution of VEGFA versus other factors is extremely well characterized.

6. Although the authors claim the role of Sdc2 on angiogenesis is mediated through the VEGFA165-VEGFR2 axis, there is also evidence that Sdc2 also interacts with integrins (Exp. Cell. Res (2000), Vol.256, 434 and Biophysics Res Commun (2009), Vol385, 231), which is important for cell migration and adhesion. In fact, there are several papers that show VEGFA-mediated functions dependent on integrins (Blood 2012 120:4892-4902; J Biol Chem. 2011 286: 1083/1092; JBC 2007 282, 15187-15196; Embo J 1999;18:882-892; Mol Cell 2000;6:851-860; Circ Res 2007;101:570-580). To assess integrin signaling, immunostaining for p-FAK or active-integrinB1 could be used.

Reply: We agree that integrins are important in VEGFR2 signaling. How that integrates with Sdc2 is less clear and is the subject for further studies. Our data clearly show that Sdc2 is essential for formation of VEGFA165-VEGFR2-Sdc2 complex which, in turn, is required for maximal VEGFR2 phosphorylation. Whether an integrin, such as beta-2 is involved is not certain. Integrins activation is one of the VEGFR2 downstream target and likely to be affected as the other downstream targets. This possibility and relative references are added to discussion as suggested. (pages 14, lines 317-323).

R#3 Reply: AS mentioned in the answer to major concern 1, since the decrease in endothelial Vegfr2 signaling in vivo is still not clear, and as the authors also mention in the discussion ("we have not investigated whether Sdc2 deletion differentially affects various VEGFR2 downstream signaling pathways (e.g. PLC γ /ERK, AKT/Src, integrin activation, etc..."), it should be analyzed how ERK/AKT/Src/Integrin signaling pathways are affected in Sdc2 mutants, to provide a better mechanistic explanation for the defects observed in EC migration/sprouting/proliferation.

Reply: Analysis of all major VEGFA-VEGFR2 downstream effectors (ERK, AKT, SRC, Integrin- β 3) have been added (Lines 169-172, Fig. S5A-B).

R#3 Reply 2: Ok

7. In vivo validation of the in vitro co-IP data would also be interesting, using for example the tissues from animals.

Reply: In the proteoglycans world, IP and WB experiments are severely limited by lack of working

antibodies against endogenous proteins. Sdc2 is no exceptions. This is the reason why most studies in this field rely on the use of epitope-tags and overexpression systems. There is no possibility of getting a Sdc2-specific IP from a tissue lysate.

8. It would be important to check by western blot if Sdc4 is regulated in the absence of Sdc2 and vice-versa.

Reply: As requested, we now show Sdc4 blot in Figure 3A (Sdc2^{-/-} ECs) and Sdc2 blot in Figure 3D (Sdc4^{-/-} ECs). We did not detect any evident upregulation.

9. In Fig3H and Fig 4G, it appears that Sdc2 and Sdc4 have opposite effects on p-VEGFR2 (comparing to cells transduced with only Ad-GFP). Could the authors comment on this?

Reply: The only statistically significant effect we see in these studies is when Sdc2 is reintroduced into Sdc2^{-/-} ECs.

10. All interaction assays were performed in cells transduced with adenovirus and expressing different mutated forms of Sdc2/4. It is known that over-expression of proteins can sometimes lead to unspecific protein interactions. How do the total levels of these proteins compare to the endogenous levels? Can the same experiments be performed with CRISPR/Cas9 induced mutations in Sdc2?

Reply: We agree this is an important point. IP experiments following adenovirus transduction (at MOI = ~ 1-2) were performed with minimal overexpression which was similar to syndecans endogenous level (supplementary figures 6A-B) CRISPR/Cas9 mutagenesis would be a whole new study and approach would still be limited by lack of good antibody against endogenous protein without epitope-tag.

11. Although the protein sequence of the D1 domains of Sdc2 are well conserved among species, I wonder whether the glycosylation events are also well conserved, since this is known to vary depending on cell context. It would be important to confirm that glycosylation pattern in mouse endothelial cells are identical to those described in the manuscript and ideally in the absence of overexpression of the proteins.

Reply: That would be a good study to do. Unfortunately, high-quality antibody for affinity purification of endogenous syndecans do not exist. Nevertheless, the point of testing a different species is well taken and we have repeated HS chain analysis of both sdc2 and sdc4 isolated from a mouse endothelial cell line. We now show that also mouse ECs generate higher level of 6-O-sulfation in Sdc2 HS chains than in Sdc4 HS chains (Supplementary Figure 6E). This is similar to what we observed in HUVECs